# Landscape of the Epstein-Barr virus-host chromatin interactome and gene regulation

Simon Zhongyuan Tian [1,2,7 ✉], Yang Yang [1,2,7], Duo Ning [1,2,7], Ting Yu [2,3,7], Tong Gao [2], Yuqing Deng [2], Ke Fang [4], Yewen Xu [1,2], Kai Jing [1,2], Guangyu Huang [2], Gengzhan Chen [2], Pengfei Yin [2], Yiming Li [4 ✉], Fuxing Zeng [2,3 ✉], Ruilin Tian [5,6 ✉] & Meizhen Zheng [1,2 ✉]

## Abstract

The three-dimensional (3D) chromatin structure of Epstein–Barr virus (EBV) within host cells and the underlying mechanisms of chromatin interaction and gene regulation, particularly those involving EBV's noncoding RNAs (ncRNAs), have remained incompletely characterized. In this study, we employed state-of-the-art techniques of 3D genome mapping, including protein-associated chromatin interaction analysis with paired-end tag sequencing (ChIA-PET), RNA-associated chromatin interaction technique (RDD), and super-resolution microscopy, to delineate the spatial architecture of EBV in human lymphoblastoid cells. We systematically analyzed EBV-to-EBV (E–E), EBV-to-host (E–H), and host-to-host (H–H) interactions linked to host proteins and EBV RNAs. Our findings reveal that EBV utilizes host CCCTC-binding factor (CTCF) and RNA polymerase II (RNAPII) to form distinct chromatin contact domains (CCDs) and RNAPII-associated inter-action domains (RAIDs). The anchors of these chromatin domains serve as platforms for extensive interactions with host chromatin, thus modulating host gene expression. Notably, EBV ncRNAs, especially Epstein–Barr-encoded RNAs (EBERs), target and interact with less accessible regions of host chromatin to repress a subset of genes via the inhibition of RNAPII-associated chromatin loops. This process involves the cofactor nucleolin (NCL) and its RNA recognition motifs, and depletion of either NCL or EBERs alters expression of genes crucial for host infection control, immune response, and cell cycle regulation. These findings unveil a sophisticated interplay between EBV and host chromatin.

**Keywords** EBV; ncRNA; 3D Genome Mapping; Host–Virus Interactions; Spatial Architecture
**Subject Categories** Chromatin, Transcription & Genomics; Microbiology, Virology & Host Pathogen Interaction

## Introduction

The intricate higher-order structure of chromatin within the cell nucleus has been extensively elucidated through the efforts of the four-dimensional nucleome (4DN) project (The 4D Nucleome Network et al, 2017). This initiative has confirmed that the ~3.1 billion base pairs (3.1 Gb) of the human genome, which would extend linearly over roughly 2 m, are remarkably compacted into the small confines of a 10-μm-diameter cell nucleus (Alberts et al, 2002). This compaction forms complex three-dimensional (3D) architectures essential for efficient genomic function and regulation (Han et al, 2024). High-throughput 3D sequencing methods and advanced imaging techniques have revealed that chromatin within cell nuclei is organized into topologically associating domains (TADs), which are fundamental units of higher-order chromatin structure playing a crucial role in genomic organization and gene regulation (Pombo and Dillon, 2015).

A central factor in the organization and assembly of TADs is the CCCTC-binding factor (CTCF), a highly conserved zinc finger protein (Rao et al, 2014). Chromatin interaction analysis with paired-end tag sequencing (ChIA-PET) (Li et al, 2017) targeting CTCF has shown that CTCF-mediated chromatin contact domains (CCDs) align closely with TADs, reinforcing the idea that CTCF, along with cohesin, is primarily responsible for shaping chromatin topology in the human nucleome (Tang et al, 2015). Convergent CTCF loops define the structures and boundaries of CCDs/TADs, while tandem loops within these regions facilitate promoter-enhancer long-range interactions mediated by RNA polymerase II (RNAPII) (Tang et al, 2015; Li et al, 2012). These interactions contribute to RNAPII-associated interaction domains (RAIDs) (Zheng et al, 2019), where many genes are located close to chromatin interaction anchors serving as structural foci, creating a coordinated framework of chromatin topology that supports transcription regulation through the interplay between CTCF foci and the RNAPII machinery (Tang et al, 2015).

Emerging research has highlighted the significant role of noncoding RNAs (ncRNAs) in both the assembly and functional

[1]Shenzhen Key Laboratory of Gene Regulation and Systems Biology, School of Life Sciences, Southern University of Science and Technology, 518055 Shenzhen, Guangdong, China. [2]Department of Systems Biology, School of Life Sciences, Southern University of Science and Technology, 518055 Shenzhen, Guangdong, China. [3]Institute for Biological Electron Microscopy, Southern University of Science and Technology, 518055 Shenzhen, Guangdong, China. [4]Department of Biomedical Engineering, Southern University of Science and Technology, 518055 Shenzhen, Guangdong, China. [5]Department of Medical Neuroscience, School of Medicine, Southern University of Science and Technology, 518055 Shenzhen, Guangdong, China. [6]Key University Laboratory of Metabolism and Health of Guangdong, Southern University of Science and Technology, 518055 Shenzhen, Guangdong, China. [7]These authors contributed equally: Simon Zhongyuan Tian, Yang Yang, Duo Ning, Ting Yu. ✉E-mail: tianzy3@sustech.edu.cn; liym2019@sustech.edu.cn; zengfx@sustech.edu.cn; tianrl@sustech.edu.cn; zhengmz@sustech.edu.cn

dynamics of these nuclear domains (Quinodoz et al, 2021). The identification of thousands of ncRNAs within mammalian genomes, many of which are intricately involved in nuclear structure formation, solidifies ncRNAs as key players in defining the architecture and regulatory frameworks of the nucleus (Rinn and Guttman, 2014). In addition, many RNAs have been found to be enriched in enhancer–promoter regions, where they participate in gene transcription regulation to some extent (Zhou et al, 2019). This discovery underscores a more profound layer of genetic regulation and structural organization within cells, emphasizing the critical role of ncRNAs in maintaining cellular integrity and functionality.

The presence and organizational state of exogenous double-stranded DNA (dsDNA), such as Epstein–Barr Virus (EBV) or extrachromosomal DNA, within the host cell nucleus raise intriguing questions. How do these exogenous entities integrate into or interact with the host chromatin landscape, and what structural configurations do they assume? Previous research has successfully delineated the three-dimensional architecture of EBV within the chromatin environment and identified that CTCF and other protein factors can bind to specific sites on the EBV genome, highlighting its effects on host gene regulation (Morgan et al, 2022; D et al, 2023; Arvey et al, 2012). Despite these advancements, substantial gaps remain in our comprehensive understanding of chromatin domains and long-range interactions directly orchestrated by the architectural protein CTCF and the transcriptional regulator RNAPII, as revealed by antibody-enriched 3D genome technologies like ChIA-PET. In particular, studies that integrate omics technologies with advanced imaging methods to explore these interactions and structures are especially sparse. Furthermore, EBV-derived ncRNAs, including Epstein–Barr encoded RNAs (*EBERs*), BamHI-A rightward transcripts (*BARTs*), and a viral small nucleolar RNA1(*v-snoRNA1*), play significant roles in gene regulation and are vital to EBV's life cycle, immune evasion, latency maintenance, and host cell transformation (Lee, 2021; Notarte et al, 2021). However, it remains unclear whether these exogenous ncRNAs exert their functions through long-range chromatin interactions. Critical questions persist about how EBV exploits host cellular machinery, particularly the key architectural protein CTCF, to establish sophisticated chromatin architecture, interact with host components associated with ncRNA, and regulate gene expression via the transcriptional regulator RNAPII.

This study aims to address these questions by utilizing approaches of high-throughput sequencing and super-resolution microscopy to investigate the 3D architectural landscape and transcriptional orchestration of EBV within GM12878 human lymphoblastoid cells. Our objective is to elucidate the viral strategies that hijack host chromatin dynamics and regulatory networks, thereby enhancing our understanding of viral latency and oncogenesis. We dissect the ncRNA- and protein-associated chromatin DNA interactome between EBV and host GM12878 cells, employing the RNA-associated chromatin interaction technique (RDD) developed in our laboratory (Tian et al, 2024) to specifically investigate chromatin interactions associated with EBV-produced ncRNAs.

Our findings reveal the 3D structure of EBV and the EBV–host chromatin interactome landscape shaped by EBV ncRNAs. We uncover unique chromatin structures associated with *EBERs* ncRNAs and their effects on genes related to viral infection,

immune response, and cell cycle regulation, mediated by altered RNAPII-associated chromatin loops. These interactions occur via various cofactors, involving the RNA recognition motif (RRM) of nucleolin (NCL)—a multifunctional protein implicated in multiple pathways, from viral interactions at the cellular membrane to essential ribosomal RNA processing in the nucleolus (Mongelard and Bouvet, 2007). This research not only offers novel strategies and methodologies for investigating virus-host interactions but also provides fresh insights into the intricate dynamics of viral pathogenesis and host defense mechanisms.

## Results

### EBV hijacks host CTCF and RNAPII to construct a 3D structure

To elucidate the mechanisms by which EBV exploits host cellular machinery for chromatin organization and the transcription of coding and noncoding RNAs, and to understand the impact of these ncRNAs on chromatin architecture and gene regulation, we generated a detailed chromatin interactome map. This map was created using targeted antibodies or biotinylated nucleotides (RNA probe) to enrich chromatin complexes associated with protein factors or ncRNAs, followed by a proximity ligation assay to capture chromatin contacts (Figs. 1A and EV1A). This approach enables us to comprehensively understand the interplay between viral ncRNAs and host chromatin architecture, including interactions of EBV-to-EBV (E–E), EBV-to-host (E–H), and host-to-host (H–H). Consequently, this provides structural insights into the molecular orchestration of gene regulation by EBV.

For our experimental model, we selected the human GM12878 lymphoblastoid cell line (LCL), known for its EBV latency III status (Jiang et al, 2017). Chromatin accessibility studies (ATAC-seq) (Grandi et al, 2022) revealed that approximately ten EBV genomic loci display higher accessibility, with significant signals detected throughout the entire EBV genome (Fig. 1B, upper panel). By comparing chromatin states in host GM12878 cells, we found that ATAC-seq peaks on the EBV episome correspond to regions with higher signal intensity than host chromatin in Chromatin State 1 (Active Transcription Start Site, TSS). Even non-peak regions of the EBV episome (EBV-base) exhibited higher signal intensity than Chromatin State 2 (Flanking Active TSS) of the host genome (Fig. EV1B,C). These findings indicate that the dsDNA EBV episome within the host maintains a relatively open chromatin structure across its entire genome, in contrast to the host chromatin, which exhibits distinct regions of heterochromatin and euchromatin (Thurman et al, 2007) (Figs. 1B, upper panel and EV1B,C). This observation is consistent with previous studies showing that histone modifications associated with facultative heterochromatin (H3K27me3) and constitutive heterochromatin (H3K9me3) are generally low across the EBV genome (Arvey et al, 2013). These results suggest that the EBV genome is chromatinized with a density similar to that of the host genome but remains in a relatively open state.

Further analysis revealed that these highly accessible loci correspond to the active A compartment, while less accessible loci correspond to the inactive B compartment, as further defined

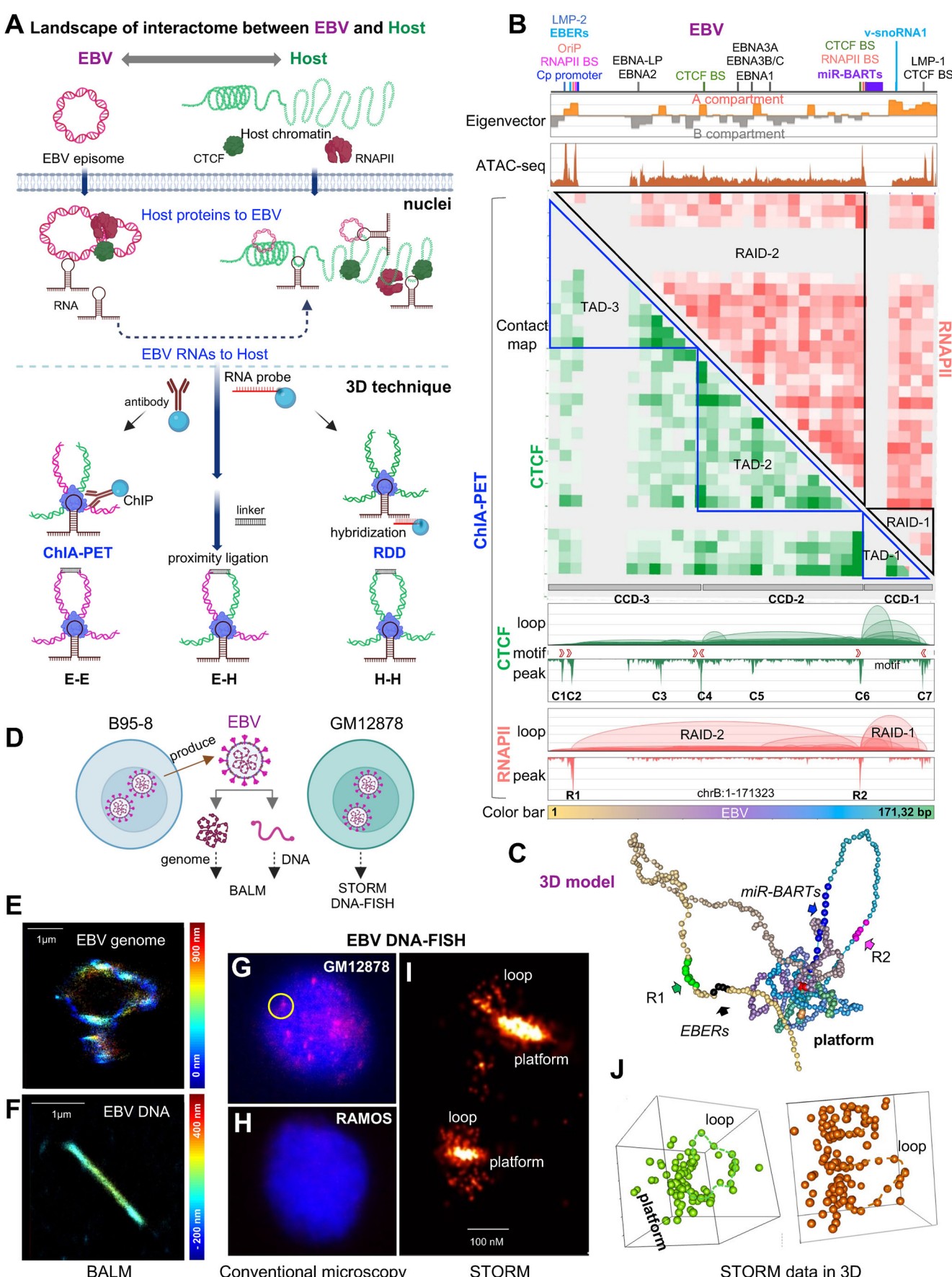

**Figure 1. The three-dimensional (3D) structure of EBV.**

(A) Schematic illustrating RNA- and protein-associated chromatin DNA–DNA interactions using 3D techniques. EBV hijacks host RNAPII and CTCF for its structure and transcribed RNAs to interact with host chromatin. RDD represents RNA-associated chromatin interactions via RNA probe hybridization, while ChIA-PET denotes protein-associated chromatin interactions via chromatin immunoprecipitation (ChIP). E–E indicates EBV–EBV interactions, E–H represents EBV–host interactions, and H–H indicates host–host interactions. (B) Screenshot of EBV chromatin with key genome annotations at the top, followed by A/B compartments identified by eigenvector, ATAC-seq for chromatin accessibility, heatmaps for chromatin interactions, and ChIA-PET results showing loops, motifs, and peaks for RNAPII and CTCF. Topologically associating domains (TADs) align with CTCF-mediated chromatin domains (CCDs). RAIDs indicate RNAPII-associated interacting domains. Binding sites (BS) for CTCF and RNAPII are labeled C1 to C7 and R1, R2, respectively, with CTCF motif directions indicated by arrows. "chrB" refers to the EBV episome. The color bar represents the EBV genome from start to end. (C) The three-dimensional (3D) model depicts the higher-order structure of EBV chromatin, showing a platform with structural foci tethered by chromatin loop anchors and two outward loops. Bead colors correspond to linear genome positions shown in the color bar (B). Arrows indicate specific regions. (D) Schematic of the sample preparation for EBV genome and linear DNA outside cells, and episome within cells, for binding-activated localization microscopy (BALM) and stochastic optical reconstruction microscopy (STORM) experiments. (E, F) Example BALM images of EBV genome (E) and purified linear DNA (F) with color scale indicating the z range. (G) Conventional microscopy images showing DNA-FISH (red) of EBV and DAPI (blue) in human GM12878 cells. (H) Conventional microscopy images of EBV-negative RAMOS cells as a negative control. (I) STORM images with a zoomed-in view of EBV foci, highlighted by a yellow circle (G). (J) Representative 3D visualizations of the EBV structure derived from STORM data. Source data are available online for this figure.

using eigenvectors ("Methods", Fig. 1B, upper panel). They are organized into three CCDs, also known as TAD structures, and two principal RAIDs identified from ChIA-PET data (Fig. 1B). Some chromatin loops associated with RAIDs were also detected using H3K27ac HiChIP, a method similar to ChIA-PET, suggesting the presence of active enhancer loops in these regions (Ding et al, 2022).

The CCDs feature seven prominent CTCF-binding peaks (designated as C1 to C7), consistent with previous data (Ding et al, 2022), with the boundary peak exhibiting a convergent motif orientation (Fig. 1B). The RAIDs feature two RNAPII binding peaks (designated as R1 and R2 in Fig. 1B): R1 is located at the OriP region, which is essential for the replication and maintenance of the EBV episome in latently infected cells, and the Cp promoter, which is crucial for the transcription of latent genes necessary for viral latency and host cell transformation (Wensing et al, 2001; Evans et al, 1996). This region is also confirmed to be a prominent enhancer (Maestri et al, 2023). R2 is located at the BamA region, which plays a critical role in the transcriptional regulation of EBV latent genes (Hoebe et al, 2013).

These contact anchors of CCDs and RAIDs create a coordinated framework further supported by a 3D model of EBV genome in the host generated using CTCF and RNAPII ChIA-PET data ("Methods", Figs. 1C and EV2A). In this model, the CTCF and RNAPII binding sites, along with anchor regions, are intricately interconnected, forming a platform of high accessibility, while the remaining EBV genome loops outward with lower accessibility (Fig. 1C; Movie EV1).

## Super-resolution microscopy confirms the spatial structure of EBV

To validate the 3D structure of EBV as predicted by our omics analyses (Fig. 1C), we utilized cutting-edge super-resolution microscopy techniques. We employed stochastic optical reconstruction microscopy (STORM) (Huang et al, 2008) with DNA fluorescence in situ hybridization (DNA-FISH) using EBV-whole genome probes for in vivo observation of the EBV chromatin structure within GM12878 cells. Additionally, we applied binding-activated localization microscopy (BALM) (Ries et al, 2013) with SYTOX™ Orange nucleic acid staining for in vitro analysis of EBV genome, which was obtained through secretion from B95-8 cells (Gradoville et al, 2002). Meanwhile, aliquots of the samples were

used to examine EBV DNA that had been de-crosslinked and purified from this secreted virion (Fig. 1D).

Observations of the in vitro EBV genome obtained from B95-8 cells displayed a ring-shaped configuration (Figs. 1E and EV2B; Movie EV2), in contrast to the linear structure of purified EBV DNA (Figs. 1F and EV2C; Movie EV3). This supports that EBV genome also is organized in a higher-order structure. We noted numerous EBV episomes dispersed throughout the GM12878 nuclei, notably forming a pronounced cloud-like aggregation absent in EBV-negative LCL RAMOS cells, as revealed by employing DNA-FISH under conventional microscopy (Figs. 1G,H and EV2D). A more detailed examination under STORM revealed the EBV episome as a light cloud with dispersed dots arranged in a loop shape (Figs. 1I and EV2D, right panel). This structure somehow resembles the ring-shaped configuration tethered on the platform observed under BALM and the 3D structure modeled from omics data (Fig. 1C).

We then took a zoomed-in view of the cloud-like aggregation area (Fig. EV2D, left panel) under super-resolution STORM (Fig. EV2D, middle panel) and simulated it as a 3D structure (Fig. EV2E), which disclosed a unique cluster formation of ring-shaped dots encircling a central luminescence, mirroring the anticipated 3D model structure (Fig. 1J).

Subsequent analytical depth was achieved through the Bayesian Information Criterion (BIC) within the Gaussian Mixture Models (GMM) clustering algorithm (Scrucca et al, 2016), identifying 58 distinct clusters corresponding to episomes within the cloud-like formation (Fig. EV2F). Each cluster, with an estimated volume of $0.02\,\mu m^3$, comprised ~25 dots that corresponding to a looping structure (Fig. EV2G). These results align with the distribution of EBV copy number in LCLs across populations from the 1000 Genome Project (Mandage et al, 2017).

This comprehensive super-resolution microscopy data lends substantial support to the omics-based modeling of the EBV genome, affirming the virus's spatial configuration within host cells.

## The platform of EBV 3D structure frequently contacts with the host chromatin

Our research further uncovers that RNAPII and CTCF extend beyond their roles in the EBV 3D structure; they are instrumental in mediating interactions between EBV and host cells (E–H, Fig. 2A). Unlike previous studies that defined the EBV regulome within the host genome by linking RNAPII ChIA-PET-detected

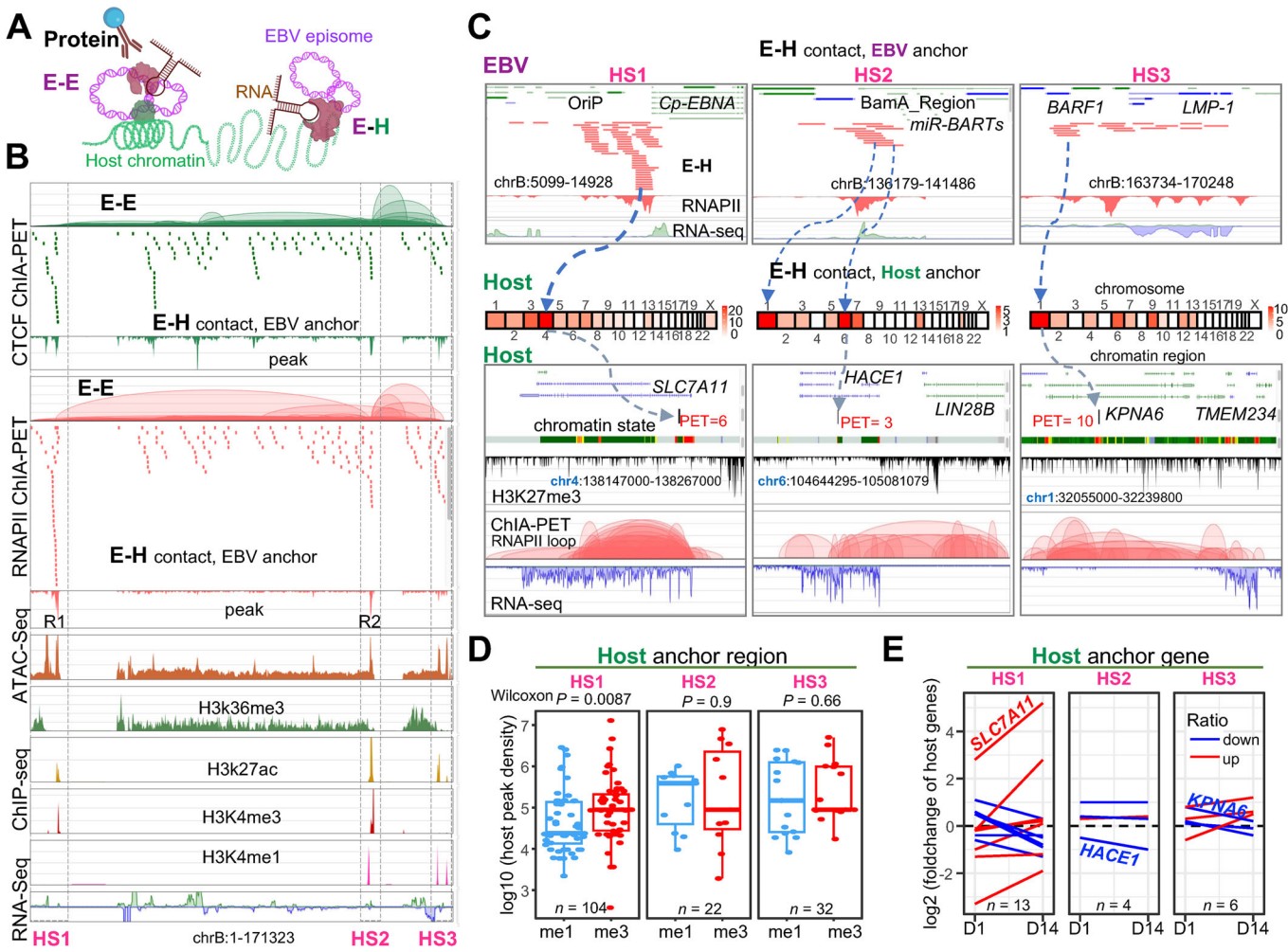

**Figure 2. Host protein-associated chromatin DNA–DNA interactions involving EBV–EBV (E–E) and EBV–host (E–H).**

(A) Diagram illustrating host protein-associated chromatin interactions between E–E and E–H. (B) BASIC Browser visualization of the EBV genome showing host protein-associated E–E and E–H interactions. HS1, HS2, and HS3 denote hotspots and are highlighted with dashed squares. The small bars represent the EBV-side anchors of the E–H contacts. (C) BASIC Browser visualization of EBV hotspots (top) in contact with the represented host chromatin loci (bottom) using RNAPII ChIA-PET data. The dashed blue line indicates the contacts from EBV HS to host chromosome, and the gray dashed line denotes the represented host chromatin region. PET represents the contact frequency. In addition, chromatin state and RNA-seq data are displayed. (D) Box plots illustrating the binding density of H3K4me1 (me1, enhancer marker) and H3K4me3 (me3, promoter marker) at the host genomic region contacted with the EBV HS region. Each box plot highlights the median (inside line), the 25th to 75th percentiles (box), and the minima/maxima values within 1.5× the interquartile range (IQR) of the box (whiskers). Two-sided Wilcoxon tests are performed. (E) Spaghetti plots depict the fold change of the host gene targeted by the EBV hotspot (HS1, HS2, and HS3) in EBV-infected human B lymphocytes on Day 1 (D1) and Day 14 (D14) compared to Day 0 (D0) using RNA-seq data. Each line represents individual genes. Red lines indicate an upregulation of target gene expression post-EBV infection (D1, D14) compared to pre-infection (D0), while blue lines indicate downregulation. The dashed line highlights no changes. Source data are available online for this figure.

enhancers to regions occupied by any of the four EBV-encoded nuclear antigens (EBNAs) or five NF-κB subunits (Jiang et al, 2017), we directly observed physical interactions between EBV DNA and host DNA (E–H). The capture of 194 E–H contacts, concentrated at three critical hotspots (HS1-3, Fig. 2B) at the EBV 3D platform structure, is characterized by open chromatin configurations and a high density of histones and proteins, highlighting their regulatory potential.

HS1 is notable for its strong promoter activity in the OriP region (close to the Cp promoter) and its overlap with RNAPII binding site (R1) as described previously. It establishes contacts with the chromatin of human GM12878 cells, specifically targeting the *SLC7A11* gene, which is involved in inhibiting viral infection (Ren

et al, 2022), in the region on chromosome 4 via RNAPII-associated chromatin loops (Figs. 2C, left panel and EV3A). In addition, it interacts with the *PRRX1* gene, associated with tumor malignancy (Joko et al, 2020). HS2, characterized by both enhancer and promoter functions in the BamA region and its overlap with RNAPII binding site (R2), forms connections with host RNAPII-mediated chromatin loops enriched on chromosomes 1 and 6 (Figs. 2C, middle panel and EV3A). This includes interaction with the potential tumor suppressor gene *HACE1* (El-Naggar et al, 2019), indicating a complex interplay between viral and host genomic elements in cancer regulation. HS3, marked by enhancer activity around the *LMP-1* region, establishes contacts with chromosome 1, including the influenza virus-associated gene

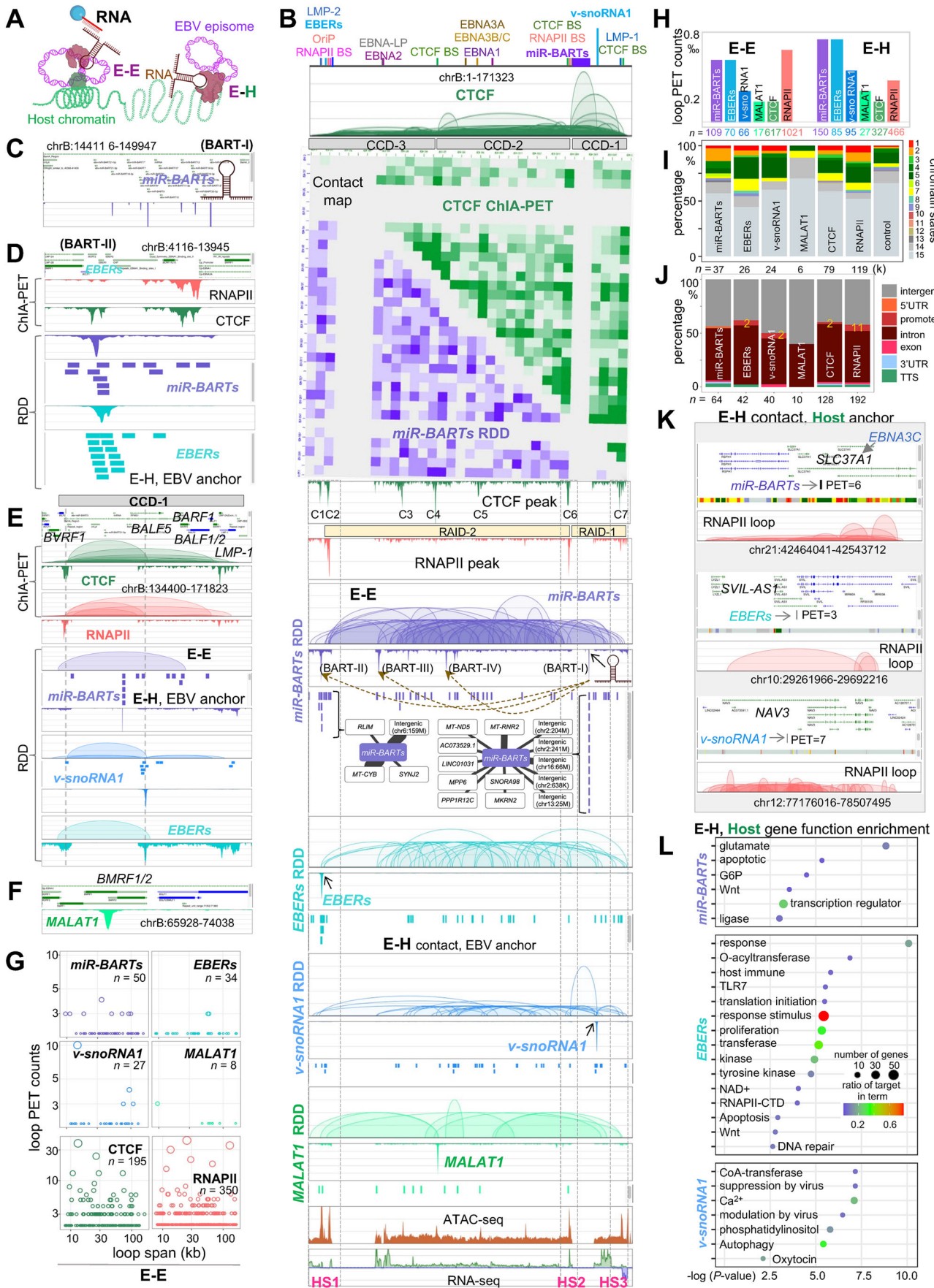

**Figure 3. EBV ncRNAs-associated chromatin DNA–DNA interactions within EBV genome (E–E) and between EBV-host (E–H).**

(A) Schematic illustrating ncRNA-associated chromatin DNA–DNA interactions within the EBV genome (E–E) and between EBV and the host (E–H). (B) Visualization of ncRNA-associated chromatin interactions for the EBV genome. Genome positions are labeled at the top, followed by CTCF-mediated chromatin loops and CCDs, heatmaps for CTCF-mediated and *miR-BARTs*-associated contacts, CTCF-binding peaks (C1 to C7), along with RAIDs and RNAPII binding peak. RDD results for loops with EBV (E–E), targeted peaks, and contacts between EBV and host (E–H) are also provided. In addition, ATAC-seq for chromatin accessibility and RNA-seq for gene expression levels are included for reference. BS: binding site; Arrows: the transcription site of ncRNAs. The inset network displays the hotspot contacts with the host chromatin (shown as genes or loci). (C) Zoomed-in view of the transcription site of *miR-BARTs* (BART-I in (B)). 'chrB' refers to the EBV episome. (D) Zoomed-in view of BART-II region within HS1 (B). (E) Zoomed-in view of the CCD-1 region (B). (F) Zoomed-in view of the host ncRNA *MALAT1*-targeted EBV binding site (B). (G) Scatter plots show the interaction frequency between E–E. (H) Chromatin interaction distribution percentage of E–E and E–H contacts. (I) Chromatin states of the host anchor from E–H contacts. Chromatin state- (1) Active Transcription Start Site (TSS), (2) Flanking Active TSS, (3) Transcription at gene 5′ and 3′ ends, (4) Strong transcription, (5) Weak transcription, (6) Genic enhancers, (7) Enhancers, (8) ZNF genes and repeats, (9) Heterochromatin, (10) Bivalent/Poised TSS, (11) Flanking Bivalent TSS/Enhancers, (12) Bivalent Enhancer, (13) Repressed PolyComb, (14) Weak Repressed PolyComb, (15) Quiescent/Low. (J) Genomic features of the host anchor from E–H contacts. (K) Examples of E–H chromatin interactions on the host side are displayed in the BASIC Browser. (L) GO term enrichment analysis of host genes enriched by E–H contacts. Source data are available online for this figure.

*KPNA6* (Yu et al, 2022) (Figs. 2C, right panel and EV3A). HS3 is also the anchor overlapped with the EBV *BARF1* gene, which is selectively expressed in EBV-driven carcinomas (Hoebe et al, 2013). Beyond the hotspot contact regions, the EBV *BDLF4* region, which is required for efficient expression of viral late lytic genes (Watanabe et al, 2015), connects to the host's enhancer region, forming loops with the promoter of the human *PTEN* gene (Fig. EV3A,B), a major tumor suppressor gene involved in *miR-BART1*-induced tumor metastasis in nasopharyngeal carcinoma (Cai et al, 2015).

To validate the regulatory functions of these hotspots with host chromatin, we then analyzed the promoter and enhancer activity targeted by these three hotspots. HS1 with strong promoter activity contacts host chromatin also characterized by higher promoter activity (as indicated by higher H3K4me3 binding density), while HS2 and HS3 exhibit both promoter and enhancer functions (Fig. 2D). By further analyzing these hotspots targeting host genes using RNA-seq data from LCLs infected with EBV at 0 days, 1 day, and 14 days (Mrozek-Gorska et al, 2019), we observed that certain genes continuously increased in expression, while others consistently decreased during the EBV infection process, especially in those contacted by HS1 (Fig. 2E). This underscores the functional roles of EBV hotspots in modulating host gene expression through which EBV interacts with the host genome via RNAPII and CTCF structural foci.

## EBV ncRNA-related chromatin DNA–DNA interactions within EBV genome (E–E)

Upon infection, EBV utilizes the host cell's systems to organize into a 3D structure, leveraging the machinery for chromatin assembly and the transcription of both coding and noncoding RNA genes within the host cell. Our study focused on the interaction profiling of three EBV ncRNAs—*v-snoRNA1*, *miR-BARTs* (*miR-BART1-21*), and *EBERs* (*EBER1,2*)—alongside the host ncRNA *MALAT1* within the GM12878 cell line. Generating at least two replicates with high reproductivity for each RNA-associated DNA–DNA interactions (RDD) dataset allowed for a comprehensive and comparative analysis of ncRNA-targeted genomic loci and chromatin interactions (Fig. EV4A–F; Datasets EV1 and EV2). Further analysis of the combined RDD libraries provided insights into the interactions of EBV ncRNAs with the viral genome, revealing intraviral interactions (E–E) (Figs. 3A and EV4D–F). We observed that ncRNAs

associate chromatin contacts do not form distinctive structure like CCDs or RAIDs (Figs. 3B and EV4G). Post-transcription from chromatin loop region within CCD-1, *miR-BARTs* specifically targets four major regions of the EBV genome, displaying distinct peaks (BART-I, BART-II, BART-III and BART-IV). Apart from its transcribed site (BART-I, Fig. 3B,C), one peak (BART-II) coincides with the *EBERs* promoter region located at HS1 (Fig. 3B,D). Another peak is found at the transcriptional end site of the *EBNA-LP* gene (BART-III) within CCD-3, and a third peak is near the *BSLF1* gene promoter (BART-IV) within CCD-2. This phenomenon suggests ncRNA's distal trans targeting to EBV genomic locations. Additionally, *v-snoRNA1*, which is also transcribed from CCD-1 and serves as an anchor gene, forms a high-frequency loop structure at its transcription site, overlapping with the boundary of the chromatin topologically associated domain CCD-1 (HS2, Fig. 3B,E). The behavior of *miR-BARTs* and *v-snoRNA1* further demonstrates the specificity of EBV ncRNAs. *EBERs*, transcribed from HS1 region (Fig. 3B), target the *v-snoRNA1* transcription region and also loop with the CCD-1 boundary within HS2 (Fig. 3E). The host's endogenous *MALAT1* showed a distinct binding peak on the EBV genome at the *BMRF1/2* gene (Fig. 3F), overlapping with the CTCF-binding site C4 (Fig. 3B). This indicates that *MALAT1* has potential regulatory functions involving the DNA polymerase processivity factor of EBV (*BMRF1*) (Tsurumi, 1993), which is important for viral DNA replication, and the DNA packaging process (*BMRF2*) (Xiao et al, 2008), which is key for the assembly of the viral capsid. These specific binding and contact properties underscore the functional specificity of ncRNAs.

It's crucial to highlight that despite utilizing over ten times the number of cells in our RDD experiments compared to the ChIA-PET experiments, we detected fewer contacts associated with ncRNAs. Typically, ncRNA-associated interactions within the EBV genome exhibit lower frequency and intensity than those mediated by protein factors such as RNAPII and CTCF (Fig. 3G). Unlike the concentrated interaction hotspots observed with protein factor RNAPII and CTCF, ncRNA interactions are generally more dispersed (Fig. 3G).

Among EBV ncRNAs with frequent interactions involving more than three paired-end tags (PETs), *v-snoRNA1* stands out by forming a high-frequency short-distance loop that aligns with a CTCF loop within CCD-1 (Fig. 3E). *miR-BARTs* is linked to seven loops of varying distances, whereas *EBERs* are associated with only two long-distance interactions. In addition, the endogenous host ncRNA, *MALAT1*, shows the fewest contacts and the lowest interaction frequency (Fig. 3B,G).

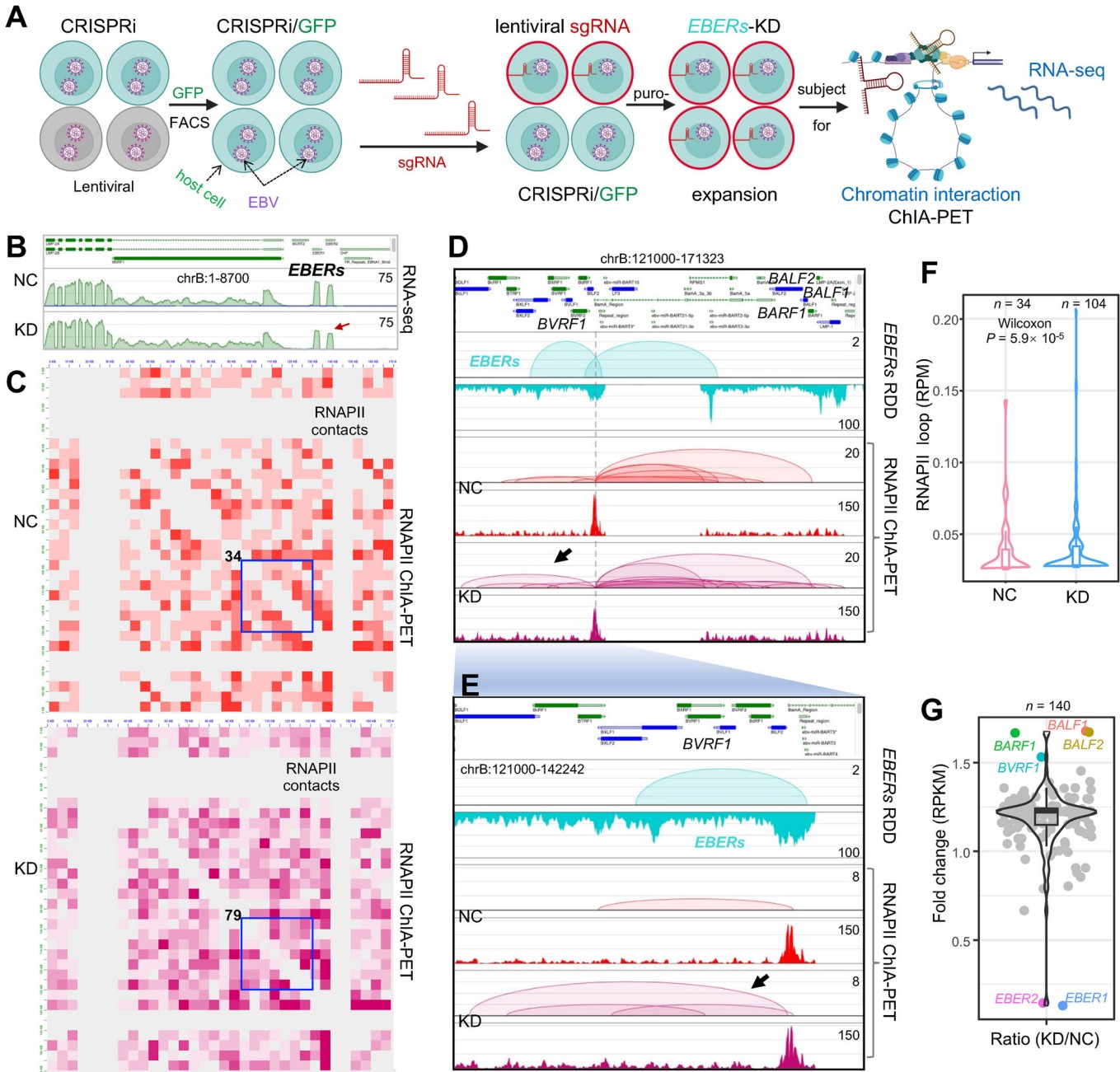

**Figure 4. EBV ncRNA *EBERs* repress viral gene expression through RNAPII-associated chromatin loops.**

(A) Schematic of CRISPRi experiments. (B) RNA levels of *EBERs* in *EBERs*-NC and *EBERs*-KD cells. "chrB" refers to the EBV episome. (C) Heatmaps display the contacts of RNAPII-mediated chromatin interactions in *EBERs*-NC and *EBERs*-KD cells. The blue squares are used to indicate significant differences. The numbers represent the interactions captured within the square. (D) BASIC Browser view showing the chromatin loop associated with RNAPII in *EBERs*-NC and *EBERs*-KD cells. The arrows highlight the significant changes. (E) Zoomed-in view of (D). (F) Violin-Box plots present the chromatin interaction PET counts from RNAPII ChIA-PET in *EBERs*-NC cells and *EBERs*-KD cells at the genomic region of EBV. Two-sided Wilcoxon tests are performed. (G) The Violin-Box-Jitter plot presents the fold change of EBV genes in *EBERs*-KD cells versus *EBERs*-NC cells. Each box plot highlights the median (inside line), the 25–75th percentiles (box), and the minima/maxima values within 1.5× the interquartile range (IQR) of the box (whiskers). Source data are available online for this figure.

In summary, our analysis reveals distinct interaction patterns of EBV and the host ncRNAs within the EBV genome, characterized by varying frequencies and distances. These patterns differ significantly from those mediated by protein factors like RNAPII and CTCF, indicating unique modes of gene regulation by ncRNAs in the context of EBV infection.

## EBV ncRNA-related chromatin DNA–DNA interactions between EBV and host (E–H)

Building on our detailed analysis of intraviral interactions (E–E), we have expanded our investigation to uncover how EBV ncRNAs

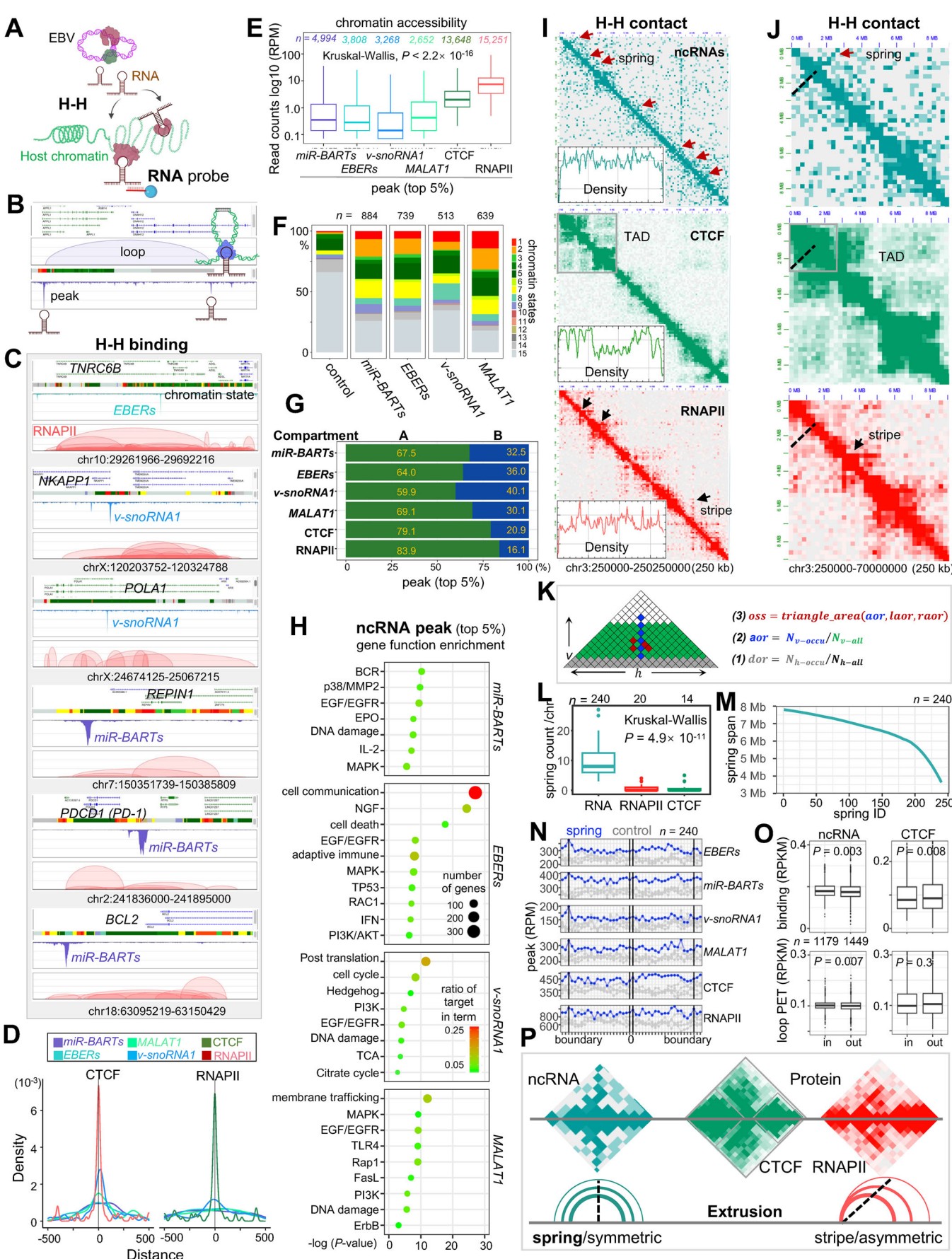

**Figure 5. EBV ncRNAs-associated chromatin DNA–DNA contacts within host genome (H–H).**

(A) Schematic of ncRNA-associated chromatin DNA–DNA interactions within the host genome (H–H). (B) Schematic of H–H contacts, including peaks and loops. A color bar is included to illustrate the 15 chromatin states (see "Methods"). (C) Display the binding peaks in host human GM12878 cells targeted by ncRNAs, with RNAPII-mediated chromatin loops as reference. (D) The relationship between ncRNA-targeted chromatin loci and the binding sites of CTCF and RNAPII. (E) Box plots present the chromatin accessibility targeted by ncRNAs and protein factors. Kruskal–Wallis tests are performed, $P < 2.2 \times 10^{-16}$. (F) The histogram shows the distribution of 15 chromatin states for ncRNA-targeted host chromatin binding peaks. (G) The percentage of chromatin targets in the A/B compartment. (H) GO term enrichment analysis of host genes targeted by ncRNAs. (I) Contact heatmaps present the H–H contacts. Arrows highlight the structure. The inset presents the density distribution of the heatmap image. (J) Zoomed-in view of H–H contacts highlighting the chromatin structure associated with ncRNAs and proteins. The dashed line highlights the spring region. (K) Algorithm for identifying the 'spring' structures associated with ncRNAs (see "Methods"). (L) Box plots illustrating the distribution of spring structures captured by the ncRNAs RDD method, as well as the RNAPII or CTCF ChIA-PET methods, on individual chromosomes. Kruskal–Wallis tests are performed, $P = 4.9 \times 10^{-11}$. (M) Point plot displaying the span (in Mb) of the spring structure. (N) Point-line plot depicting the enrichment of ncRNAs, CTCF, and RNAPII at the boundaries of the spring structures. The blue line represents the spring structures, while the gray line indicates randomly selected genomic regions of equivalent size used as a negative control. (O) Box plot illustrates the enrichment of ncRNAs (binding and loop density of merged biological replicates: *EBERs*, $n = 3$; *miR-BARTs*, $n = 2$; *v-snoRNA*, $n = 2$) within the TADs/CCDs regions contained in the springs, whereas CTCF (density of merged two biological replicates) is evenly distributed across TADs/CCDs both inside (in, $n = 1179$) and outside (out, $n = 1449$) the spring structures. Two-sided Wilcoxon tests are performed, and *P* values are indicated on the plots. (P) Comparison of the chromatin extrusion model potentially mediated by ncRNA factors to that mediated by CTCF/RNAPII protein factors. Each box plot highlights the median (inside line), the 25–75th percentiles (box), and the minima/maxima values within 1.5× the interquartile range (IQR) of the box (whiskers). Source data are available online for this figure.

engage in inter-species chromatin interactions with the host genome (E–H). As illustrated in Fig. 3B, interactions associated with *v-snoRNA1* are widely dispersed across the EBV genome. *EBERs* and *miR-BARTs* also demonstrate significant frequencies of contact with the host genome, particularly concentrated around their transcriptional regions. Notably, we observed two principal regions of interaction between *miR-BARTs* and the host genome: one at its transcriptional site (BART-I), which interacts with 13 loci of the host genome, and another around the OriP region (BART-II), forming contacts with four host genomic loci. Generally, ncRNA-associated E–H interactions tend to be scattered and occur predominantly between individual loci. It is rare to observe a single locus extending contact to multiple other loci, as evidenced by BART-I and BART-II (Fig. EV5). This indicates that the EBV episome primarily interacts with the host chromatin in a one-to-one manner. This phenomenon may reflect the relatively high mobility of each EBV episome, unlike the host chromosomes, which have relatively fixed chromosomal territories, while interacting as a unified entity with the host chromatin.

These RNA and protein factor-associated E–H interactions are scattered across the host genome, with *miR-BARTs* and *EBERs* facilitating a higher proportion of EBV–host genome contacts compared to *MALAT1* (Fig. 3H). These interactions occur across various chromatin states, including both open and closed conformations (Fig. 3I). The interaction patterns of *EBERs* are reminiscent of those mediated by RNAPII, whereas *v-snoRNA1*'s interactions are more similar to those facilitated by CTCF (Fig. 3I). This distribution largely targets gene intron regions (Fig. 3J). For instance, *miR-BARTs* interacts with the *SLC37A1* gene, *v-snoRNA1* associates with *NAV3*, and *EBERs* engage with *SVIL-AS1* (Fig. 3K), with these interactions occurring within RAID regions.

Overall, these ncRNA-associated E–H interactions are functionally enriched in genes related to cell cycle regulation, apoptosis, and immune-related pathways (Fig. 3L).

## EBV ncRNA *EBERs* represses viral gene expression via RNAPII-associated chromatin loops

To delve deeper into the role of ncRNAs in gene regulation within the EBV genome, we employed CRISPR interference (CRISPRi) to inhibit the transcription of the highly abundant *EBERs* ncRNA (Gilbert et al, 2014). As illustrated in Fig. 4A, we modified the

lenti_dCas9-KRAB-MeCP2 construct by replacing the BSD element with GFP and infected GM12878 cells. Subsequently, we employed fluorescence-activated cell sorting (FACS) to isolate GFP-positive cells across two rounds ("Methods"). These GM12878 cells, now expressing dCas9-KRAB-MeCP2, were then infected with the sgRNA-expressing pLG15 lentivirus and subjected to puromycin selection, creating stable cell lines for further analysis. This included transcriptome analysis by RNA-seq and RNAPII-associated chromatin interaction studies using ChIA-PET.

Analysis of gene expression of *EBERs*, based on stranded RNA-seq data from CRISPRi cells that inhibit *EBERs* transcription (*EBERs*-KD) and non-targeting control CRISPRi cells (*EBERs*-NC), revealed a decrease rather than complete suppression of *EBERs* (Fig. 4B). This partial repression could be attributed to the presence of multiple copies of the EBV episome in GM12878 cells (Figs. 1G and EV2D–G). In exploring the regulatory mechanism of *EBERs*, we generated RNAPII ChIA-PET libraries for both *EBERs*-KD and *EBERs*-NC cells.

Overall, CRISPRi targeting of *EBERs* region led to the emergence of new small compartments (Fig. 4C), revealing an increased frequency of chromatin loop formation (Fig. 4D), particularly long-range loop structures in this region (Fig. 4E). RNAPII-mediated chromatin loops were generally enhanced in *EBERs*-KD cells (Fig. 4F). This enhancement was accompanied by the upregulation of essential components in the lytic phase of EBV DNA replication, including a viral oncogene *BARF1* (Hoebe et al, 2013), the BCL-2-like antagonists BALF1/2 (Bellows et al, 2002), which were proven to be involved in functional loops by H3K27ac HiChIP (Ding et al, 2022), and the *BVRF1* gene—a capsid vertex-specific component involved in EBV DNA encapsidation, ensuring accurate DNA genome cleavage and stabilizing capsids (Li et al, 2018a)—that are targeted by *EBERs*, around these newly formed loops (Fig. 4G). This suggests that *EBERs* ncRNA may inhibit RNAPII-mediated chromatin loops, thereby downregulating the expression of target genes.

## EBV ncRNA-related chromatin DNA–DNA interactions between host–host (H–H)

Viral ncRNAs transcribed from EBV diffuse into the host genome, where they target and facilitate chromatin DNA–DNA interactions (H–H). This process is illustrated in Fig. 5A, with sequencing

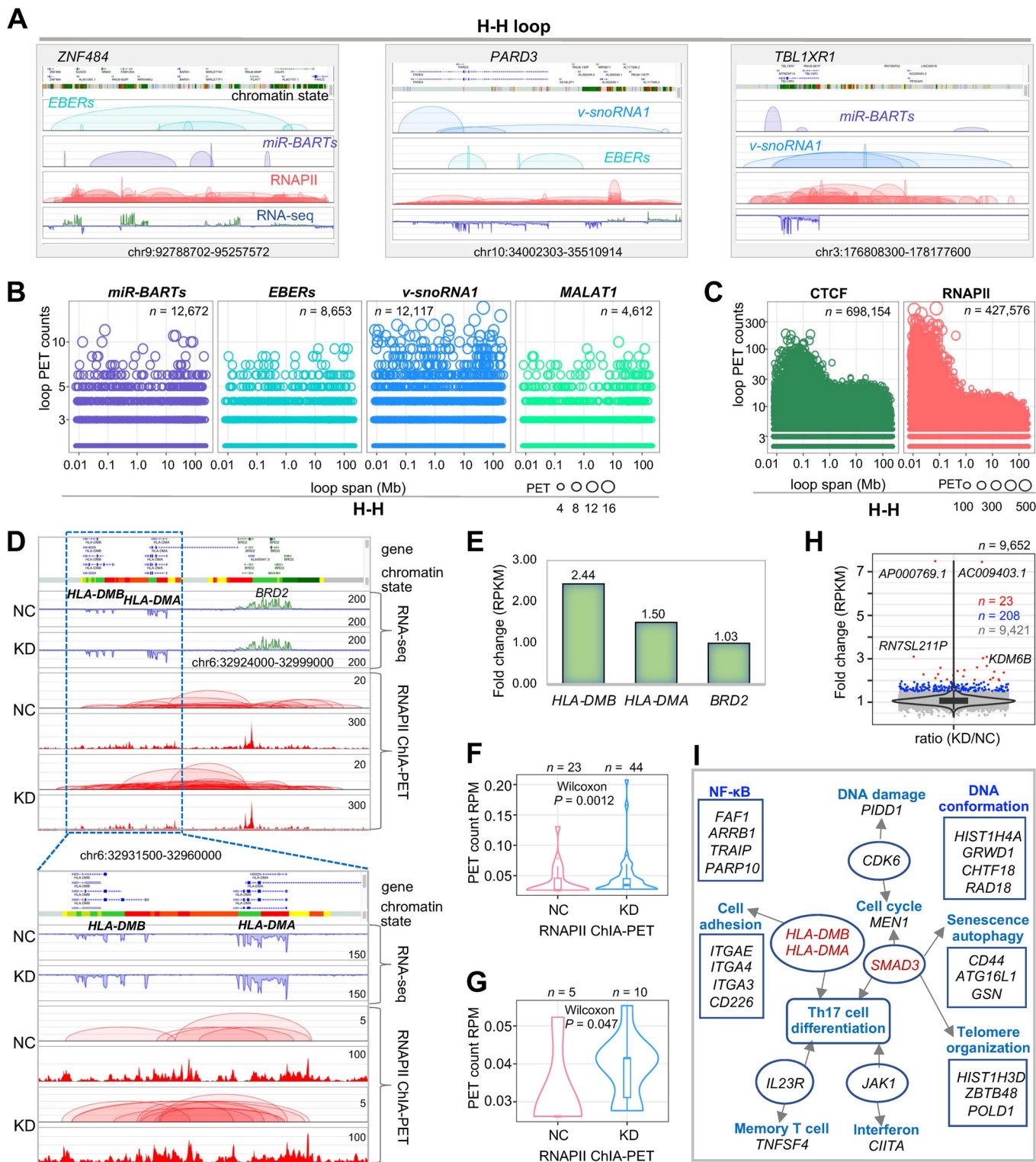

coverage depicted as binding peaks, contact as a heatmap, and high-frequency contact as loops in Figs. 5B and EV4B. The binding peaks reveal a significant enrichment of EBV ncRNAs binding within the host genome, indicative of specific targeting. For example, as illustrated in Fig. 5C, *EBERs* specifically target the

intron of the host *TNRC6B* gene. Similarly, *v-snoRNA1* targets the *NKAPP1* gene, while *miR-BARTs* associates with the genes that are important target for specific drugs such as *PDCD1* (*PD-1*) and *BCL-2* genes (Kuszczak et al, 2023). These ncRNAs seldom overlap with RNAPII or CTCF anchors, instead localizing within related RAIDs.

**Figure 6.   *EBERs* ncRNA repress host gene expression via RNAPII-mediated chromatin loops.**

(**A**) BASIC Browser view displays the chromatin loops at host human GM12878 cells targeted by EBV ncRNAs. A color bar is included to illustrate the 15 chromatin states (see "Methods"). (**B**) Scatter plots display the frequency of chromatin loops associated with ncRNAs between H–H. (**C**) Scatter plots show the frequency of chromatin loop-associated proteins (CTCF and RNAPII) between H–H. (**D**) Display of RNA expression levels and RNAPII-associated chromatin loops in *EBERs*-NC cells and *EBERs*-KD cells. A color bar is included to illustrate the 15 chromatin states (see "Methods"). (**E**) Bar charts display the fold changes from RNA-seq on *HLA-DMB*, *HLA-DMA*, and *BRD2* genes shown in (**D**). (**F, G**) Violin-Box plots present the chromatin interaction PET counts from RNAPII ChIA-PET in *EBERs*-NC cells and *EBERs*-KD cells at the genomic region of (**D**, top) and (**D**, bottom). Two-sided Wilcoxon tests are performed, with P values of $P = 0.0012$ and $P = 0.047$, respectively. (**H**) The Violin-Box-Jitter plot illustrates the fold change from RNA-seq data on human host genes in *EBERs*-KD cells compared to *EBERs*-NC cells. Red dots represent a fold change greater than 2. Blue dots represent a fold change between 1.5 and 2. Gray dots present no changes. (**I**) The key gene network (within the blue square) and enriched pathways (in blue text) are associated with EBV *EBERs* ncRNA. Ellipses indicate genes enriched in multiple pathways, with red indicating important genes. Each box plot highlights the median (inside line), the 25th to 75th percentiles (box), and the minima/maxima values within 1.5× the interquartile range (IQR) of the box (whiskers). Source data are available online for this figure.

Only, *v-snoRNA1* and *EBERs* exhibits a certain correlation with CTCF-binding regions (Figs. 5D and 3I).

To elucidate the genomic preferences of these RNAs, we analyzed the chromatin states and histone modification signatures within the targeted peak regions. The areas targeted by ncRNAs predominantly reside in weakly open chromatin regions, contrasting with the more accessible chromatin regions associated with RNAPII protein anchors (Fig. 5E,F). In comparison, *MALAT1* is found in more active regions, including promoters, enhancers, and transcriptional regions, reflecting its primary location in the speckle region near RNAPII transcriptional hubs (Quinodoz et al, 2021).

These RNA and protein factor targets at the host chromatin are widely distributed, showing a preference for active A compartments (Fig. 5G). The genes targeted are enriched in functions associated with EGF/EGFR, MAPK, PI3K/AKT signaling pathways, immune response, and cell cycle (Richardo et al, 2020) (Fig. 5H).

When comparing genome structures from RDD data presented as contact maps to those obtained by ChIA-PET method, we found high concordance, with similar structural features noted at whole genome. Interactions within the same chromosome are preferentially observed (Fig. EV6A). At higher resolution, ncRNA-associated interactions show more off-diagonal signals and weaker diagonal signals in RDD pairwise contact maps (Figs. 5I and EV6B). Unlike the domain or compartment structures typically mediated by CTCF or RNAPII, such as TADs or RAIDs, we noted a distinct pattern: a well-defined line perpendicular to the diagonal, referred to as the spring structure (Figs. 5J and EV6C). The spring calling algorithm, which identifies these spring structures in contact heatmaps, involves a series of steps aimed at calculating baseline data, determining spring bandwidth along the antidiagonal, selecting candidate springs based on specific criteria, and assessing a prominence score to confirm the presence of a candidate spring ("Methods", Fig. 5K). A total of 240 spring structures were identified, spanning across 1179 CCDs, which represent 44.8% of the total CCDs. On average, there are about eight spring structures per chromosome, with their lengths ranging from 3 to 8 Mb and including between 5 and 7 CCDs each (Figs. 5L,M and EV6D,E).

The spring structures often extend across the boundaries of CCDs, indicating a possible organizational function that might involve embedding within several adjacent CCDs (Figs. 5I,J and EV6F). To analyze their relationships, we divided each spring into 40 bins and compared these to control regions of the same size that were randomly selected from the genome. We examined the binding and interaction signals of ncRNAs and CTCF, discovering that the boundaries of springs are enriched with both ncRNAs and CTCF signals, while

differences exist in the internal signal distribution (Fig. 5N). Furthermore, by analyzing the enrichment of ncRNAs and CTCF within CCDs located inside or outside the spring region, we found that ncRNAs binding and interaction signals are enriched within the CCDs inside the spring, whereas CTCF signals are evenly distributed or more enriched within CCDs outside the spring region (Fig. 5O). This distinctive pattern potentially indicates a symmetric loop organization facilitated by ncRNAs (Fig. 5P), differing from the asymmetric DNA loop extrusion commonly driven by RNAPII or CTCF (Banigan et al, 2020; Zheng et al, 2019). This organizational feature suggests the potential role of ncRNAs in the arrangement and structural assembly of CCD boundaries or several adjacent CCDs.

Exploring the high-frequency higher-order structures identified by RDD, we cataloged interactions occurring two or more times (PET ≥2) as chromatin loops (Fig. 5B). Consistent with pairwise contact findings, ncRNA-associated loops appear more scattered and less frequent, with a larger span compared to protein-associated loops (Fig. 6A–C), and tend to exhibit a symmetric pattern rather than the concentrated asymmetric pattern seen in RNAPII- and CTCF-associated structures (Fig. EV7A). These targeted regions also enrich for functional genes in pathways like EGF, PI3K-AKT, NF-κB, DNA damage, CD40, and others (Fig. EV7B). Their association with chromatin signatures revealed more enrichment in inactive B compartment regions (Fig. EV7C,D) compared to their binding peak enrichment (Fig. 5F,G).

## EBV ncRNA *EBERs* repress host gene expression via RNAPII-associated chromatin loops

In examining the chromatin conformation captured via RNAPII ChIA-PET and gene expression via RNA-seq in *EBERs*-KD and *EBERs*-NC cells (Fig. EV8A–C), no obvious changes were observed in the A/B compartment (Fig. EV8D). Notably, the reduction of *EBERs* led to increased expression of HLA class II histocompatibility-associated genes *HLA-DMA* and *HLA-DMB* (Ghasemi et al, 2020), accompanied by an increase in corresponding RNAPII-mediated chromatin loops (Fig. 6D–G). Similarly, the *MCUB* gene exhibited enhanced expression post *EBERs* reduction, with increased RNAPII chromatin loop contacts around its promoter region (Fig. EV8E,F). Conversely, genes like *MALAT1* and *NEAT1* ncRNAs, which remained unchanged, displayed consistent loop contacts before and after *EBERs* CRISPRi (Fig. EV8G–I). A total of 231 host genes were upregulated concurrent with the reduction of *EBERs* (Fig. 6H), suggesting that the primary function of *EBERs* is to suppress gene expression. The

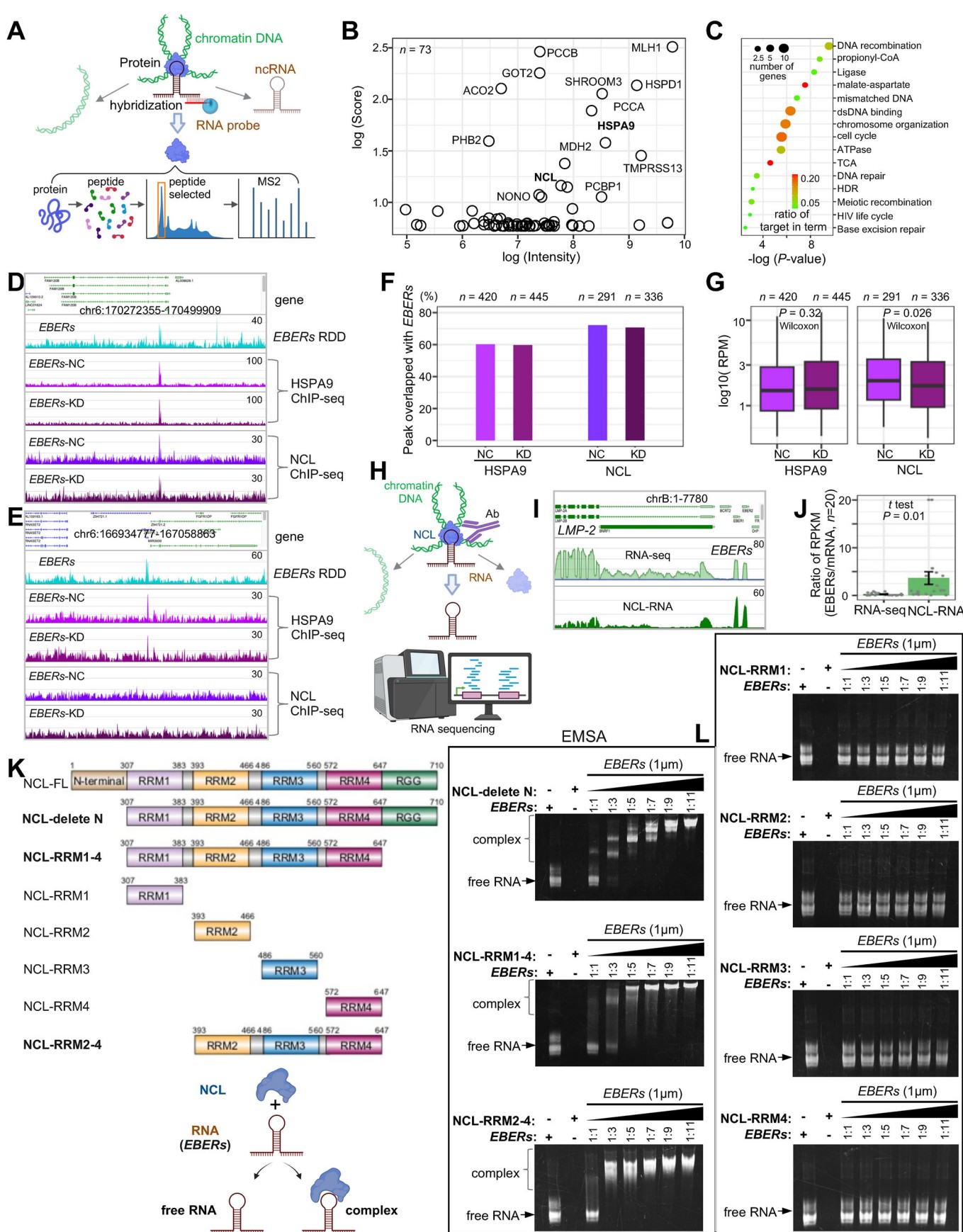

**Figure 7.　NCL interaction with ncRNA *EBERs*.**

(**A**) Schematic of capturing *EBERs* ncRNA cofactors via annotated tandem (MS2) mass spectrum. (**B**) The scatter plot displays the cofactors associated with *EBERs* ncRNA captured by MASS. (**C**) GO term enrichment analysis of *EBERs* cofactors. The *x*-axis represents -log10 (P-value) from the hypergeometric test. (**D**) Display of the binding peak of EBV cofactors HSPA9 and NCL in *EBERs*-NC cells and *EBERs*-KD cells, overlapping with EBERs-binding peaks. (**E**) Display of the binding peak of *EBERs*, without detecting peaks of cofactors NCL in *EBERs*-KD cells and *EBERs*-NC cells. (**F**) Bar charts display the proportion of cofactors HSPA9 and NCL that overlapped with *EBERs* targeting regions. *n* indicates the total number of peaks for HSPA9 and NCL captured by ChIP-seq. (**G**) Box plots illustrating the change of binding signal of HSPA9 and NCL before and after *EBERs* knockdown. Two-sided Wilcoxon tests are performed. Each box plot highlights the median (inside line), the 25–75th percentiles (box), and the minima/maxima values within 1.5× the interquartile range (IQR) of the box (whiskers). (**H**) Schematic representation of NCL-interacted chromatin RNA assay (termed NCL-RNA). (**I**) Basic Browser View of *EBERs* ncRNA and mRNA, such as LMP-2, as observed in RNA-seq and NCL-RNA. (**J**) Enrichment analysis of *EBERs* ncRNA is depicted by the ratio of RPKM of *EBERs* compared to other mRNAs, with *t* tests performed. Bars represent the mean values of the data, while error bars indicate the standard deviation (SD). (**K**) Schematic illustration of NCL truncations and the principle of EMSA. (**L**) EMSA results demonstrate the interaction at varying concentrations of NCL protein mixed with 1 μM *EBERs*. Source data are available online for this figure.

host genes suppressed by *EBERs* are associated with DNA conformation, NF-κB signaling, immune responses including MHC-II, T cell differentiation and memory, cell cycle regulation, and DNA damage pathways (Fig. 6I). The repression mechanism involves the modulation of RNAPII-mediated chromatin loops (Fig. EV9). The significant increase in interaction between *HLA-DMA* and *HLA-DMB* after *EBERs* reduction suggests it plays a crucial role in the class II antigen presentation pathway (Fling et al, 1994).

## Nucleolin interaction with ncRNA *EBERs* in host gene expression repression

These ncRNAs lack specific chromatin binding motifs, suggesting their role in facilitating chromatin interactions through the recruitment of divergent protein factors. To further explore the cofactors of EBV ncRNAs, we employed biotin-modified antisense *EBERs* to hybridize with chromatin complexes, thereby enriching *EBERs*-associated complexes (Fig. 7A). Subsequent peptide analysis via mass spectrometry revealed that *EBERs*-binding protein factors are enriched in functions related to DNA recombination, chromosome organization, the cell cycle, DNA repair, and metabolism, specifically the TCA cycle (Fig. 7B,C). Notably, some of these functions correspond with the targeted gene functions of EBV ncRNAs, aligning with previous observations.

Subsequent ChIP-seq was conducted focusing on two cofactors associated with the virus: heat shock protein family A member 9 (HSPA9) and nucleolin (NCL) (Fig. EV10A,B). HSPA9 plays a crucial role in the viral life cycle, including viral entry, translation during infection, and the assembly of viral particles (Su et al, 2020). In contrast, NCL is directly involved in mediating the immune evasion strategies of the Epstein–Barr virus (Lista et al, 2017). These experiments were performed in *EBERs*-KD and *EBERs*-NC cells, revealing several hundred binding regions with more than 60% overlap with EBERs-binding sites (Fig. 7D–F). The binding signals for HSPA9 remained consistent even after the knockdown of *EBERs*, whereas a more obvious reduction in NCL binding was noted (Fig. 7G). This suggests that *EBERs* interact with HSPA9 that is already bound to chromatin, which may subsequently facilitate the recruitment of NCL.

To confirm the interaction between NCL protein and *EBERs* ncRNA, we conducted an NCL-interacted chromatin RNA assay (termed NCL-RNA) using a modified CLIP-seq approach (Stork and Zheng, 2016). Instead of using UV crosslinking, we prepared

EGS- and formaldehyde-crosslinked chromatin complexes, similar to the methods used in previous ChIA-PET and RDD experiments (Fig. 1A, "Methods"). Chromatin complexes were enriched using an NCL antibody, and the associated RNA was sequenced (Figs. 7H and EV10C). Comparison with RNA-seq data indicated that ncRNA *EBERs* signals were higher than those of the EBV mRNA (Fig. 7I,J), supporting NCL's in vivo association with *EBERs*. We also found that NCL associates with many host RNAs (Fig. EV10D,E), mainly rRNA, snRNA, and snoRNA, which is consistent with previously reported functions of NCL (Mongelard and Bouvet, 2007). However, we did not obtain high-resolution motif information. While we speculate that NCL may bind preferentially to RNA stem-loop structures, it is also possible that the crosslinking method employed in our study captured a broader and less specific range of associated RNAs. The specific mechanisms underlying NCL's RNA interactions require further investigation in future studies.

To further elucidate which domains of the NCL protein interact with *EBERs* ncRNA, we engineered various truncated constructs of NCL in the PET28a vector (Fig. 7K). These constructs were expressed in *E. coli*, and the proteins were purified. Electrophoretic mobility shift assays (EMSA) were then employed using in vitro transcribed *EBERs* ncRNA (with *EBER1* as the representative) and purified NCL proteins to assess their interaction potential (Fig. 7K,L). In these assays, free RNA displayed the highest electrophoretic mobility, while stronger binding interactions led to reduced mobility. The EMSA results (Fig. 7L) showed that *EBERs* ncRNA naturally exhibits two conformational states, which may correspond to the stem-loop structure of the hairpin and the linear structure, as the assay was performed using a native PAGE gel rather than a denaturing gel ("Methods"). In binding assays with NCL-deleted N (N-terminal deletion), the binding strength increased with higher protein concentrations. Further deletion of the C-terminal of NCL (in NCL-RRM1-4) further enhanced *EBERs* ncRNA binding compared to NCL-deleted N, suggesting that the RGG domain influences binding interactions. Experiments with individual RRM domains (RRM1, RRM2, RRM3, RRM4) did not detect binding under the tested conditions to *EBERs* ncRNA. However, binding signals were observed with the combined RRM2-4 domains, indicating that these domains collaborate to mediate interactions with *EBERs* ncRNA. This is generally aligned with previous studies on NCL's interaction with RNAs, such as telomeric repeat-containing RNA (*TERRA*) (Khan et al, 2023). These results suggest a role for cooperative RRM interactions in facilitating NCL's binding to RNA targets.

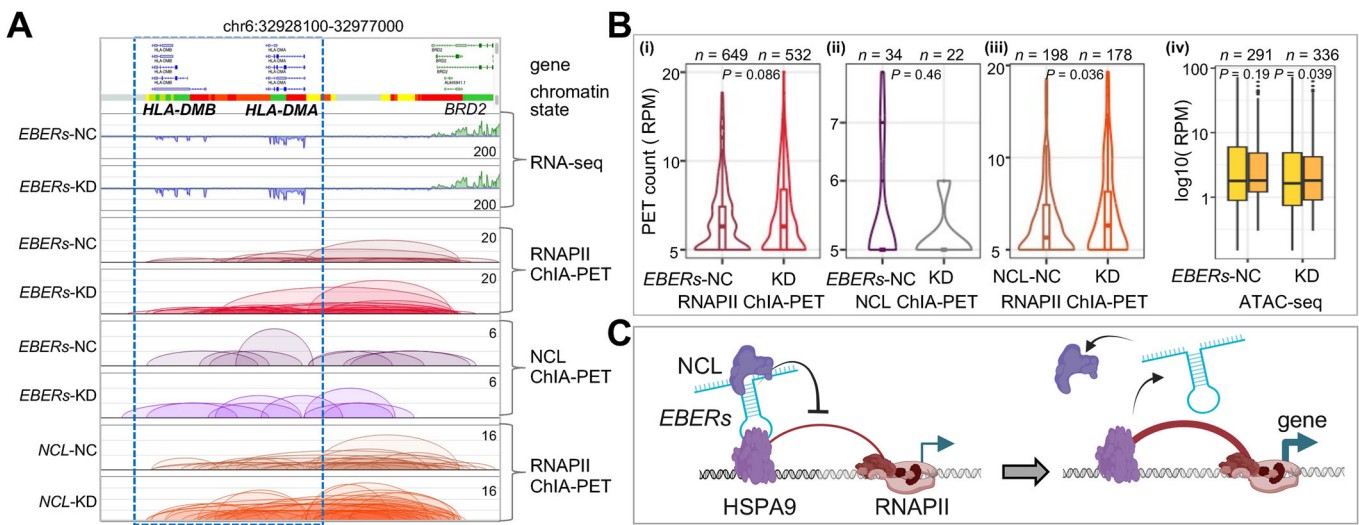

◀ **Figure 8.   ncRNA *EBERs* with NCL to repress gene expression.**

(A) Basic Browser View illustrating changes in chromatin interactions upon the downregulation of *EBERs* or NCL. (B) Violin box plot displaying alterations in chromatin interactions: (i) RNAPII-associated chromatin in *EBERs* target regions between *EBERs*-NC and *EBERs*-KD groups, (ii) NCL-associated chromatin in *EBERs* target regions between *EBERs*-NC and *EBERs*-KD groups, (iii) RNAPII-associated chromatin between NCL-NC and NCL-KD groups. The number of chromatin loops overlapping these target regions is denoted by '*n*'. And (iv) changes in chromatin accessibility of *EBERs*-NC (yellow) compared to *EBERs*-KD (orange) at NCL binding regions, obtained from *EBERs*-NC ($n = 291$) and *EBERs*-KD ($n = 336$). Statistical significance was assessed using the Wilcoxon test, with *P* values displayed above the plots. Each box plot highlights the median (inside line), the 25th to 75th percentiles (box), and the minima/maxima values within 1.5× the interquartile range (IQR) of the box (whiskers). (C) The mechanistic model is presented: The left panel illustrates the interactions between *EBERs*, HSPA9, and NCL, which inhibit the RNAPII-associated chromatin loop to repress gene expression. The right panel shows an increase in gene expression following the reduction of *EBERs* and subsequently NCL, thereby alleviating the inhibition of the RNAPII-associated chromatin loop. (D) The model of the interactome between EBV and the host human GM12878 cells. On the left, the EBV 3D structure is depicted; in the middle, the interaction between EBV and the host is shown, along with the enriched key functions and pathways; on the right, the dynamic chromatin extrusion model of ncRNAs and proteins is illustrated, highlighting their targeting to chromatin with low accessibility. Source data are available online for this figure.

To investigate whether the interaction between NCL protein and *EBERs* ncRNA affects gene expression, we performed NCL knockdown using siRNA from Santa Cruz (sc-29230, siNCL), which comprises a pool of four target-specific siRNAs, each 19–25 nucleotides in length (Fig. EV10F). By comparing the downregulation of NCL (NCL-KD) using siNCL to the control (NCL-NC) using scrambled siRNA (siNC) (Fig. EV10F), we observed a tendency toward an increase in RNAPII-associated chromatin loops in the target regions of *EBERs*' genes (Fig. 8A,B). This response appeared to mirror the effects of *EBERs* knockdown (*EBERs*-KD), where RNAPII-associated chromatin loops similarly increased (Fig. 6D–I). In addition, *EBERs* knockdown tended to reduce chromatin interactions associated with NCL and led to more open chromatin states in the regions of downregulated genes (Figs. 8B and EV10G). These findings suggest *EBERs* repress target gene expression partially through interaction with NCL, inhibiting chromatin opening, restraining the formation of RNAPII-associated chromatin loops necessary for gene activation (Fig. 8C).

In summary, our findings provide a detailed view of the structural and functional dynamics mediated by ncRNAs between the virus and host genomes. EBV co-opts host RNAPII, CTCF, and other histone factors to construct its three-dimensional structure, thereby facilitating the transcription of both coding and noncoding genes. EBV ncRNAs, along with cofactors, such as NCL, target to the weakly accessible chromatin in the host to inhibit RNAPII-mediated chromatin loop, which could contribute to suppressing the expression of genes crucial for chromatin structure organization, immune response, cell cycle regulation, and DNA repair mechanisms (Fig. 8D).

## Discussion

In this study, we integrated 3D high-throughput sequencing techniques and super-resolution microscopy to elucidate the 3D structure of EBV episome within host lymphocytes. Our findings reveal that the exogenous double-stranded DNA of EBV exists in an open chromatin state within host cells. Chromatin state analysis indicates that the EBV episome lacks distinct heterochromatin regions typically observed in host chromatin, consistent with previous studies showing low H3K27me3 binding signals (Kim and Lieberman, 2024). In addition, certain regions of the episome exhibit higher activity and are enriched with host protein factors such as histones, CTCF, and RNAPII. Previous analyses of RNAPII-mediated promoter-enhancer interaction networks in the

host genome have shown associations with regions bound by EBNAs or NF-κB, collectively defined as the "EBV regulome." (Jiang et al, 2017) However, these findings were based on indirect association analyses. To directly identify EBV–host interaction sites, subsequent studies employed circular chromatin conformation capture followed by deep sequencing (4C-seq) and high-resolution chromatin conformation capture (Hi-C) assays to capture EBV–host chromatin interactions. These studies revealed that EBV predominantly interacts with intronic and intergenic regions of the host genome, ~10% of which are enriched with active histone marks (Wang et al, 2020). This observation aligns with the proportion of RNAPII-mediated EBV–host enhancer–promoter interactions identified in this study. Further chromatin state analysis of these host loci revealed that these interactions primarily occur in regions of inactive chromatin, indicating that the regions enriched with active histone marks are actually in a non-active state. This provides additional detail and refinement to our understanding of EBV–host chromatin interactions.

In addition, by integrating CTCF and RNAPII ChIA-PET data, we defined the 3D structure of the EBV episome, demonstrating that it hijacks host protein factors, CTCF and RNAPII, to organize three CCDs and two RAIDs. By building upon previous studies involving 4C, Hi-C, HiChIP, and CTCF ChIP-seq (Mizokami et al, 2024; Morgan et al, 2022; D et al, 2023; Arvey et al, 2012), our findings provide more precise and direct insights into the CCD chromatin structure and RAID gene regulatory structures of EBV, vividly captured through super-resolution microscopy.

These domains facilitate the transcription of both coding and noncoding RNAs. Remarkably, these viral structures resemble those of the host, displaying similar characteristics such as the convergent motif of CTCF at the boundaries of CCDs. The anchors of these CCDs and RAIDs serve as structural foci that provide a directional framework for chromatin topology, enabling coordinated transcriptional regulation. This pattern mirrors the gene transcription structures established by CTCF and RNAPII in the human host, enhancing the efficiency of gene transcription and regulation (Tang et al, 2015). Notably, we identified three hotspots of EBV–host chromatin interactions (HS1, HS2, and HS3) within the structural foci, which regulate target gene expression through these interactions.

Our investigation sheds light on the intricate interplay between EBV-encoded ncRNAs and host chromatin architecture. We elucidate the mechanisms by which viral ncRNAs, including *EBERs*, *v-snoRNA1*, and *miR-BARTs*, disrupt host gene expression. Our findings indicate that EBV ncRNAs preferentially target

genomic regions characterized by less accessible chromatin, significantly impacting genes involved in critical signaling pathways such as NF-κB and PI3K-AKT. These pathways are crucial for viral infection, immune responses, cell cycle regulation, and DNA repair processes, highlighting EBV's strategic manipulation of host cellular machinery to ensure its own survival and proliferation. This behavior contrasts with previous findings where endogenous RNAs are typically enriched in open chromatin regions, particularly at enhancer–promoter regions, whereas EBV ncRNAs localize mainly in weakly accessible chromatin regions (Zhou et al, 2019).

While prior research has emphasized the regulatory roles of individual viral ncRNAs, our study advances the field by providing a detailed chromatin interactome map associated with EBV ncRNAs. Utilizing both RDD and ChIA-PET techniques, we visualize the three-dimensional chromatin structure and reveal the extensive network of interactions between EBV and the host genome. These interactions involve both protein factors and ncRNAs, offering a holistic view of the EBV's sophisticated strategies to commandeer host transcriptional machinery and providing new perspectives on viral-host interactions.

By applying CRISPR interference (CRISPRi) to knock down the EBV ncRNA *EBERs*, we discovered that *EBERs* engage with numerous cofactors, including HSPA9, also known as GRP75 or mortalin, which bind to host chromatin. This recruitment of additional cofactors such as NCL, results in the repression of gene expression. They achieve this by inhibiting RNAPII-associated chromatin loops, further illustrating the complex regulatory mechanisms at play. Furthermore, we discovered that the RNA recognition motif (RRM) of the NCL protein does not bind to *EBERs* in isolation; instead, multiple domains must collaborate to bind *EBERs*, supporting the presence of four RRMs (RRM1-4) in NCL rather than just one. This cooperative binding mechanism aligns with NCL's interactions with other RNAs, such as *TERRA* (Khan et al, 2023), where multiple domains work in concert to achieve high-affinity binding. The collaborative nature of NCL's RRM domains significantly enhances binding affinity to RNA targets. While individual RRM domains display limited binding capability, effective recognition of *EBERs* requires the coordinated action of multiple domains. This multi-domain requirement underscores the intricate nature of NCL-RNA interactions and indicates NCL's versatile role in RNA metabolism, encompassing its interactions with both viral and host RNAs (Mongelard and Bouvet, 2007). However, analysis of our chromatin-associated NCL-interacted RNA data did not identify specific RNA-binding motifs for NCL or significantly enriched binding peaks. While *EBERs* RNA was enriched, certain low-abundance host RNAs were also captured. This could potentially be due to the use of a dual-crosslinking method, which may have captured RNAs indirectly associated with NCL. Additional experiments with an optimized crosslinking approach could provide further insights into this observation.

Understanding this cooperative mechanism offers valuable insights into potential therapeutic strategies aimed at disrupting pathogenic NCL-RNA complexes involved in EBV-related diseases. The elucidation of the mechanisms by which EBV ncRNAs affect host gene expression and chromatin architecture might open new avenues for therapeutic intervention. Targeting specific ncRNAs or the chromatin interactions they mediate might disrupt the viral life cycle or mitigate the effects of EBV-associated diseases. Additionally, our findings contribute to the broader understanding of ncRNA biology and the role of chromatin organization in gene regulation, with implications that extend beyond viral pathogenesis.

# Methods

**Reagents and tools table**

| Reagent/resource | Reference or source | Identifier or catalog number |
|---|---|---|
| **Experimental models** | | |
| Human GM12878 cell line | Coriell Cell Repositories | N/A |
| Marmoset B95-8 cell line | Pricella | Cat# CL-0476 |
| Human Ramos cell line | ATCC | Cat# CRL-1596 |
| HEK293T cell line | ATCC | Cat# CRL-3216 |
| **Recombinant DNA** | | |
| pCMV-mCherry-NCL(human)-3×HA-Neo plasmid | Miaolingbio | Cat# P38028 |
| PET28a vector | Addgene | Cat# 69864-3 |
| PET28a-NCL | This study | N/A |
| PET28a-NCL-ΔN (delete N) | This study | N/A |
| PET28a-NCL-RRM1-4 | This study | N/A |
| PET28a-NCL-RRM1 | This study | N/A |
| PET28a-NCL-RRM2 | This study | N/A |
| PET28a-NCL-RRM3 | This study | N/A |
| PET28a-NCL-RRM4 | This study | N/A |
| PET28a-NCL-RRM2-4 | This study | N/A |
| lenti_dCas9-KRAB-MeCP2 plasmid | Addgene | Cat# 122205 |
| pRS001 | This study | N/A |
| pLG15 | Addgene | Cat# 157893 |
| psPAX2 | Addgene | Cat# 12260 |
| pMD2-G | Addgene | Cat# 12259 |
| **Antibodies** | | |
| Anti-RNA Polymerase II RPB1 Antibody (mouse) | Biolegend | Cat# 664906; RRID: AB_2565554 |
| Anti-nucleolin (NCL) antibody | Abcam | ab22758 |
| Anti-HSPA9 antibody | Abclonal | A0558 |
| **Oligonucleotides and other sequence-based reagents** | | |
| RDD probes, linker, primer | This study | Table EV2 |
| **Chemicals, enzymes, and other reagents** | | |
| Formaldehyde solution | Sigma | Cat# 47608 |
| EGS (ethylene glycol bis (succinimidyl succinate)) | Thermo Fisher | Cat# 21565 |
| Glycine | Sigma | Cat# 50046 |
| SDS, 10% Solution, RNase-free | Thermo Fisher | Cat# AM9822 |
| cOmplete™, Mini, EDTA-free Protease Inhibitor Cocktail | Roche | Cat# C11836170001 |

| Reagent/resource | Reference or source | Identifier or catalog number |
| --- | --- | --- |
| Triton X-100 | Acros Organics | Cat# 327371000 |
| DPBS, no calcium, no magnesium | Thermo Fisher | Cat# 14190250 |
| Nuclease-Free Water | Thermo Fisher | Cat# AM9937 |
| dATP (10 mM) | Thermo Fisher | Cat# 18252015 |
| Bovine Serum Albumin (BSA) | TaKaRa | Cat# 2320 |
| Klenow Fragment (3′→5′ exo-) | NEB | Cat# M0212L |
| T4 DNA ligase | NEB | Cat# M0202S |
| Recombinant Proteinase K Solution (20 mg/mL) | Thermo Fisher | Cat# AM2548 |
| TE, pH 8.0, RNase-free | Thermo Fisher | Cat# AM9849 |
| Buffer EB | Qiagen | Cat# 19086 |
| I-Block™ Protein-Based Blocking Reagent | Thermo Fisher | Cat# T2015 |
| Ethyl alcohol, Pure | Sigma | Cat# E7023 |
| Tris (1 M), pH 7.0, RNase-free | Thermo Fisher | Cat# AM9851 |
| Tris (1 M), pH 8.0, RNase-free | Thermo Fisher | Cat# AM9856 |
| EDTA (0.5 M), pH 8.0, RNase-free | Thermo Fisher | Cat# AM9260G |
| NaCl (5 M), RNase-free | Thermo Fisher | Cat# AM9759 |
| SSC (20X), RNase-free | Thermo Fisher | Cat# AM9770 |
| Dimethyl sulfoxide | Sigma | Cat# D2650 |
| TRIzol™ Reagent | Thermo Fisher | Cat# 15596026 |
| Fetal Bovine Serum | Excell Bio | Cat# FSP500 |
| Dynabeads™ M-280 Streptavidin | Thermo Fisher | Cat# 11205D |
| Sodium Acetate (3 M), pH 5.5, RNase-free | Thermo Fisher | Cat# AM9740 |
| Formamide (Deionized) | Thermo Fisher | Cat# AM9342 |
| Dynabeads™ MyOne™ Streptavidin C1 | Thermo Fisher | Cat# 65001 |
| Deoxynucleotide (dNTP) Solution Mix | NEB | Cat# N0447S |
| GlycoBlue™ Coprecipitant (15 mg/mL) | Thermo Fisher | Cat# AM9516 |
| EZ-Link™ Iodoacetyl-PEG2-Biotin | Thermo Fisher | Cat# 21334 |
| Isopropyl alcohol | Sigma | Cat# I9030 |
| DNA Extraction Reagent | Solarbio | Cat# P1012-100ml |
| SUPERase•In™ RNase Inhibitor (20 U/μL) | Thermo Fisher | Cat# AM2696 |
| T4 DNA Polymerase | NEB | Cat# M0203S |
| Dynabeads™ Protein G for Immunoprecipitation | Invitrogen | Cat# 10003D |
| 2-Mercaptoethanol | Sigma | Cat# M6250 |
| Opti-MEM | Gibco | Cat# 31985070 |
| NCL siRNA | Santa Cruz | Cat# sc-29230 |
| Tn5 Transposase | Vazyme | Cat# TD502 |
| Glucose | Sigma-Aldrich | Cat# G8270 |
| glucose oxidase | Sigma-Aldrich | Cat# G7141 |
| catalase | Sigma-Aldrich | Cat# C100 |

| Reagent/resource | Reference or source | Identifier or catalog number |
| --- | --- | --- |
| MEA | Sigma-Aldrich | Cat# 30070 |
| Schneider's Drosophila Medium | Thermo Fisher | Cat# 21720024 |
| DNA Clean & Concentrator-5 | Zymo Research | Cat# D4013 |
| Qubit™ dsDNA Quantification Assay Kits | Thermo Fisher | Cat# Q32851 |
| TruePrep DNA Library Prep Kit V2 for Illumina | Vazyme | Cat# TD501-01 |
| TruePrep Index Kit V2 for Illumina | Vazyme | Cat# TD202 |
| AMPure XP Reagent | Beckman Coulter | Cat# A63881 |
| High Sensitivity Cartridge (N1) | BiOptic | Cat# C105105 |
| Epicentre Ribo-zeroTM rRNA Removal Kit | Epicentre | Cat# MRZH116 |
| NEBNext® UltraTM Directional RNA Library Prep Kit for Illumina® | NEB | Cat# E7420L |
| Poly-ʟ-Lysine | Sigma | Cat# P4707 |
| ascorbic acid | Sigma | Cat# PHR1008 |
| Sytox Orange | Thermo Fisher | Cat# S11368 |
| Fisherbrand Cover Glasses | Fisher Scientific | Cat# 12-545-81 |
| High NA objective lens (NA 1.5) | Olympus | UPLAPO 100XOHR |
| 405 nm laser | TOPTICA Photonics | iBEAM-SMART-405-S |
| 488 nm laser | TOPTICA Photonics | iBEAM-SMART-488-S-HP |
| 561 nm laser | CNI | MGL-FN-561nm |
| 640 nm laser | TOPTICA Photonics | iBEAM-SMART-640-S-HP |
| Fiber coupler | Thorlabs | PAF2-A4A |
| Single-mode fiber | Thorlabs | P3-405BPM-FC-2 |
| **Software** | | |
| GNU Bash (v4.3.48) | https://www.gnu.org/software/bash | |
| Python (v3.7.7) | https://www.python.org | |
| Ubuntu (v16.04) | https://ubuntu.com | |
| R (v4.11) | https://cran.r-project.org | |
| R packages include: ggplot2 (v3.4.0), ggforce (v0.4.1), gridExtra (v2.3), scales (v1.2.1), dplyr (v1.0.10), ggpubr (v0.5.0), rgl (v0.106.8), Shiny (v1.6.0), circlize (v0.4.13), mclust (v6.0.0). | CRAN https://cran.r-project.org | |
| RStudio (v2021.09.0 Build 351) | https://posit.co | |
| BWA (v0.7.17) | Li and Durbin, 2009 https://bio-bwa.sourceforge.net | |
| STAR (v2.7.9a) | Dobin et al, 2013 https://github.com/alexdobin/STAR | |
| RSEM (v1.3.1) | Li and Dewey, 2011 https://github.com/deweylab/RSEM | |
| UCSC Genome Browser | Kent et al, 2002 https://genome.ucsc.edu | |

| Reagent/resource | Reference or source | Identifier or catalog number |
|---|---|---|
| ChIA-PIPE | Lee et al, 2020 https://github.com/TheJacksonLaboratory/ChIA-PIPE | |
| BASIC Browser | Lee et al, 2020 https://github.com/TheJacksonLaboratory/basic-browser | |
| Juicer (v1.19.01) | Durand et al, 2016b https://github.com/aidenlab/juicer | |
| Juicebox (v1.11.08) | Durand et al, 2016a https://github.com/aidenlab/Juicebox | |
| SAMtools (v0.1.19) | Li et al, 2009 http://samtools.sourceforge.net | |
| Deeptools (v3.5.4) | Ramírez et al, 2016 https://deeptools.readthedocs.io/en/develop | |
| BEDTools (v2.29.2) | Quinlan and Hall, 2010 https://bedtools.readthedocs.io/en/latest | |
| MACS2 (v2.2.9.1) | Zhang et al, 2008 https://pypi.org/project/MACS2 | |
| HOMER (v4.10) | Heinz et al, 2010 http://homer.ucsd.edu/homer | |
| HSA | Zou et al, 2016 https://people.umass.edu/ouyanglab/hsa | |
| shinyCircos | Yu et al, 2018 https://github.com/YaoLab-Bioinfo/shinyCircos | |
| Cytoscape (v3.10.1) | Shannon et al, 2003 https://cytoscape.org | |
| HicRep (v1.12.2) | Yang et al, 2017 https://github.com/TaoYang-dev/hicrep | |
| MaxQuant (v.1.6.2.10) | Cox and Mann, 2008 https://maxquant.org | |
| **Other** | | |
| Hg38 reference genome | UCSC Genome Browser https://hgdownload.soe.ucsc.edu/goldenPath/hg38/bigZips/hg38.fa.gz | |
| Human gammaherpesvirus 4 complete genome (EBV-B95-8) | GeneBank https://www.ncbi.nlm.nih.gov/nuccore/NC_007605.1 | |
| Chromatin states of hg38 | NIH Roadmap Epigenomics Project https://egg2.wustl.edu/roadmap/data/byFileType/chromhmmSegmentations/ChmmModels/coreMarks/jointModel/final/E116_15_coreMarks_hg38lift_dense.bed.gz | |
| Description of chromatin states of hg38 | Ting Wang's Lab (https://wang.wustl.edu) https://egg2.wustl.edu/roadmap/web_portal/chr_state_learning.html | |

## Cell culture

Lymphoblastoid cell line GM12878 was obtained from the Coriell Cell Repositories, and the Ramos cell line was obtained from ATCC. GM12878 cell line was established by Epstein–Barr Virus

(EBV) transformation of peripheral blood mononuclear cells using phytohemagglutinin as a mitogen. The Ramos cell line is EBV-negative. Both cell lines were cultured in RPMI 1640 (Gibco, A1049101), supplemented with 15% fetal bovine serum (Gibco, 10082147) at 37 °C in a 5% $CO_2$ atmosphere.

B95-8 is a marmoset B lymphoblastoid cell line that is widely used in research for its ability to continuously produce Epstein–Barr virus (EBV). To culture B95-8 cells, start by mixing 89% Gibco™ RPMI 1640 Medium (ATCC modification) with 10% FBS to create the growth medium. B95-8 cells were cultured in a T75 flask with 10–30 mL of growth medium, keeping the flask upright, and incubate at 37 °C in a 5% $CO_2$ atmosphere.

HEK293T cells (ATCC), derived from human embryonic kidney cells, were cultured at 37 °C in a 5% CO2 atmosphere using 90% Gibco™ DMEM (Gibco, C11965500BT) supplemented with 10% fetal bovine serum (FBS, ExCell Bio, FSP500).

## RNA-associated chromatin DNA–DNA interactions (RDD)

The construction of the RDD library primarily involves several key steps (Tian et al, 2024): crosslinking, lysing of cells and nuclei, tagmentation of chromatin, hybridization using biotinylated probes targeting specific ncRNAs of interest, immobilization of chromatin, blocking of beads with denatured IPB, proximity ligation, elution of the ligated DNA, and preparation of the sequencing library (Fig. EV1A).

### Crosslinking

Cells underwent a dual-crosslinking process using formaldehyde (Sigma-Aldrich, 47608-250ML-F) and ethylene glycol bis (succinimidyl succinate) (EGS, Thermo Fisher Scientific, 21565) to achieve stable crosslinked nuclei, essential for capturing high-affinity chromatin in subsequent hash reactions. In detail, cells in their exponential growth phase were transferred into a 50 mL conical tube and treated with 1% formaldehyde at 37 °C for 20 min with rotation. The reaction was then quenched with 0.125 M Glycine (Sigma-Aldrich, 50046) for 5 min, followed by two washes with DPBS. The formaldehyde-crosslinked cell pellets could either be stored at −80 °C or immediately proceed to EGS crosslinking. For EGS crosslinking, cells were resuspended in 40 mL of DPBS containing 1.5 mM EGS and gently rotated for 40 min at room temperature. The reaction was quenched by adding 0.125 M Glycine, followed by two DPBS washes. The supernatant was discarded, and cells were stored at −80 °C until use.

### Cell lysis and nuclear lysis

For the dual-crosslinked cell pellets (generally, 100 million cells), they were resuspended in 20 mL of cell lysis buffer (50 mM Tris-HCl pH 7.0, 150 mM NaCl, 1 mM EDTA, 0.1% SDS, and 1× Protease Inhibitor cocktail, Roche, 11836170001) and incubated at room temperature for 15 min with rotation. Subsequently, 1.8 mL of 10% SDS was added to the mixture and incubated at 37 °C for 10 min with rotation to facilitate nuclear membrane permeabilization. The permeabilization process can be monitored under a microscope, extending the incubation time as necessary. The nuclei were then centrifuged at 4000 rpm for 20 min and washed once with cell lysis buffer. After discarding the supernatant, 15 mL of chromatin tagmentation buffer (50 mM Tris-HCl pH 7.0, 150 mM

NaCl, 1 mM EDTA, 1% Triton X-100, 0.1% SDS, and 1× Protease Inhibitor cocktail) was added. The sample was then incubated on ice in preparation for chromatin tagmentation.

### Chromatin tagmentation

Following the lysis of cells and nuclei, the nuclei were subjected to sonication to achieve chromatin fragmentation into pieces ~3 kb in size. The fragmented chromatin was then centrifuged at 4000 rpm for 20 min at room temperature. The resulting supernatant was divided equally into three 14-mL tubes to be used in subsequent experiments.

### Preclear chromatin with C1 beads

Approximately 14 mL of the fragmented chromatin was mixed with 50 μL of MyOne Streptavidin C1 beads (Thermo Fisher Scientific, 65001) and rotated at room temperature for 20 min. This step, known as preclearing, helps reduce the nonspecific adherence of chromatin fragments to magnetic beads in subsequent steps. Following this, the mixture was placed on a magnetic rack for 1 min to separate the beads. The clear supernatant was then carefully transferred into a fresh 50-mL tube, setting the stage for the hybridization process.

### Hybridization

The precleared chromatin was mixed with a volume of hybridization buffer (50 mM Tris-HCl pH 7.0, 750 mM NaCl, 1 mM EDTA, 1% SDS, 15% Formamide) that was double its own, and the mixture was rotated at room temperature for 30 min. Following this, 2 μL of 100 μM biotinylated probes were added to each tube. The tubes were then incubated at 37 °C with rotation overnight.

### Chromatin immobilization

The chromatin bound to biotinylated probes was immobilized by incubating with 300 μL of C1 beads pre-blocked with 2% I-Block Buffer at room temperature for 2.5 h. Afterward, the C1 beads with the bound biotinylated probes and RNA-chromatin complex were separated using magnets and transferred to a new 1.5 mL tube. The beads were then washed five times with 800 μL of wash buffer (2× SSC, 1% SDS) and three times with TE buffer. Each wash was performed at 37 °C for 5 min with rotation.

### Blocking extra sites on C1 beads

To ensure that the streptavidin on the C1 beads does not interact with biotinylated linkers in the next steps (notably, the bridge linker), the procedure involves treating the chromatin-bound C1 beads with β-mercaptoethanol (β-ME, Thermo Fisher Scientific, 21985023) neutralized IPB (Thermo Fisher Scientific, 21334) for 15 min. This step is critical to saturate any unbound streptavidin sites on the beads, preventing unwanted interactions. After the incubation, the chromatin is rigorously washed five times with TE buffer supplemented with a 1× Protease Inhibitor cocktail. This thorough washing ensures the removal of any residual β-ME and prepares the chromatin-coated beads for the next stage in the RDD library construction process.

### Chromatin DNA end repair

The DNA fragments of chromatin complexes, which were enriched for specific RNA and bound to C1 streptavidin beads for RDD, underwent an end-repair process. This preparation was crucial for subsequent ligation with the RDD-linker (also known as the bridge linker). In detail, the chromatin complexes attached to the beads were processed with a reaction mixture containing T4 DNA polymerase (NEB, M0203S)—composed of 70 μL of 10× NEB buffer 2.1, 7 μL of 10 mM dNTP, 613 μL of Nuclease-free water, and 7.0 μL of T4 DNA polymerase. This mixture was supplemented with 3.0 μL of SUPERase•In™ RNase Inhibitor (Thermo Fisher Scientific, AM2696, referred to as RI in this study). The mixture was then incubated with rotation at 37 °C for 40 min.

### A-tailing

The chromatin complexes were then processed for A-tailing using a 700 μL master mix of Klenow (3′-5′ exo-, NEB, M0212L). This mix consisted of 70 μL of 10× NEB Buffer 2, 7 μL of 10 mM dATP, 613 μL of Nuclease-free water, and 7 μL of Klenow enzyme. The mixture was incubated at 37 °C for 50 min with rotation. Following incubation, the complexes were washed three times with ice-cold Wash Buffer and once with TE buffer.

### Proximity ligation

The chromatin DNA, after undergoing A-tailing, was subjected to proximity ligation using an RDD-linker that featured a biotin modification (T) and 3′ nucleotide T over-hanging (T), and a 5′ phosphate modification ($_P$). The detailed structure of this linker is described below.

$$5'-_PCGCGATGGC\underline{\mathbf{T}}ACTCTGACT-3'$$
$$3'-TGCGCTACCGATGAGACTG_P-5'$$

For the ligation process, a proximity ligation buffer was prepared, consisting of 140 μL of 10× T4 DNA ligase buffer, 700 ng of RDD-linker, 1243 μL of Nuclease-free water, 3 μL of RI (RNase Inhibitor), and 10 μL of 20 U/μL T4 DNA ligase (NEB, M0202s). This buffer was added to the A-tailed chromatin complexes, and the mixture was rotated at room temperature for 60 min. Subsequently, the mixture was transferred to 16 °C to allow ligation to proceed overnight.

### Elution of ligated DNA

Chromatin DNA was released from the C1 beads by incubation in 480 μL of Elution Buffer (50 mM Tris.HCl pH 7.5, 0.5% SDS), supplemented with 20 μL of proteinase K (Ambion, AM2546), at 65 °C overnight while being agitated at 950 rpm on a Thermomixer (Eppendorf). Following incubation, the supernatant was carefully transferred into a Maxtract tube (Qiagen, 129046) and gently mixed with 500 μL of phenol:chloroform:IAA (Solarbio, P1012). The mixture was then centrifuged at 12,000 rpm for 10 min at room temperature. The aqueous phase was collected into a new 1.5 mL tube, to which 52 μL of 3 M sodium acetate pH 5.5 (Ambion, AM9740), 1 μL GlycoBlue (Thermo Fisher, AM9516), and 520 μL of ice-cold isopropanol (Sigma, I9516) were added. This mixture was then stored at −80 °C overnight to facilitate DNA precipitation.

### Purification of ligated DNA

The chromatin DNA mixture, previously stored at −80 °C, was removed and centrifuged at 12,000 rpm for 1 h at 4 °C. Subsequently, the DNA pellet that precipitated was washed twice with 75% ethanol and then dried using a vacuum. After drying, the pellet

was dissolved in EB buffer. To further purify the DNA and remove short DNA fragments, AMPure XP beads (Beckman, A63881) were used, following the instructions provided by the manufacturer.

### RDD library generation and sequencing

The purified DNA was fragmented using Tn5 transposase (Vazyme, TD502). Subsequently, DNA fragments containing the RDD-linker were captured using Streptavidin beads (M-280, Life Technologies, 11205D) and then amplified via PCR. The PCR products underwent a double size-selection process using AMPure beads to isolate DNA fragments ranging from 300 to 600 bp in size. These selected fragments were then prepared for sequencing on the Illumina NovaSeq 6000 system, using paired-end sequencing with read lengths of 150 bp.

## Protein-associated chromatin DNA–DNA interactions (ChIA-PET)

In this research, interactions between chromatin DNA associated with the protein factor were identified using the ChIA-PET technique, adhering to the previously established ChIA-PET protocol. First, chromatin was enriched using specific antibodies through the ChIP method, including an anti-RNAPII monoclonal antibody (8WG16, BioLegend, Cat# 664906), an anti-HSPA9 antibody (ABclonal, Cat# A0558), and an anti-nucleolin (NCL) antibody (Abcam, Cat# ab22758). Chromatin enriched with RNAPII and NCL was subsequently used for ChIA-PET library construction.

## BALM imaging

### Sample preparation

To induce EBV production from B95-8 cells, initially treat the cells with TPA for 1 h, which activates viral replication. Following TPA treatment, collect the culture medium and centrifuge it at 4 °C, $600 \times g$ for 10 min to separate cells and debris, retaining the supernatant which contains EBV. Finally, filter this supernatant through a 0.45-μm filter to purify the virus, removing any remaining cellular debris and ensuring a preparation rich in EBV particles for further use (Hui-Yuen et al, 2011).

EBV genome was extracted from B95-8 cells following the above protocols. The sample was then washed with DPBS, fixed with 1% formaldehyde, and concentrated using an ultracentrifugal filter. After extraction, the EBV genome was treated with proteinase K, followed by a purification process. Subsequently, both the EBV genome and the purified DNA were attached onto the polylysine-coated coverslips for visualization through super-resolution imaging.

### Microscope setup

Super-resolution imaging was conducted at room temperature (RT, 24 °C) using a custom-built microscope outfitted with a high numerical aperture (NA) objective lens (NA 1.5, UPLAPO 100XOHR, Olympus). The setup utilized four lasers: a 405 nm laser (iBEAM-SMART-405-S, 150 mW, TOPTICA Photonics), a 488 nm laser (iBEAM-SMART-488-S-HP, 200 mW, TOPTICA Photonics), a 561 nm laser (MGL-FN-561nm, 300 mW, CNI), and a 640 nm laser (iBEAM-SMART-640-S-HP, 200 mW, TOPTICA Photonics). These lasers were combined by a fiber coupler (PAF2-A4A, Thorlabs) and transmitted through a single-mode fiber (P3-405BPM-FC-2, Thorlabs).

### BALM imaging

BALM imaging was performed with 561 nm laser (4 kW cm–2 intensity) at 0.5-5 nM Sytox Orange (S11368, Thermo Fisher) in imaging buffer. The imaging buffer contained 50 mM Tris pH 8.0, 1 mM EDTA, 10 mM NaCl supplemented with 40 mg/mL glucose (G8270, Sigma-Aldrich), 0.5 mg/mL glucose oxidase (G7141, Sigma-Aldrich), 40 μg/mL catalase (C100, Sigma-Aldrich), 35 mM MEA (30070, Sigma-Aldrich) and 10 mM ascorbic acid (PHR1008, Sigma-Aldrich). The astigmatic 3D imaging was acquired using a cylindrical lens to introduce astigmatism. A total of 50,000 frames were acquired with 25 ms exposure. These raw images were then processed and reconstructed into a super-resolution 3D image using the Superresolution Microscopy Analysis Platform (SMAP).

## DNA-FISH STORM

### DNA probe design and generation

To generate DNA probes across the entire EBV genome, a series of primers are designed to target sequences, resulting in PCR products between 2 and 5 kb. Following PCR amplification, the products undergo homemade Tn5-Cy5 tagmentation to fragment and simultaneously label them with Cy5, aiming for fragment sizes of 100–400 bp. The final Cy5-tagged DNA fragments serve as probes for DNA-FISH.

### Cell preparation

RAMOS and GM12878 cells in culture medium were seeded on the Fisherbrand Cover Glasses (Fisher Scientific 12-545-81) coated with Poly-L-Lysine (Sigma, P4707) for 30 min at room temperature followed by washed with DPBS once and kept dry in air for around 2 h.

### DNA-FISH

The cells on the coverslips were subjected to a DNA-FISH experiment as previously described (Zheng et al, 2019).

### STORM imaging and analysis

The sample was mounted onto a custom-manufactured sample holder in freshly prepared imaging buffer, and the holder was sealed with parafilm. Astigmatic 3D STORM imaging was performed using a 640 nm laser, with a cylindrical lens introduced to create astigmatism. The microscope's software control was integrated into Micro-Manager with EMU (Deschamps and Ries, 2020). We employed a highly inclined and laminated optical sheet (HILO) technique and acquired 30,000 frames at a 20 ms exposure time, with 640 nm laser power densities of ~15 kW/cm². The pulse length of the 405 nm laser was automatically adjusted to maintain a constant number of localizations per frame. The raw images were analyzed and compiled into a super-resolution 3D image by the Super Resolution Microscopy Analysis Platform (SMAP) using a spline PSF fitter (Ries, 2020; Li et al, 2018b).

## CRISPRi-mediated knockdown of ncRNA EBERs in GM12878

### Plasmid generation

The plasmid pRS001, which expresses dCas9-KRAB-MECP2 and GFP, was derived from the lenti_dCas9-KRAB-MeCP2 plasmid (Addgene #122205). The BSD element in this plasmid was replaced

with GFP using the Gibson cloning method. To construct plasmids expressing sgRNAs, annealed oligonucleotides containing protospacer sequences were ligated into the BlpI and BstXI restriction sites of the sgRNA expression backbone, pLG15. The sgRNA sequences used in this study were as follows: for *EBERs*, sg1: GGCTGGTACTTGACCGAAGA, sg2: GAAAACCTCTAGGG-CAGCGT, sg3: GAGACACCGTCCTCACCACC; and for the non-targeting control (NC) sgRNA: GTCCACCCTTATCTAGGCTA.

### Lentivirus generation

Lentivirus was generated by transfecting HEK293FT cells grown in 10-cm dishes with complexes containing 5 µg of the transfer plasmid, 5 µg of psPAX2, 5 µg of pMD2-G, and 45 µg of PEI diluted in Opti-MEM. Three days post-transfection, the supernatant was collected and filtered through 0.45-µm sterile filters. The lentivirus was then concentrated using the Lenti-X Concentrator (Takara) and resuspended in DPBS.

### Obtained stable CRISPRi-EBERs GM12878 cells

To achieve stable expression of dCas9 in GM12878 cells, we utilized flow cytometry to sort GM12878 cells containing GFP twice. Following this, the cells were prepared for sgRNA lentivirus infection to assess the efficiency of the infection process. Initially, well-vital GM12878-Mecp2 stable cell lines were centrifuged at 1000 rpm for 5 min in a 15-mL tube, six to ten hours before the infection. The cells were then resuspended in freshly prepared complete medium, pre-warmed to 37 °C, and seeded at a density of $1.6 \times 10^6$ cells per well in 1 mL of medium across four wells of a six-well plate. For the infection, 1 mL of the corresponding sgRNA lentivirus was added to each well, and 2 µL of polybrene was added at a final dilution of 1:1000 with the medium, followed by gentle mixing and incubation. After three hours, the six-well plate was gently shaken to mix the cell-virus system thoroughly. Two days post-infection, the cell suspension was centrifuged at 1000 rpm for 5 min, and the cells were resuspended in 2 mL of fresh complete medium, pre-warmed to 37 °C, for a medium change. An appropriate dose of puromycin was then added to select for strains stably transduced with sgRNA.

### siRNA-mediated knockdown of NCL

Cells were seeded onto plates and cultured for 10–16 h. Thirty min before transfection, the medium was replaced with a mixture of fresh growth medium and Opti-MEM to optimize cell preparation. siRNA (Santa Cruz, sc-29230, siNCL; scrambled siRNA, siNC) and Lipofectamine 2000 solutions (Invitrogen, 11668019) were separately incubated in Opti-MEM at room temperature for 10 min each, then combined and incubated for an additional 20 min. This transfection solution was added drop by drop to the cells, which were then incubated at 37 °C for 10 h. After transfection, the medium was replaced with fresh growth medium, and the cells were cultured at 37 °C until ready for harvesting and western blot testing.

### RNA-seq

The RNA-seq experiment commenced with the extraction of total RNA from GM12878 cells using the Phenol-chloroform method. Subsequently, ribosomal RNA (rRNA) was depleted from the total RNA samples using the Ribo-zero™ Kit. After the removal of rRNA, stranded RNA sequencing libraries were prepared utilizing the NEBNext® Ultra™ Directional RNA Library Prep Kit for Illumina®, following strictly to the guidelines provided by the manufacturer. The RNA-seq libraries were then sequenced on an Illumina NovaSeq 6000 sequencer with a $2 \times 150$ bp sequencing module.

## NCL-interacted chromatin RNA assay (termed NCL-RNA)

### Chromatin complexes preparation

To investigate NCL-interacted chromatin RNAs, we employed a dual-crosslinking strategy using formaldehyde and EGS to preserve chromatin integrity and capture chromatin-associated complexes, as described in the RDD and ChIA-PET methods. Crosslinked chromatin was subjected to immunoprecipitation using NCL-specific antibodies (Abcam, Cat# ab22758) to enrich chromatin-associated complexes. The NCL-enriched complexes were then processed with TRIzol reagent to isolate RNA for downstream analysis. This method enables the identification of RNAs associated with NCL within the chromatin context.

### RNA isolation

To isolate RNA, add 100 µL of chloroform to the TRIzol-treated samples, followed by vigorous vortexing and centrifugation at 16,100 rcf for 15 min at 4 °C. Carefully transfer the aqueous supernatant to a new tube and mix it with 600 µL of 100% ethanol. Perform RNA purification using a miRNeasy Micro Kit, loading the sample onto the RNeasy MinElute spin column, centrifuging, and discarding the flow-through. Wash the column with Buffer RWT and treat with DNase I to remove DNA contaminants, followed by additional washes using Buffer RWT, Buffer RPE, and 80% ethanol. Elute the RNA with RNase-free water, and finally, heat the RNA sample at 65 °C for 15 min to inactivate any residual DNase.

### First-strand cDNA synthesis

For first-strand cDNA synthesis using the kit (Invitrogen, 18080051), start by combining 13 µL of RNA, 1 µL of random hexamers, and 1 µL of dNTP mix in a reaction tube. Incubate this RNA mixture at 65 °C for 5 min, then transfer it to ice for at least 1 min. Next, prepare the cDNA Synthesis Mix by sequentially adding the following components to the above RNA mixture: 3 µL of RT buffer, 4 µL of nuclease-free water, 4 µL of MgCl₂, 2 µL of DTT, 1 µL of RNaseOUT, and 1 µL of SuperScript III RT, mixing gently after each addition. Incubate the reaction at 25 °C for 10 min, followed by 50 °C for 50 min.

### Second-strand cDNA synthesis

For second-strand cDNA synthesis, start by placing the 30 µL first-strand cDNA reaction sample on ice. Add the following components to the first-strand reaction tube: 81 µL of nuclease-free water, 30 µL of 1× Second-Strand Reaction Buffer (Thermo Fisher, 10812014), 3 µL of dNTP mix, 1 µL of *E. coli* DNA Ligase (NEB, M0205S), 4 µL of E. coli DNA Polymerase I (NEB, M0209S), and 1 µL of *E. coli* RNase H. Mix the solution thoroughly and incubate it for 2 h at 16 °C. Then, add 3.3 µL of T4 DNA Polymerase (NEB, M0203S) to each reaction, and continue incubation on a Thermomixer at 1,100 rpm and 12 °C for 15 min Once completed, add 10 µL of 0.5 M EDTA to stop the reaction.

Finally, purify the cDNA to prepare it for library construction and sequencing.

## ATAC-seq

For the ATAC-seq experiments, 1 million cells were collected and subjected to permeabilization. Following this, the lysed nuclei were resuspended in a Tagmentation reaction mixture to achieve a concentration of 10,000 nuclei per microliter (µL), with 50,000 nuclei being utilized for the Transposition reaction using the Vazyme kit (TD501). This mixture was incubated at 37 °C for 30 min with agitation. Afterwards, the transposed DNA was purified using either a Qiagen MinElute or a Zymo kit, and then it was subjected to the next steps of PCR amplification and sequencing.

## Validation of NCL protein interact with *EBERs* ncRNA

### Plasmid construction

The NCL gene was PCR-amplified from the pCMV-mCherry-NCL (human)-3×HA-Neo plasmid (obtained from Miaolingbio, http://www.miaolingbio.com/) and cloned into the PET28a vector between EcoR1 and BamH1 sites, including an N-terminal 8His-TEV tag. This strategy generated several constructs: PET28a-NCL, PET28a-NCL-ΔN (delete N), PET28a-NCL-RRM1-4, PET28a-NCL-RRM1, PET28a-NCL-RRM2, PET28a-NCL-RRM3, PET28a-NCL-RRM4, and PET28a-NCL-RRM2-4. The plasmids were transformed into E. coli Rosetta cells for protein expression.

### Protein purification

NCL protein variants were produced in E. coli Rosetta. Cultures were grown at 37 °C to an $OD_{600}$ of about 0.6, induced with 3 mM IPTG, and incubated overnight at 16 °C. Cells were harvested, lysed by sonication in a buffer containing 50 mM Tris-HCl (pH 8.0) and 200 mM NaCl, then centrifuged at $20,000 \times g$ for 40 min at 4 °C. The supernatant was loaded onto $Ni^{2+}$-NTA agarose beads, washed with lysis buffer containing 20 mM imidazole, and eluted with 300 mM imidazole. The proteins were dialyzed against buffer A (25 mM Tris-HCl, pH 8.0, 100 mM NaCl), purified by anion exchange chromatography using a HiTrap Q HP column with a 0–100% gradient of buffer B (50 mM Tris-HCl, pH 8.0, 1 M NaCl) over 60 mL, then eluting at ~20% buffer B. After exchanging into buffer C (50 mM Tris-HCl, pH 8.0, 150 mM NaCl), proteins were further purified by size exclusion chromatography on a Superdex 200 Increase column. Purified proteins were concentrated, flash-frozen in liquid nitrogen, and stored at −80 °C. Using this protocol, various NCL variants were obtained, except for full-length NCL.

### ncRNA purification

ncRNA was isolated using in vitro transcription from a plasmid containing the T7 promoter and ncRNA coding sequence. The T7 DNA fragment was PCR-amplified and used in a reaction with 22 mM HEPES (pH 7.3), 120 mM KOAc, 2 mM MgOAc, 0.75 mM ATP, 0.1 mM GTP, 25 mM creatine phosphate, 1.6 mM DTT, 0.75 mg/mL T7 polymerase, and PCR product, carried out at 37 °C for 2 h. Post-transcription, the mixture was centrifuged at $20,000 \times g$ for 10 min at 4 °C. The supernatant was mixed 1:1 with 5 M LiCl and precipitated at −20 °C for at least 1 h. The precipitate

was re-centrifuged, resuspended in nuclease-free water, aliquoted, and stored at −80 °C.

### Electrophoretic mobility shift assay (EMSA)

EMSA was performed using RNA transcribed in vitro and NCL proteins. Each 30 µL reaction contained 1 µM RNA (*EBER1*) substrate and varying molar ratios of NCL proteins (1:1 to 1:11) in 50 mM Tris-HCl (pH 8.0) and 150 mM NaCl. After 30 min on ice, 6 µL of 6× loading buffer (50% sucrose, 0.06% bromophenol blue, 0.06% xylene cyanol) was added. The samples were run on a non-denaturing 8% polyacrylamide gel in 1× TBE buffer. The gel was stained with GelRed, and band intensities were visualized using a Tanon 3500 gel imaging system.

## Data analyzing environment

Unless otherwise specified, in this study, data processing is performed under the GNU Bash (v4.3.48) and Python (v3.7.7) environments in Ubuntu (v16.04). Statistical analysis and plot drawing are conducted using R language (v4.11) within the RStudio (v2021.09.0 Build 351) platform.

## Preparing indexing of hg38-EBV hybrid reference genome

To map the GM2878 data to both the human and EBV genomes, a hybrid reference genome of *hg38B* was prepared, combining the human genome (*hg38*) with the EBV genome (B95-8, *chrB*). The *hg38* reference genome (FASTA) was obtained from the UCSC Genome Browser (Kent et al, 2002) (https://hgdownload.soe.ucsc.edu/goldenPath/hg38/bigZips/hg38.fa.gz). The EBV genome (B95-8, *chrB*) reference genome (FASTA) and gene annotation (GTF) were downloaded from GeneBank (Benson et al, 2012) (Human gammaherpesvirus 4 complete genome, https://www.ncbi.nlm.nih.gov/nuccore/NC_007605.1).

## RDD and ChIA-PET data processing

We utilized the ChIA-PIPE (Lee et al, 2020) pipeline to analyze RDD and ChIA-PET datasets, incorporating tailored functions and modifications to boost its performance. This advanced pipeline streamlines the processing through multiple critical steps: linker filtering to identify valid reads, precise data alignment and qualification for accurate mapping, coverage generation to highlight genomic regions of interest, interaction clustering for identifying chromatin interactions, rigorous quality control to ensure data integrity, and species-specific data classification.

Initially, sequencing generates two FASTQ files for R1 and R2 reads. These reads are screened for linker sequences according to RDD or ChIA-PET protocols, identifying chromatin interactions. Reads containing linker sequences are split into two tags at the linker; if both tags are longer than 18 bp, they form paired-end tags (PETs), which are essential for detecting chromatin interactions. These sequences are then mapped to the *hg38B* hybrid reference genome. Only properly (MAPQ ≥ 30) and uniquely mapped, non-redundant sequences are retained as qualified PETs. Next, these qualified PETs are classified based on their genomic span into self-ligation and inter-ligation categories, using an 8-kb threshold. Self-ligation PETs represent tags originating from the same genomic fragment, indicating local interactions, whereas inter-ligation PETs

point to long-distance chromatin interactions, likely mediated by specific RNA molecules or proteins.

For the final contact results and visualization, inter-ligation PETs are processed through two clustering methods. The first method segments the entire genome into fixed-length segments, or bins, and evaluates the intensity of interaction between any two bins by tallying all PETs precisely positioned between them. The software Juicer (Durand et al, 2016b) (v1.19.01) is used to generate a HIC file ('.hic'), while Juicebox (Durand et al, 2016a) (v1.11.08) browser facilitates the visualization of inter-ligation PETs via a contact heatmap. In addition, Juicebox produces and displays a Pearson correlation heatmap, indicating the spatial segregation of chromatin into A/B compartments (Lieberman-Aiden et al, 2009). The color depth of each cell in the matrix reflects the interaction intensity between the two genomic loci (one from the x-axis bin and the other from the y axis bin). The second visualization approach aims to identify confident chromatin interaction clusters (Loops) by merging similar PETs, with the requirement that the tags on both ends of these PETs overlap separately with an extension of 500 bp. These loops are then classified as either intrachromosomal or inter-chromosomal, depending on the chromosomes of the two anchors. They are also categorized into H–H, E–E, and E–H types, based on the species genome to which the two anchors are mapped. The H–H and E–E loops can be displayed as either a "Cluster" or a "Loop" track in the BASIC Browser (Lee et al, 2020), where the height of a loop indicates the interaction frequency (strength) between the two anchors. An E–H interaction is represented by two anchors located in separate tracks for different genomes (EBV and hg38) within the BASIC Browser.

In addition, Sequences with only one valid tag indicate regions targeted by a specific factor of RNA or protein, aiding in the creation of genomic "Coverage" and "Peak" tracks to identify enriched loci.

Every step of the ChIA-PIPE process is overseen by guardant scripts, which compile a statistical table known as the quality control (QC) table to reflect the quality of the library. This QC table offers crucial insights into the quality of a RDD or ChIA-PET library. Metrics such as the "Fraction_read_pairs_with_linker" gauge the efficiency of linker ligation. A higher "ratio of intra/inter_PET" and "ratio of intra/inter_Loops" typically signals a library of superior quality. Furthermore, the "Loops_of_PET_count" within "Inter-chr_Loops" helps in determining the frequency-based distribution of interactions. Categories like "H-H_Loops", "H-E_Loops", and "E-E_Loops" shed light on the interaction patterns between host and EBV.

## ChIP-seq and ATAC-seq data processing

ChIP-seq and ATAC-seq data were processed using BWA (Li and Durbin, 2009) (v0.7.17) to align the raw FASTQ to the hg38B reference genome. Subsequently, SAMtools (Li et al, 2009) (v0.1.19) was employed to obtain high-quality reads that are uniquely mapped and de-duplicated, with a mapping quality (MAPQ) greater than 30. Following the alignment and processing, "Coverage" and "Peak" tracks visualized in the BASIC Browser were generated using the deepTools (Ramírez et al, 2016) (v3.5.4) and MACS2 (Zhang et al, 2008) (v2.2.9.1), respectively.

## ChIP-seq and ATAC-seq data reproducibility

To assess the reproducibility of ChIP-seq and ATAC-seq data, signal strength at binding or accessible chromatin sites was evaluated. The process involved merging peaks from two replicates to create union peaks, followed by calculating the pileup read counts for each peak. These union peaks, containing values from both replicates, were then used to generate a scatter plot to demonstrate repeatability. Additionally, a two-sided Pearson correlation coefficient was commonly computed to quantify the correlation between the replicates, providing a numerical measure of reproducibility (Fig. EV10).

## RNA-seq data and NCL-interacted chromatin RNA data processing

The RNA-seq data and the NCL-interacted chromatin RNA data were processed using STAR (Dobin et al, 2013) (v2.7.9a) to align the raw FASTQs to the hg38 and chrB genomes separately. Subsequently, RSEM (Li and Dewey, 2011) (v1.3.1) was utilized to calculate fragments per kilobase per million (FPKM) values of genes. SAMtools was then used to filter reverse-strand reads ("-f 128") to generate a two-value BEDGRAPH file (one value for "strand+", another value for "strand–") for RNA-seq "Coverage" in the BASIC Browser.

## RNA-seq data and NCL-RNA data reproducibility

To evaluate the reproducibility of RNA-seq and NCL-RNA data, FPKM values of genes obtained from the processing pipeline were utilized. These values were used to generate a scatter plot, which visually demonstrated the repeatability of the data. A two-sided Pearson correlation coefficient was calculated to quantify the correlation between replicates (Figs. EV8A and EV10C).

## Evaluation of EBERs expression enrichment in RNA-seq and NCL-RNA in EBV

To evaluate the expression levels of EBERs in comparison to selected control coding genes (LMP-2B, Cp-EBNA3A, Cp-EBNA3B, LMP-1, LMP-2A) of EBV, we calculated the expression ratios of EBERs to these genes using RNA-seq and NCL-RNA data, respectively. The results were then plotted as a bar chart, and statistical significance was assessed using a t test.

## Simulation of EBV 3D model using chromatin interaction contacts

The EBV 3D chromatin structure model was constructed using the HSA tool (Zou et al, 2016) in R (v3.6.3). The model utilized E–E loop files (PET count ≥2) as input data, derived from CTCF and RNAPII ChIA-PET libraries, as well as EBERs, miR-BART, v-snoRNA1, and MALAT1 RDD libraries. The entire procedure comprised three steps: matrix preparation, backbone simulation, and fine-tune simulation. Finally, 3D structure was visualized using the package rgl (v0.106.8) in R (v4.11). We have developed a sophisticated Shiny (v1.8.1.1) application that hosts an online interactive 3D genome structure model of EBV (https://3dgenome.shinyapps.io/EBV3DMODEL_HSA/).

Preparedly, according to PET count (m), each loop was extracted into m PETs with the same coordinates as their original loop. Then these PETs were used to generate the HSA-required contact matrix at a certain resolution using the middle points of the

anchors. This matrix includes ($n$) rows and ($n + 2$) columns (Column 1 is the start position, Column 2 is the end position, and Columns 3 to ($n + 2$) represent the ($n \times n$) contact map in matrix form with all entries being nonnegative). Thus, the HSA-required matrix is prepared for simulating in the HSA toolkit.

For the Round-1 backbone simulation, we generated a 5-kb resolution HSA-required matrix using method above, then, the matrix was processed by cstruct1.R function of HSA for 300 iterations). This step is to study its three-dimensional coordinates, with parameters configured in myR.R script as follows: "maxiter = 300, submaxiter = 150, lamda = 50, leapfrog = 50, epslon = 0.0003, mkfix = 0, rho = 0, mk = 0, initialS = NULL".

For the Round-2 fine-tune simulation, we inherited the calculation result from Round-1 as the initial 3D structure to re-run HSA program, at a 200-bp bin scale. The Markov coefficients and Indicator are set to 1 because the percentage of nonzero entries in the contact map is lower than 10%. The parameters for this simulation are: "maxiter = 1, submaxiter = 30, lamda = 10, leapfrog = 20, epslon = 0.0003, mkfix = 1, rho = 0, mk = 1, initialS = backbone".

## A/B compartments calling

We called A/B compartments from GM12878 CTCF ChIA-PET contact map (HIC file) using Juicer with 'eigenvector' command. For EBV genome *chrB*, we used a resolution parameter of 3 kb, and for *hg38* genome we used 250 kb.

## Circos plot visualizing E–E interaction

The Circos plots consist of two components: surrounded annotation tracks and center interaction curves. The surrounded tracks were plotted using an Shiny (v1.6.0) application called shinyCircos (Yu et al, 2018), incorporating data signals from ChIP-seq, ATAC-seq, ChIA-PET, RNA-seq, or A/B compartment. The center curves represent RDD chromatin interactions of E–E loops and were generated using the R package circlize (Gu et al, 2014) (v0.4.13).

## Cytoscape displaying chromatin interaction

The chromatin interacting loops were organized as networks in Cytoscape (Shannon et al, 2003) (v3.10.1), with the interacted DNA fragments represented as nodes connected by loops represented as edges. The stroke of each edge reflects its PET count.

## STORM image foci remodeling and clustering

To further analyze the high-resolution imaging STORM data of EBV in individual host cells, we utilized the Bayesian Information Criterion (BIC) prediction method within the Gaussian Mixture Models (GMM) clustering algorithm to automatically generate clustering results, using the R package mclust (Scrucca et al, 2016). The BIC analysis indicated a most suitable clustering number. The results were visually represented as a scatter plot using the original image coordinates, employing the plotly (v.4.10.2) package in R to display the entire picture, and using the rgl (v0.106.8) package in R to visualize individual clusters.

## HiCRep analysis reproducibility of a pair of chromatin interaction data

The similarity of RDD and/or ChIA-PET libraries was assessed using the R package HiCRep (Yang et al, 2017) (v1.12.2), which calculates the stratum-adjusted correlation coefficient (*scc*) of the ChIA-PIPE produced contact maps for all pairs of RDD libraries. HiCRep was executed with the following parameters: contact maps were at 250 kb resolutions; the normalization option used was 'NONE' (without consideration of the assumptions of matrix balancing); the smoothing size ($h$) was set to 3; and the maximum interaction distance was defined as 5 Mb. Subsequently, the *scc* was averaged across chromosomes of the RDD libraries with pairs, with ChIA-PET libraries serving as the control (Figs. EV4F and EV8C).

## Spring calling algorithm

Spring calling algorithm is designed to identify 'spring' structures, characterized in contact heatmaps by bins arranged in a solitary line along the antidiagonal direction, which involves four steps.

### Calculating the diagonal occupancy ratio to define baseline data

In contact heatmaps, interactions near the diagonal are typically dense, with bin occupancy rates reaching at least 50%. Therefore, when defining structures on the antidiagonal, it's essential to disregard the dense areas near the diagonal, referred to as baseline data in this text. Starting from the diagonal data and moving along the antidiagonal direction, we calculate the data occupancy rate (*dor*) for each bin layer. For bin layers with a *dor* greater than the baseline threshold, these are defined as baseline data. The pseudocode is as follows:

1. from diagnose along antidiagonal direction:
2. if *dor>BG*:
3. baseline data
4. continue searching
5. else:
6. break loop
7. end searching

where *dor* = (count(antidiag.bin_all)/(count(antidiag.bin_all), and *BG* is the baseline threshold, and *BG* is set to 0.75 in this work.

### Defining bandwidth

Moving in the opposite direction along the antidiagonal, and disregarding baseline data, we select a certain width (in bins) as our bandwidth for calling a spring.

### Calculating antidiagonal occupancy ratio to select candidate springs

Within the bandwidth, we calculate the antidiagonal occupancy ratio (*aor*) as:

$$aor = \frac{\text{count(antidiag.bin\_withdata\_within\_bandwidth)}}{\text{count(antidiag.bin\_all\_within\_bandwidth)}}$$

### Calculating prominence score to eliminate spring noise

A candidate spring is considered sufficiently prominent and defined as a spring if (I) the area of the triangle formed by the height of the

bin furthest from the diagonal (bin number from diagnose) and the average height of its flanking area bins (±2 bins) is greater than or equal to the prominence threshold (PG); (II) the antidiagonal average occupancy ratio of the flanking area is below the low point threshold (LG); and (III) the difference in the average height of the flanking area bins is less than the difference threshold (DG).

$$spring = \begin{cases} ps \geq PG \\ laor, raor < LG \\ |laor - raor| < DG \end{cases}$$

where, $ps$ is a prominence score, $laor$ is the left flanking area antidiagonal mean occupancy ratio of the candidate spring, $raor$ is the right flanking area antidiagonal mean occupancy ratio of the candidate spring. The thresholds used here are "$PG = 0.3$, $LG = 0.5$, $DG = 0.5$".

### Calculating prominence score to eliminate spring noise

To analyze the characteristics of spring, we categorized the defined 2628 CCDs into two groups: "CCD inside spring" for those with at least a 1 base pair overlap with spring, and "CCD outside spring" for those without any overlap. We mapped uniquely mapped reads or loop anchors (with a PET count of ≥2) of ncRNA from RDD and CTCF from ChIA-PET to these groups. Subsequently, we calculated the RPKM to evaluate the signal enrichment and interactivity of the CCDs. These RPKM values were visualized using box plots, and a Wilcoxon test was performed to assess statistical differences between the groups.

### Signal aggregation of spring

We selected the 240 "spring" regions and, based on their lengths, randomly chose four groups of 240 control regions from genomic areas outside the 'spring' regions. Each region was evenly divided into 40 bins of equal length. To account for 'spring' boundary effects, two additional bins were added on each side, resulting in a total of 44 bins per region. The pileup values from ncRNA- and protein-enriched peaks within these regions were summed and normalized to Reads Per Million (RPM). Finally, the values for corresponding bins across all regions were aggregated to create a line plot.

### Time course transcriptomic analysis of EBV

Download the 'All expression results in the experiment' dataset from the study on the infection of naïve primary human B lymphocytes with EBV—a time course transcriptomic analysis during the pre-latent phase of viral infection (Mrozek-Gorska et al, 2019), available in the Expression Atlas portal (https://www.ebi.ac.uk/gxa/experiments/E-MTAB-7805/Downloads?ref=biostudies) of the European Molecular Biology Laboratory's European Bioinformatics Institute (EMBL-EBI) (https://www.ebi.ac.uk).

For the data we downloaded, we first isolated genes that coincide with those targeted by loops within the EBV hotspot regions: HS1, HS2, and HS3. Following this, we extracted the fold change values for EBV at "1" day compared to none at "0" days, and similarly "14" day compared to none at "0" days. Utilizing ggplot2 (v3.4.4) in R, we then created spaghetti plots to visually represent these changes.

### Genomic region annotation by genomic location

In this study, genomic regions, including RNA or protein-enriched peaks and loop anchors, were annotated to genomic locations using the annotatePeaks.pl script from the HOMER (Heinz et al, 2010) (v4.10) suite.

### Genomic region annotation of chromatin states and A/B compartments

In this study, including RNA or protein-enriched peaks and loop anchors, genomic regions were annotated to the chromatin states or A/B compartments using BEDTools (Quinlan and Hall, 2010) (v2.29.1). The chromatin states of *hg38* were obtained from the public NIH Roadmap Epigenomics Project (https://egg2.wustl.edu/roadmap/data/byFileType/chromhmmSegmentations/ChmmModels/coreMarks/jointModel/final/E116_15_coreMarks_hg38lift_dense.bed.gz), and described (https://egg2.wustl.edu/roadmap/web_portal/chr_state_learning.html) as follows:

| State | Mnemonic | Description | Color name | Color code |
|---|---|---|---|---|
| 1 | TssA | Active TSS | Red | 255,0,0 |
| 2 | TssAFlnk | Flanking Active TSS | Orange Red | 255,69,0 |
| 3 | TxFlnk | Transcr. at gene 5' and 3' | LimeGreen | 50,205,50 |
| 4 | Tx | Strong transcription | Green | 0,128,0 |
| 5 | TxWk | Weak transcription | DarkGreen | 0,100,0 |
| 6 | EnhG | Genic enhancers | GreenYellow | 194,225,5 |
| 7 | Enh | Enhancers | Yellow | 255,255,0 |
| 8 | ZNF/Rpts | ZNF genes & repeats | Medium Aquamarine | 102,205,170 |
| 9 | Het | Heterochromatin | PaleTurquoise | 138,145,208 |
| 10 | TssBiv | Bivalent/Poised TSS | IndianRed | 205,92,92 |
| 11 | BivFlnk | Flanking Bivalent TSS/Enh | DarkSalmon | 233,150,122 |
| 12 | EnhBiv | Bivalent Enhancer | DarkKhaki | 189,183,107 |
| 13 | ReprPC | Repressed PolyComb | Silver | 128,128,128 |
| 14 | ReprPCWk | Weak Repressed PolyComb | Gainsboro | 192,192,192 |
| 15 | Quies | Quiescent/Low | White | 255,255,255 |

### Gene ontology analysis

In this study, gene ontology analysis is conducted using the findGO.pl script of the HOMER suite. The findGO.pl script requires a list of gene names as input and subsequently outputs gene-enriched categories, such as biological processes, molecular functions, cellular components, KEGG pathways, and protein interactions, among others. The gene list can be obtained directly from the data (e.g., MS results) or from genomic regions (e.g., peaks, loop anchors) that have been annotated to gene bodies using the annotatePeaks.pl script of HOMER followed by filtering out intergenic annotations.

## Aggregating distances from RNA-associated peaks to CTCF and RNAPII binding peaks

The top 5 percent of peaks from each RDD library were selected to calculate the distance from their summit to the CTCF and RNAPII ChIA-PET peak summit using closest function of BEDTools. These distances were then aggregated and visualized as density plots.

## Comparing chromatin accessibility of RNA and protein factors

RPM of uniquely mapped read counts of the top 5 percentile peaks of CTCF and RNAPII ChIA-PET, as well as *EBERs*, *miR-BART*, *v-snoRNA1*, and *MALAT1* RDD libraries, were illustrated using box plot, of which the middle line denotes median; box boundary encloses interquartile range (IQR); and whiskers denote 1.5× IQR. Two-sided multi-comparison Kruskal–Wallis test for genes is performed among box plots using stat_compare_means() function of the R package ggpubr (v0.5.0).

## Dynamic of gene expression between *EBERs* knockdown and non-treatment control

The gene expression levels from RNA-seq were obtained from RSEM pipeline, and the fold change ratio was calculated by comparing the FPKM of *EBERs* knockdown RNA-seq with that of the non-treatment control RNA-seq. A combination of box, violin, and jitter plots was used to visualize the data, with red dots denoting those genes that exhibit a fold change of ≥2.

## Dynamic of chromatin interaction between *EBERs* knockdown and non-treatment control

In a typical region includes *HLA-DMS* and *HLA-DMA* were selected with a zoomed-in vision, we calculated PET count sum of loops inside using *EBERs* knockdown and non-treatment control RNAPII ChIA-PET data, output as box and violin plots. Additionally, we selected regions (~150 kb) around 17 genes (*AC009403.1*, *RN7SL211P*, *INPP5F*, *KDM6B*, *HBE1*, *SNHG25*, *TEX13D*, *CDH11*, *CALN1*, *CLEC2D*, *ZNF260*, *RN7SL842P*, *GINS4*, *ZFP1*, *BTNL9*, *MCUB*, *ITGA3*) of interest, along with *MALAT1* as the control. The cumulated PET count sum of loops within each region was then depicted as box and violin plots in RPM. Two-sided Wilcoxon tests were performed between the data pairs.

## *EBERs*-targeted genes associated interaction

To investigate the interaction among the 231 *EBERs*-targeted genes, we defined the gene body region as spanning from 2 kb upstream of the transcription start site (TSS) to the transcription termination site (TTS). We then overlapped these gene body regions with the anchors of loops from ChIA-PET data against RNAPII or NCL in *EBERs* knockdown (KD) and negative control (NC) GM12878 cells, as well as *NCL*-NC and *NCL*-KD GM12878 cells. The PET counts for these selected loops were calculated in terms of RPM and visualized using violin plots to illustrate distribution patterns. A two-sided Wilcoxon test was performed to assess statistical differences between the groups, providing insight into the effects of *EBERs* and knockdowns on loop interactions.

## Chromatin accessibility at NCL binding sites in *EBERs*-NC and *EBERs*-KD cells

To investigate chromatin accessibility at NCL binding sites in *EBERs*-NC and *EBERs*-KD cells, we selected peaks from NCL-enriched ChIP-seq data of *EBERs*-NC and *EBERs*-KD cells. Uniquely mapped reads from ATAC-seq of *EBERs*-NC and *EBERs*-KD cells were aligned to these peaks to assess accessibility. The final read counts were normalized to RPM and visualized using box plots. To evaluate statistical differences in chromatin accessibility between the conditions, two-sided Wilcoxon tests were conducted.

## Mass analyzing *EBERs*-enriched proteins

The raw mass spectrometry files of *EBERs*-associated proteins were analyzed and searched against target protein references, which were downloaded from the UniProt database (The UniProt Consortium, 2017) (https://www.uniprot.org), specific to the human and EBV samples using MaxQuant (Cox and Mann, 2008) (v.1.6.2.10). The following parameters were set: carbamidomethylation (C) was considered a fixed protein modification, while oxidation (M) and acetylation (protein N-term) were considered variable modifications. The enzyme specificity was set to trypsin, allowing for a maximum of 2 missed cleavages. The precursor ion mass tolerance was set to 20 ppm, and the MS/MS tolerance was also set to 20 ppm. Only peptides identified with high confidence were selected for downstream protein identification analysis. As a result, a protein intensity vs. protein score scatter plot was generated to show *EBERs* interacting proteins.

## Statistics and reproducibility

Statistical analyses were conducted using R (v4.4.0). For paired data, two-sided Wilcoxon or *t* tests were utilized, and for multiple comparisons, a two-sided Kruskal–Wallis test was employed. Exact sample sizes and *P* values are detailed in the figures, figure legends, or the main text, including the "Methods". $P \leq 0.05$ was considered statistically significant. Pearson correlation coefficients were used to identify correlations between two samples. In box plots, the bounds represent the 25 and 75th percentiles, with the median indicated by a line within the box. The whiskers extend to 1.5 times the interquartile range (IQR).

# Data availability

Cell lines and other materials are available upon request. The sequencing raw data and associated results reported in this paper are deposited in the Gene Expression Omnibus (GEO) with accession code GSE281522. Data from public databases used in this article are listed in Table EV1. All scripts used to generate the source data figures have been deposited on Zenodo (https://doi.org/10.5281/zenodo.11215425). Any additional information required to reanalyze the data reported in this paper is available from the corresponding author upon request.

The source data of this paper are collected in the following database record: biostudies:S-SCDT-10_1038-S44318-025-00466-5.

## Peer review information

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

## Acknowledgements

We thank Jinxin Bei, Yuxin Lin, Lin Feng, Guoliang Li, and Zhonghui Tang for helpful discussions. Fuxing Zeng is an investigator of SUSTech Institute for Biological Electron Microscopy. This work was supported by the National Key R&D Program of China (2022YFC3400400, 2022YFC3400401), the National Natural Science Foundation of China (32170644, 62375116, 32250710678), the Shenzhen Science and Technology Program (JCYJ20220530115211026, JCYJ20220818100416036, KQTD20200820113012029), and the Guangdong Basic and Applied Basic Research Foundation (2023A1515011231, 2022A1515011174).

## Author contributions

**Simon Zhongyuan Tian**: Conceptualization; Data curation; Software; Formal analysis; Validation; Investigation; Visualization; Methodology; Writing—original draft; Writing—review and editing. **Yang Yang**: Validation; Methodology; Writing—original draft. **Duo Ning**: Validation; Methodology. **Ting Yu**: Validation; Writing—original draft. **Tong Gao**: Methodology. **Yuqing Deng**: Data curation. **Ke Fang**: Funding acquisition; Validation; Writing—original draft. **Yewen Xu**: Validation. **Kai Jing**: Data curation. **Guangyu Huang**: Validation. **Gengzhan Chen**: Validation. **Pengfei Yin**: Data curation. **Yiming Li**: Resources; Supervision; Funding acquisition; Validation; Investigation; Writing—original draft; Writing—review and editing. **Fuxing Zeng**: Resources; Validation; Investigation; Writing—original draft; Writing—review and editing. **Ruilin Tian**: Resources; Supervision; Validation; Investigation; Methodology; Writing—original draft; Writing—review and editing. **Meizhen Zheng**: Conceptualization; Resources; Data curation; Software; Formal analysis; Supervision; Funding acquisition; Validation; Investigation; Visualization; Methodology; Writing—original draft; Project administration; Writing—review and editing.

Source data underlying figure panels in this paper may have individual authorship assigned. Where available, figure panel/source data authorship is listed in the following database record: biostudies:S-SCDT-10_1038-S44318-025-00466-5.

## Disclosure and competing interests statement

The authors declare no competing interests.

# Expanded View Figures

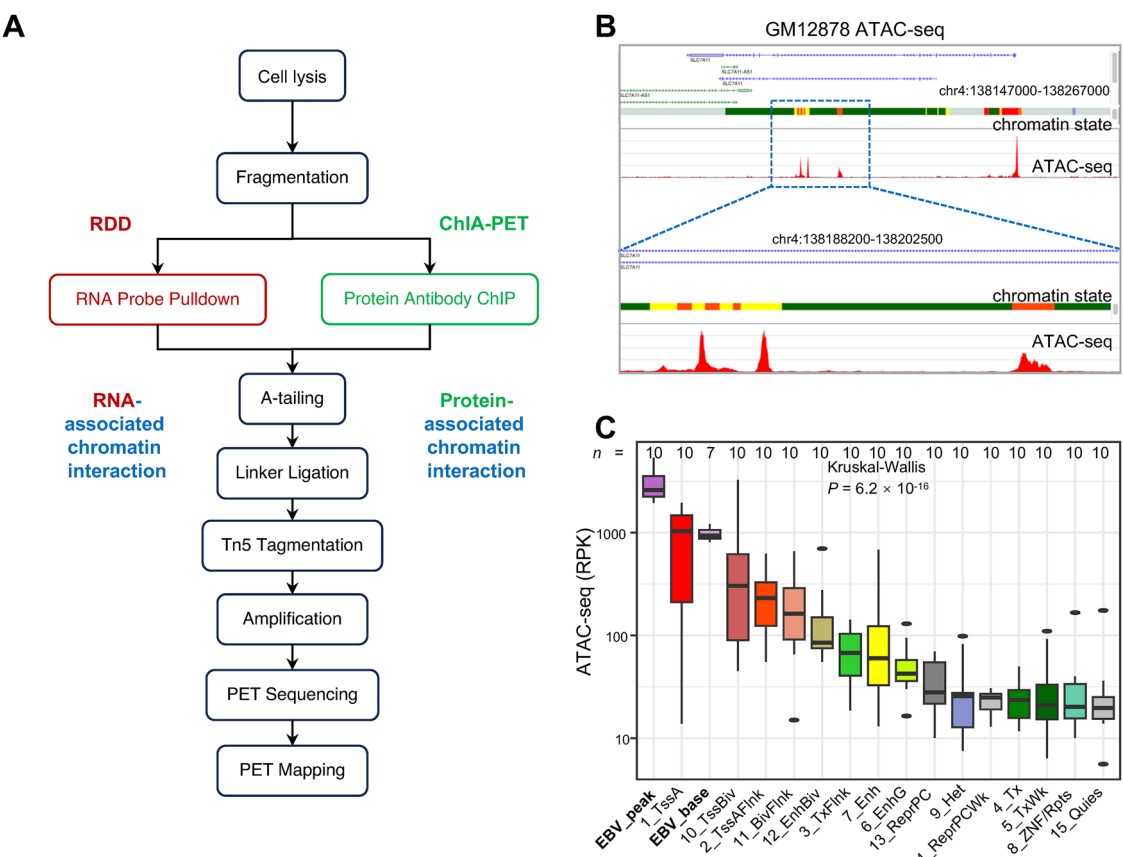

**Figure EV1.  Chromatin interactions and chromatin accessibility.**

(A) Flowchart of chromatin interactions associated with protein factors detected by the ChIA-PET method and those associated with ncRNA factors from the RDD method. (B) BASIC Browser visualization of distinctive chromatin accessible and inaccessible regions in host human GM12878 cells from ATAC-seq data. (C) Box plots illustrating the chromatin accessibility across the 15 human chromatin states, as well as the EBV ATAC-seq peak region (EBV_peak) and non-peak region (EBV_base) indicating the baseline level. Chromatin state- 1: Active Transcription Start Site (TSS), 2: Flanking Active TSS, 3: Transcription at gene 5' and 3' ends, 4: Strong transcription, 5: Weak transcription, 6: Genic enhancers, 7: Enhancers, 8: ZNF genes and repeats, 9: Heterochromatin, 10: Bivalent/Poised TSS, 11: Flanking Bivalent TSS/ Enhancers, 12: Bivalent Enhancer, 13: Repressed PolyComb, 14: Weak Repressed PolyComb, 15: Quiescent/Low. Kruskal–Wallis tests are performed, $P = 6.2 \times 10^{-14}$. Each box plot highlights the median (inside line), the 25–75th percentiles (box), and the minima/maxima values within 1.5× the interquartile range (IQR) of the box (whiskers).

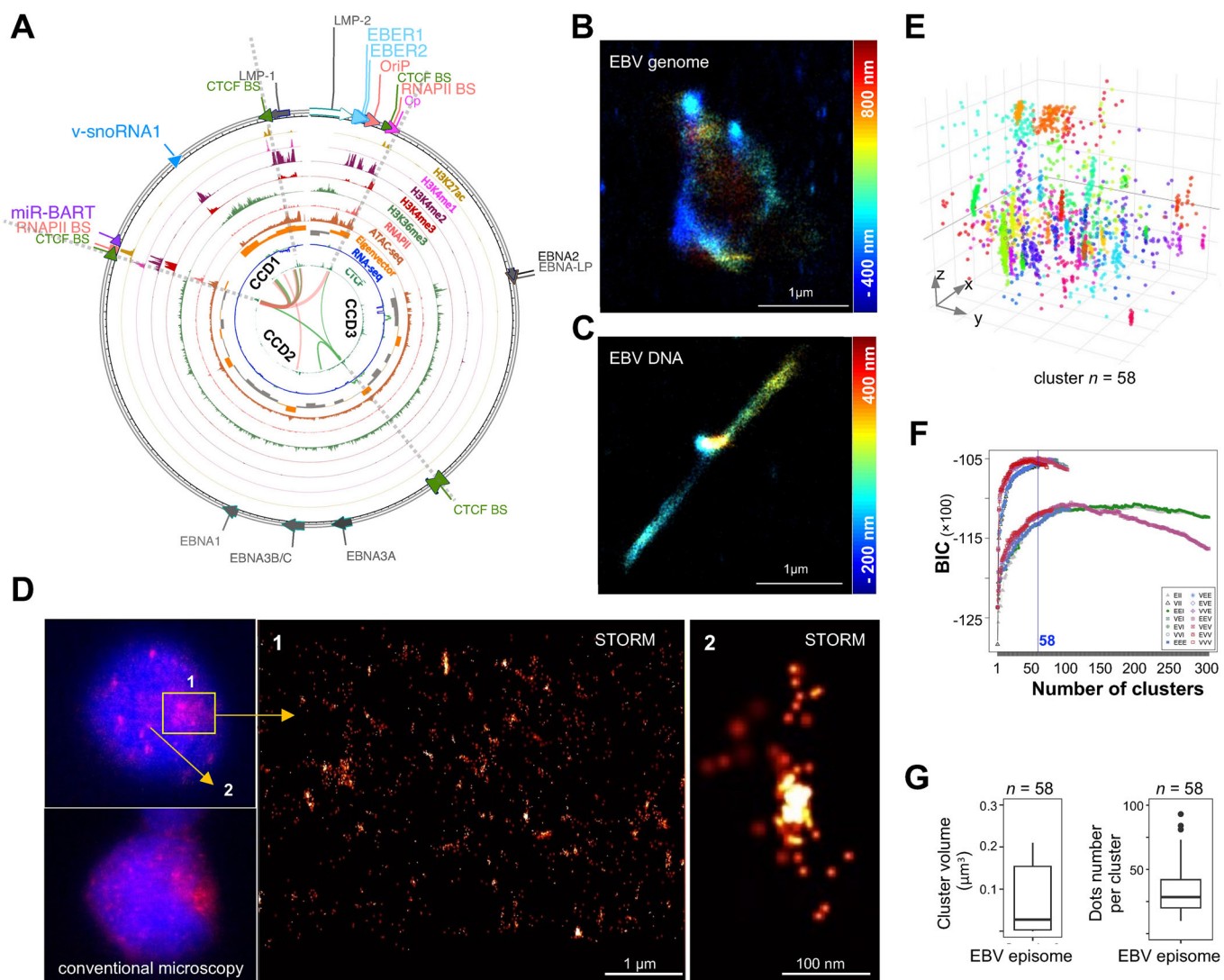

**Figure EV2. EBV 3D structure.**

(A) Circos plot displays CTCF- and RNAPII-mediated EBV chromatin interactions (in the innermost layer, with RNAPII-mediated loops in red and CTCF-mediated loops in green), along with histone marker signals. The outermost layer represents the EBV genome, with arrows indicating gene directions, and "CTCF BS" denoting CTCF-binding sites along with motif orientation. (B) Example images of BALM on EBV genome from B95-8 cells. The color scale corresponds to the z range. (C) Example images of BALM on EBV DNA. (D) Example images of conventional microscopy from DNA-FISH (red) of EBV and DAPI (blue) performed in GM12878 cells, along with STORM images showing a zoomed-in view of the EBV cloud highlighted by a yellow square labeled as '1' in the middle and EBV foci labeled as '2' on the right. Here, the STORM images presented the high-resolution structure from different loci of cells shown in Fig. 1G. (E) 3D structure of EBV clusters in the zoomed-in square in (D, middle), with each cluster corresponding to an EBV episome represented by a different color. (F) Bayesian information criterion (BIC) prediction method in the GMM (Gaussian mixture models) clustering algorithm for EBV clustering. (G) Box plots showing the density of EBV episome sizes (left panel) and the number of dots in individual EBV clusters (episomes, right panel). Each box plot highlights the median (inside line), the 25–75th percentiles (box), and the minima/maxima values within 1.5× the interquartile range (IQR) of the box (whiskers).

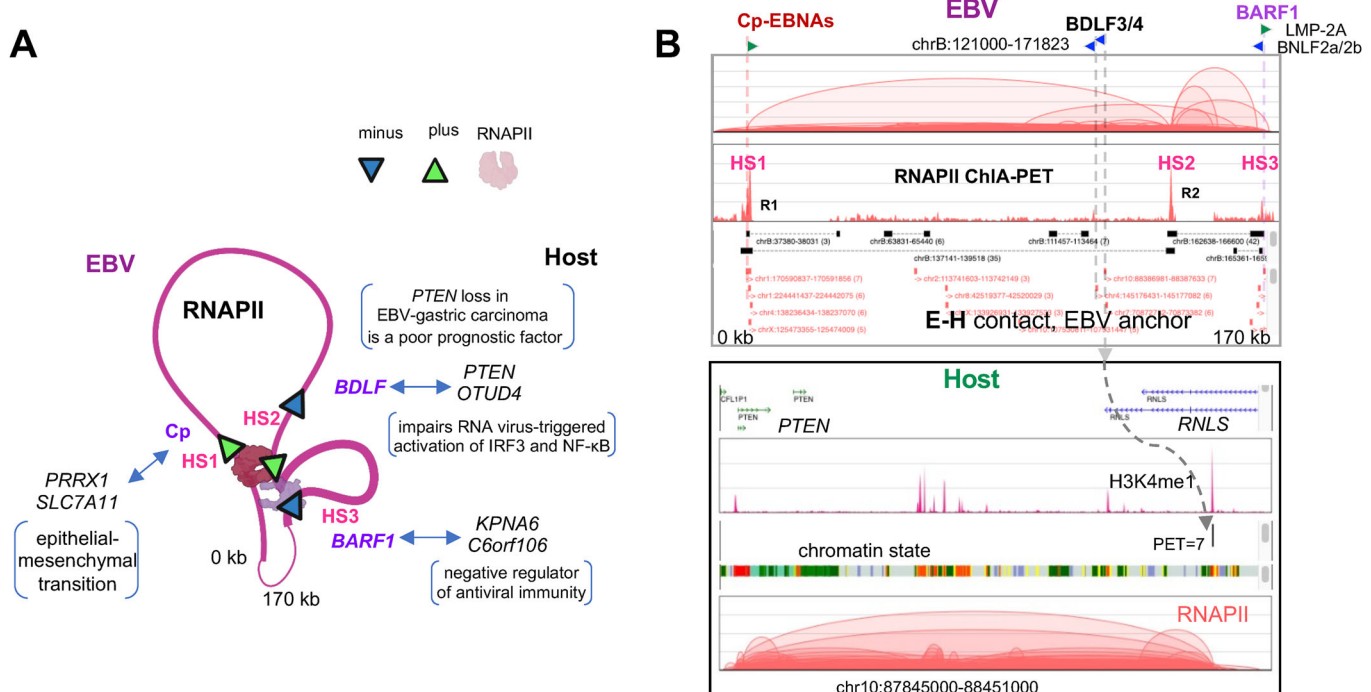

**Figure EV3.  Protein-associated E–H interaction.**

(A) Diagram shows RNAPII-mediated chromatin interaction on EBV side. The arrow line represents the contact genes between EBV and the host; the parentheses indicate the function of the gene. Arrowhead presents gene transcribed orientation. (B) RNAPII-mediated chromatin interactions between EBV chromatin and host chromatin (E–H). EBV genes were highlighted on the top, followed by tracks for RNAPII loop, binding peak, and EBV interacting anchor. The dashed line points to the contact region of the host around *PTEN* gene, with H3K4me1 peak denoting the active enhancer region, alongside chromatin state and RNAPII-mediated loops.

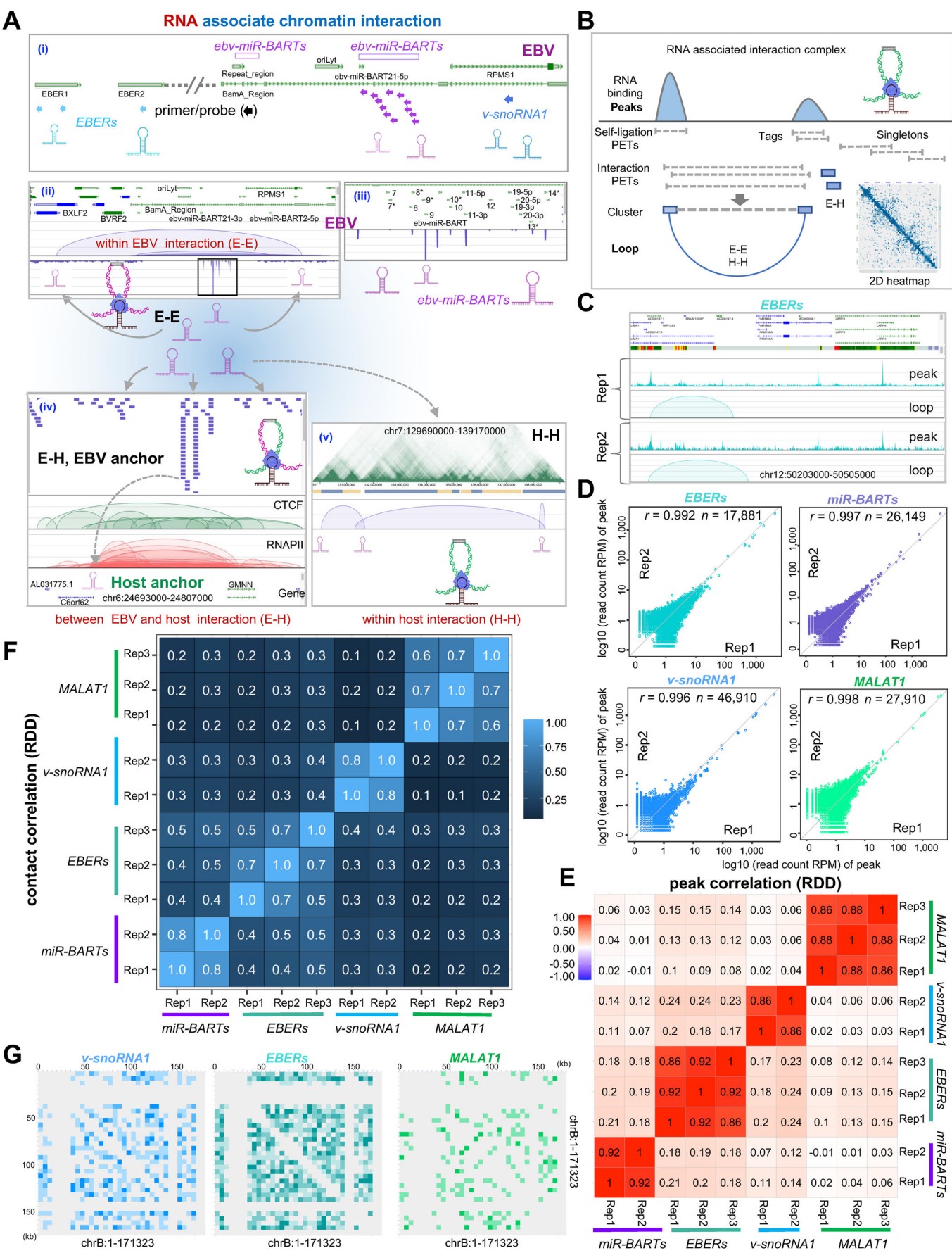

**Figure EV4.  The RDD method detects EBV ncRNA-associated chromatin interactions.**

(A) Positions of primers/probes of ncRNAs along EBV genome that are used in this study (i), Arrowheads indicate the primer/probes direction, Hairpin RNA indicates transcripts of target RNAs (Cyan represents *EBERs*, Purple represents *miR-BARTs*, Light blue represents *v-snoRNA1*); (ii) Viral RNA-associated EBV chromatin DNA–DNA interaction loops along with EBV genome reference (EBV-to-EBV: E–E), square highlights target RNA origin transcript loci that were zoomed in (iii); (iv) Viral RNA facilitates the chromatin DNA–DNA interactions between EBV and host (E–H). Purple bar represents the contact anchor at EBV chromatin, dash line with arrowhead directs to the other contact anchor at host chromatin annotated with CTCF (green) and RNAPII (red) mediated chromatin loops. (v) Viral RNA-associated host chromatin DNA–DNA interaction loops (host-to-host: H–H) along with 2D heatmap contacts. (B) Graphic of RDD mapping properties including binding peaks piled up from all tags, self-ligation PETs (two ends of tags are from the same DNA fragment), interaction PETs (two ends of tags are from different DNA fragments), singletons (individual PETs), clusters or loops (overlapped PETs), interacting anchor between E–H or inter-chromosomes, and visualization of pairwise contacts in form of loops or contact heatmaps. (C) BASIC Browser visualization of EBV ncRNA *EBERs*-associated chromatin loops and peaks in two RDD replicates. (D–F) Reproducibility analyses of RDD data were conducted using different methods: (D) Scatter plots of RDD peaks between replicates. The *r* value represents the Pearson correlation coefficient. (E) the Spearman correlation coefficient of RDD peaks among all replicates, and (F) RDD contact data using HiCRep. The stratum-adjusted correlation coefficient (SCC) was computed for pairs of RDD libraries, with the SCC value displayed. "Rep" denotes replicate. (G) 2D heatmaps represent the EBV ncRNA-associated chromatin interactions within the EBV genome (E–E).

## chromatin interaction (E-H)

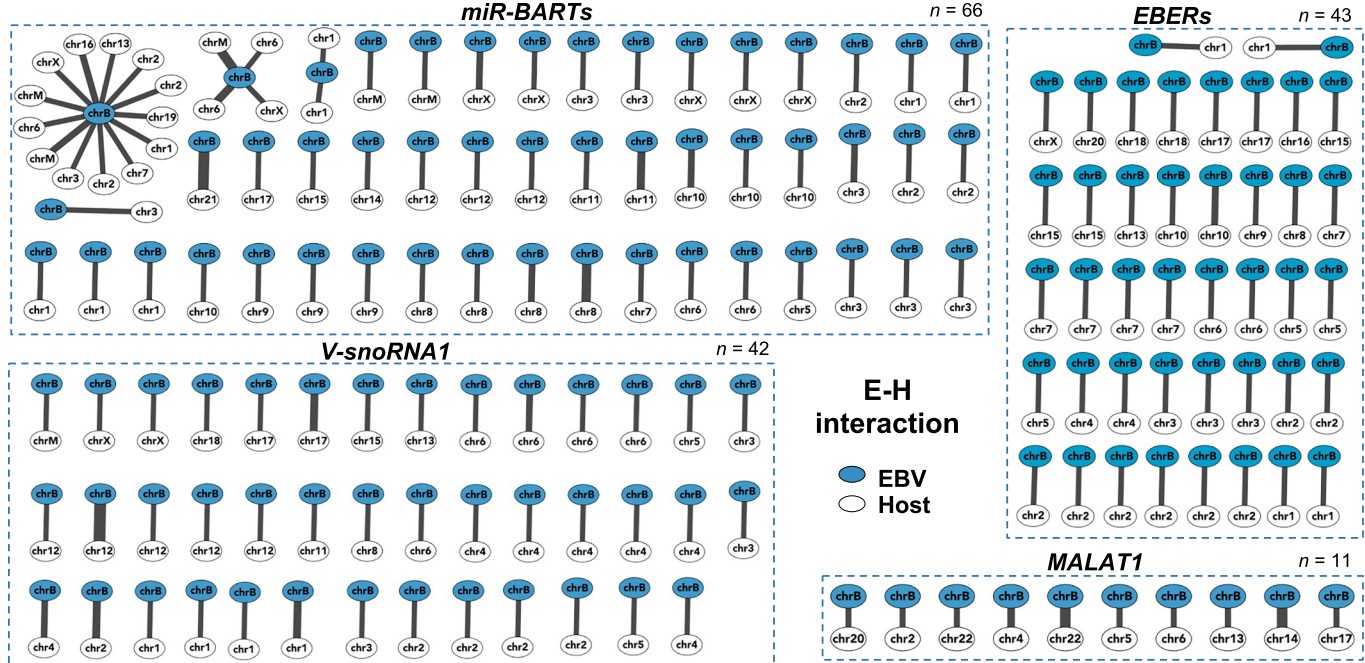

**Figure EV5. EBV and host chromatin interaction network (E–H).**

Cytoscape illustration of the contact network (PET ≥ 2) between EBV and the host (E–H) associated with ncRNA factors (*EBERs, miR-BARTs, v-snoRNA1,* and *MALAT1*). Blue ellipses represent EBV, labeled as "chrB," while white ellipses represent the host, labeled as "chr#." Most interactions involve a one-to-one correspondence between a "chrB" and a "chr#," except for *miR-BARTs*, which show a single contact where one "chrB" corresponds to multiple "chr#" instances. This indicates that the EBV episome primarily contacts the host chromatin in a one-to-one manner.

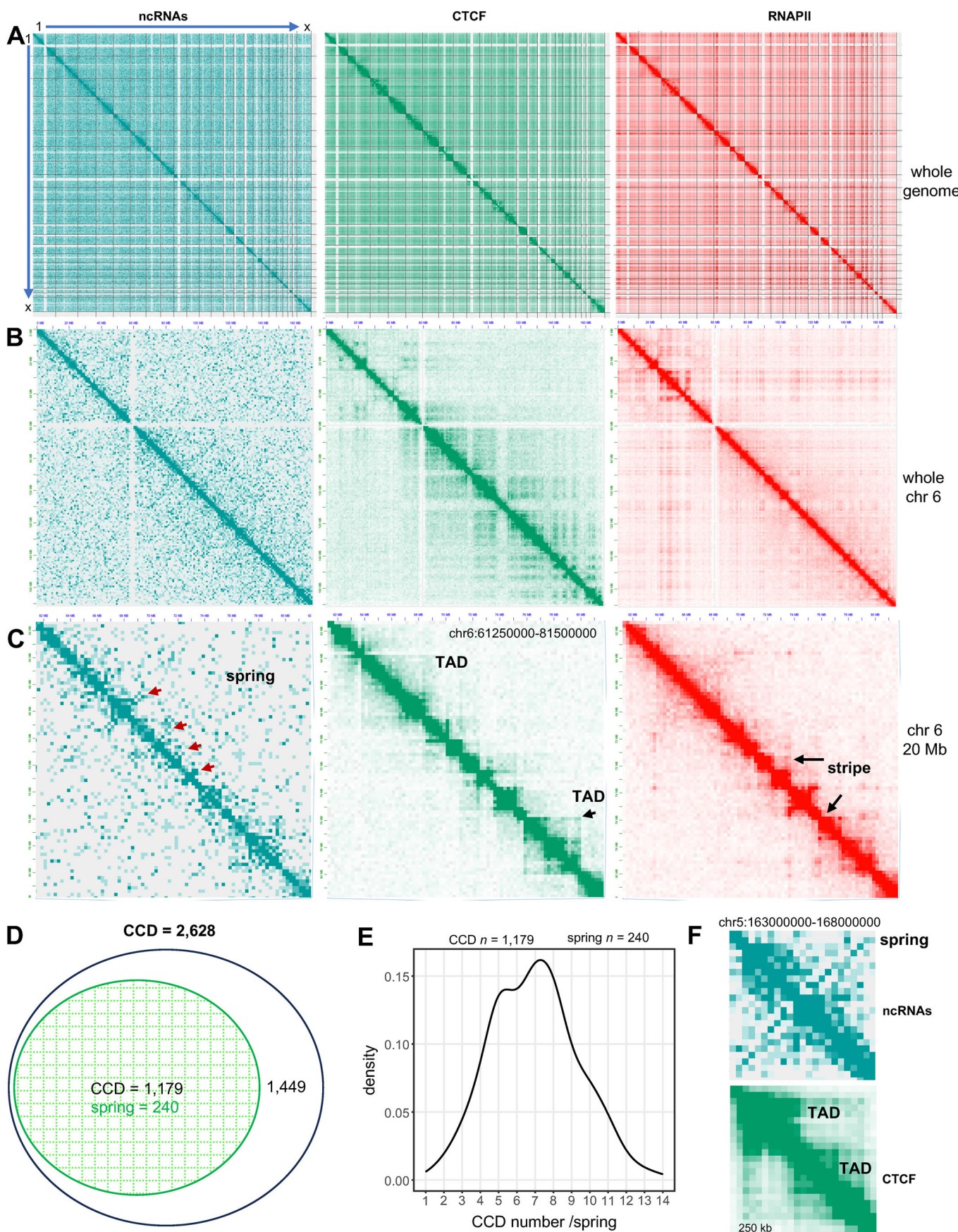

◄ **Figure EV6. Contact heatmaps present chromatin contact profiling associated with ncRNAs or proteins.**

(A–C) Contact heatmaps cover the whole genome (A), whole chromosome level (B), and a zoomed-in genomic region spanning 20 Mb at 100 kb resolution (C), with chromatin structures of 'spring', 'TAD' and 'stripe' marked. (D) Venn diagram illustrates the number of CCDs covered by the springs. (E) Density plot displays the distribution of CCD numbers within individual springs. (F) Representative examples highlighting the relationship between springs and CCDs/TADs.

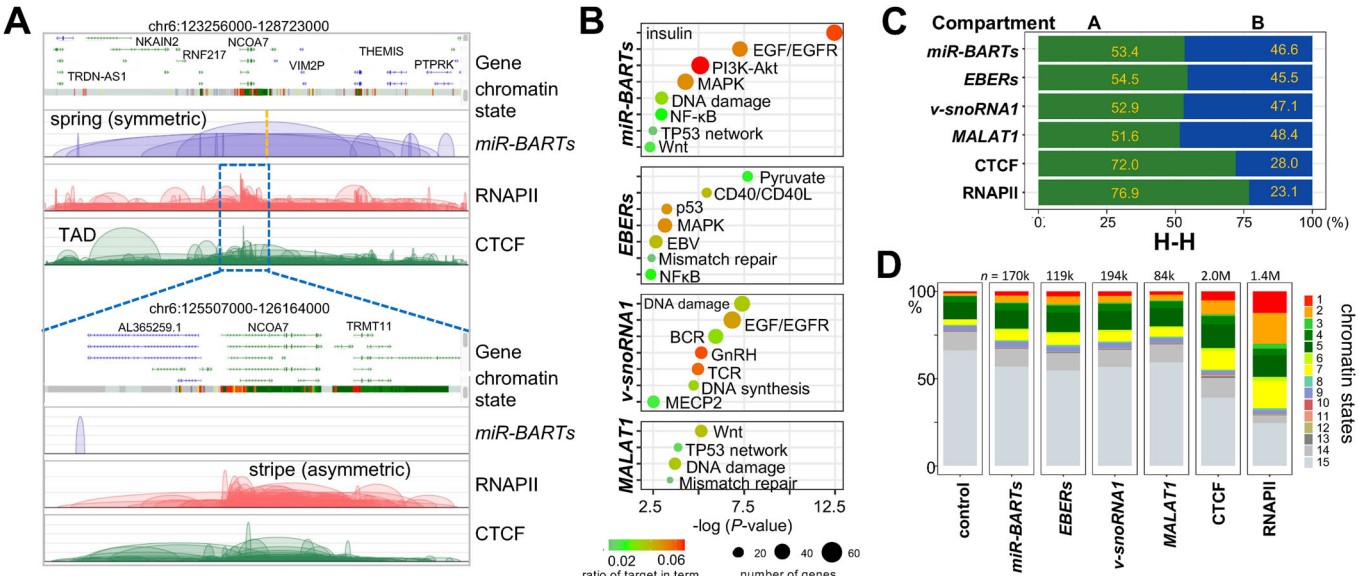

**Figure EV7. EBV ncRNAs-associated chromatin interaction within host genome (H–H).**

(A) BASIC Browser visualization of chromatin loop profiling involving ncRNA *BARTs*-associated and RNAPII/CTCF-mediated interactions. Yellow dashed lines highlight the ncRNA-associated chromatin symmetric midpoint, and dashed boxes provide a zoomed-in view of the RNAPII/CTCF-mediated chromatin loop structure. (B) GO term enrichment analysis of host chromatin loop-anchored genes targeted by ncRNAs. (C) Histogram shows percentage of chromatin loop-anchored targets in the A/B compartment. (D) Histogram shows the distribution of 15 chromatin states (see "Methods") for ncRNA-targeted host chromatin loops (H–H).

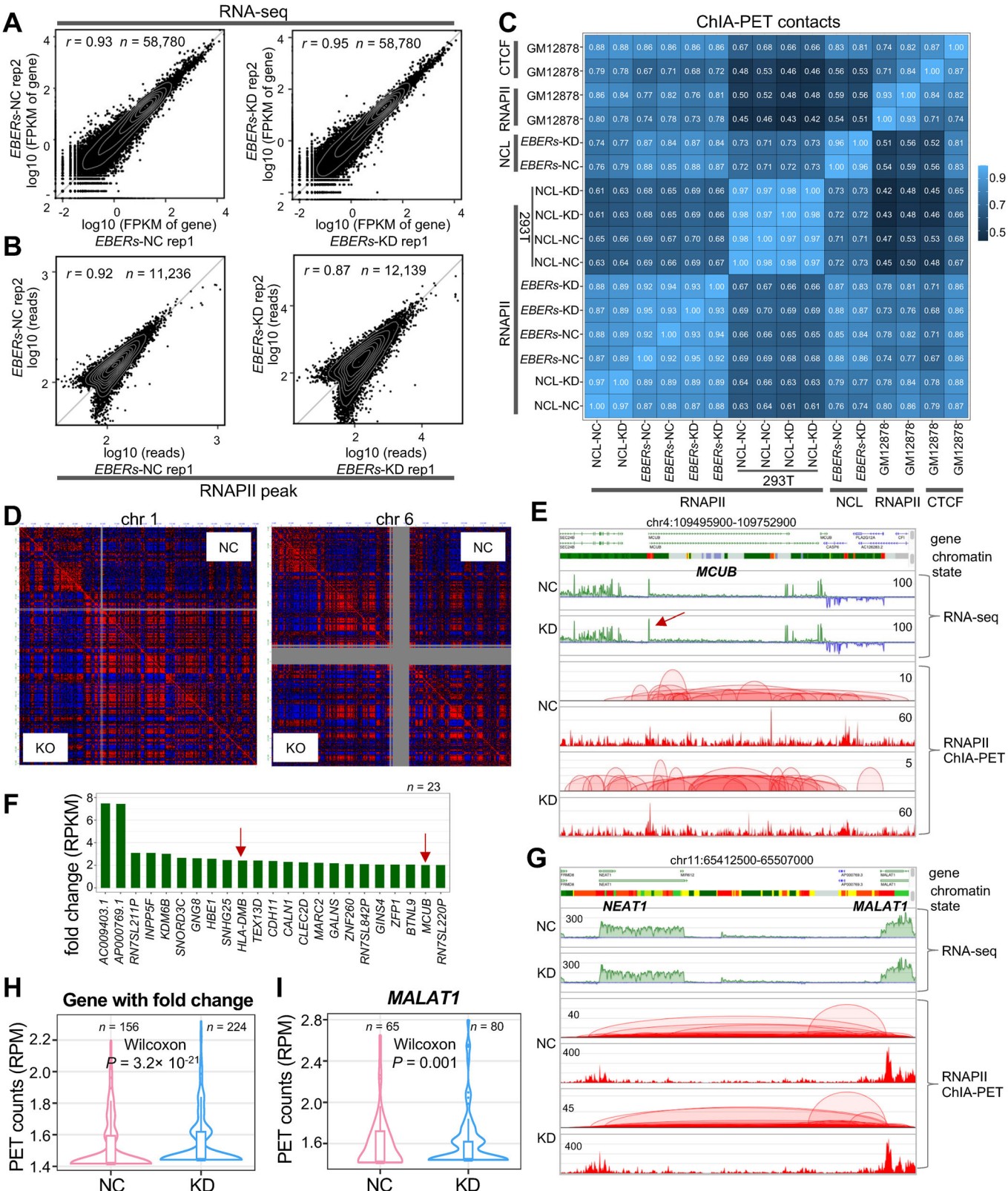

◀

**Figure EV8. EBV ncRNA *EBERs* repress host gene expression through RNAPII-associated chromatin loops.**

(A, B) Reproducibility analyses of RNA-seq data (A) and RNAPII peak data (B) in *EBERs*-NC and *EBERs*-KD. The *r* value represents the Pearson correlation coefficient. "Rep" denotes replicate. (C) Reproducibility analyses of ChIA-PET contact data using HiCRep. The stratum-adjusted correlation coefficient (SCC) was computed for pairs of RDD libraries, with the SCC value displayed. (D) Observed Pearson Heatmaps for chromosome 1 and 6. The correlation matrix depicts the strength of correlation [from −1 (blue) to +1 (red)] between the intrachromosomal interaction profiles of every pair of 250-kb loci spanning the entire chromosome. The distinctive plaid pattern indicates the presence of A/B compartments within the chromosome. (E) Display of RNA expression levels and RNAPII-associated chromatin loops around the *MCUB* gene in *EBERs*-NC cells and *EBERs*-KD cells. (F) Bars presents the genes with more than a two-fold change expression when comparing *EBERs*-KD cells to *EBERs*-NC cells. Arrows indicate the *MCUB* and *HLA-DMB* genes, visualized with the BASIC Browser in (E) and Fig. 6D, respectively. (G) Display of RNA expression levels and RNAPII-associated chromatin loops around the *MALAT1* and *NEAT1* gene in *EBERs*-NC cells and *EBERs*-KD cells. (H) Violin-Box plots present the chromatin interaction PET counts from RNAPII ChIA-PET associated with the differentiated 231 genes in *EBERs*-KD cells versus *EBERs*-NC cells. Two-sided Wilcoxon tests are performed, $P = 3.2 \times 10^{-21}$. (I) Violin-Box plots present the chromatin interaction PET counts from RNAPII ChIA-PET in *EBERs*-NC cells and *EBERs*-KD cells at the genomic region of the *MALAT1* gene (G). Two-sided Wilcoxon tests are performed, $P = 0.001$. Each box plot highlights the median (inside line), the 25–75th percentiles (box), and the minima/maxima values within 1.5× the interquartile range (IQR) of the box (whiskers).

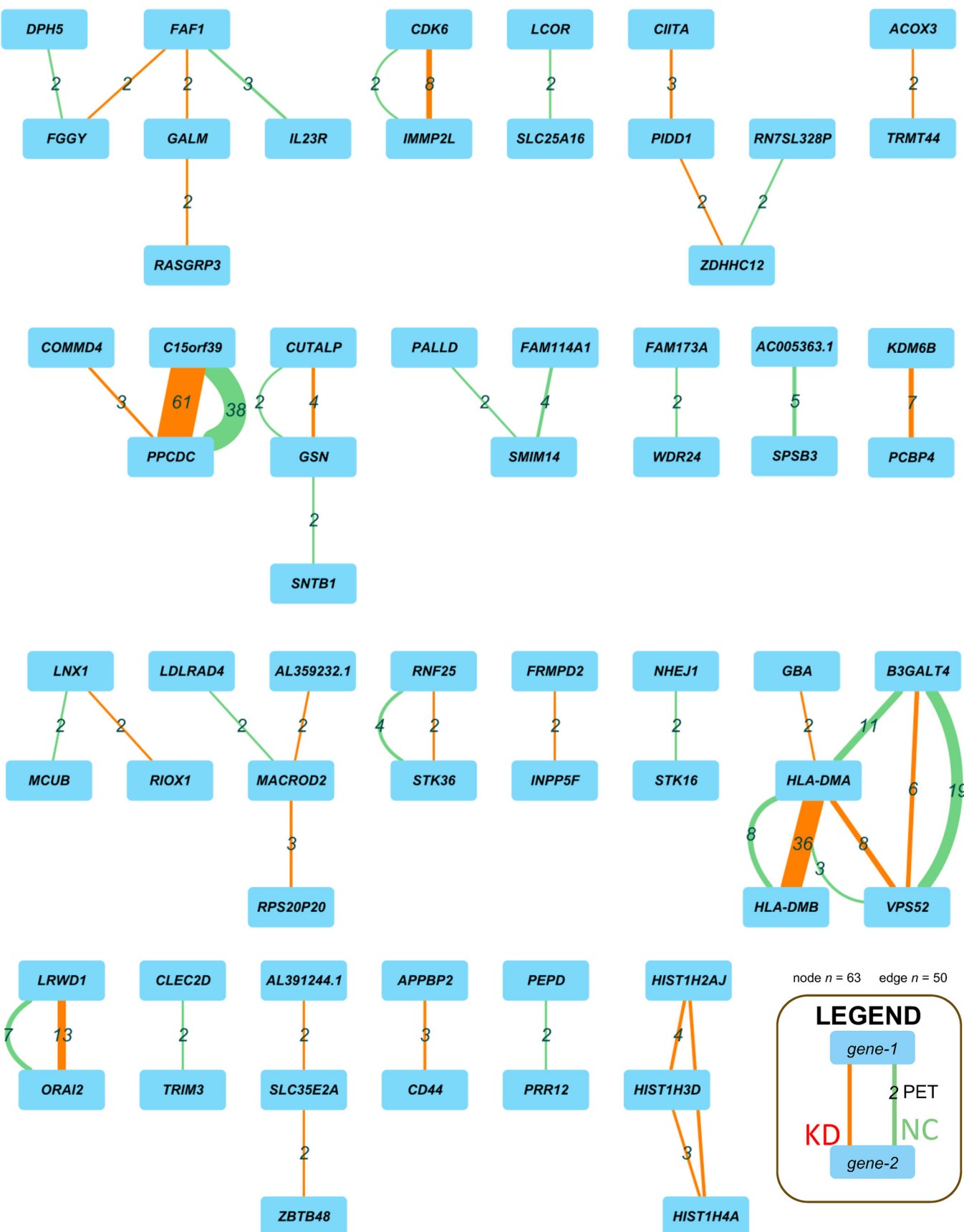

◀ **Figure EV9. The connections among genes repressed by *EBERs*.**

Cytoscape illustrates the interaction network among the 231 *EBERs*-targeted genes captured by RNAPII ChIA-PET in *EBERs*-NC and *EBERs*-KD cells. The thickness of the edge reflects the interaction frequency. Orange color indicates RNAPII interaction loops in the *EBERs*-KD cells, while green color indicates RNAPII interaction loops in the *EBERs*-NC cells. The numbers represent PET counts, indicating the interaction frequency.

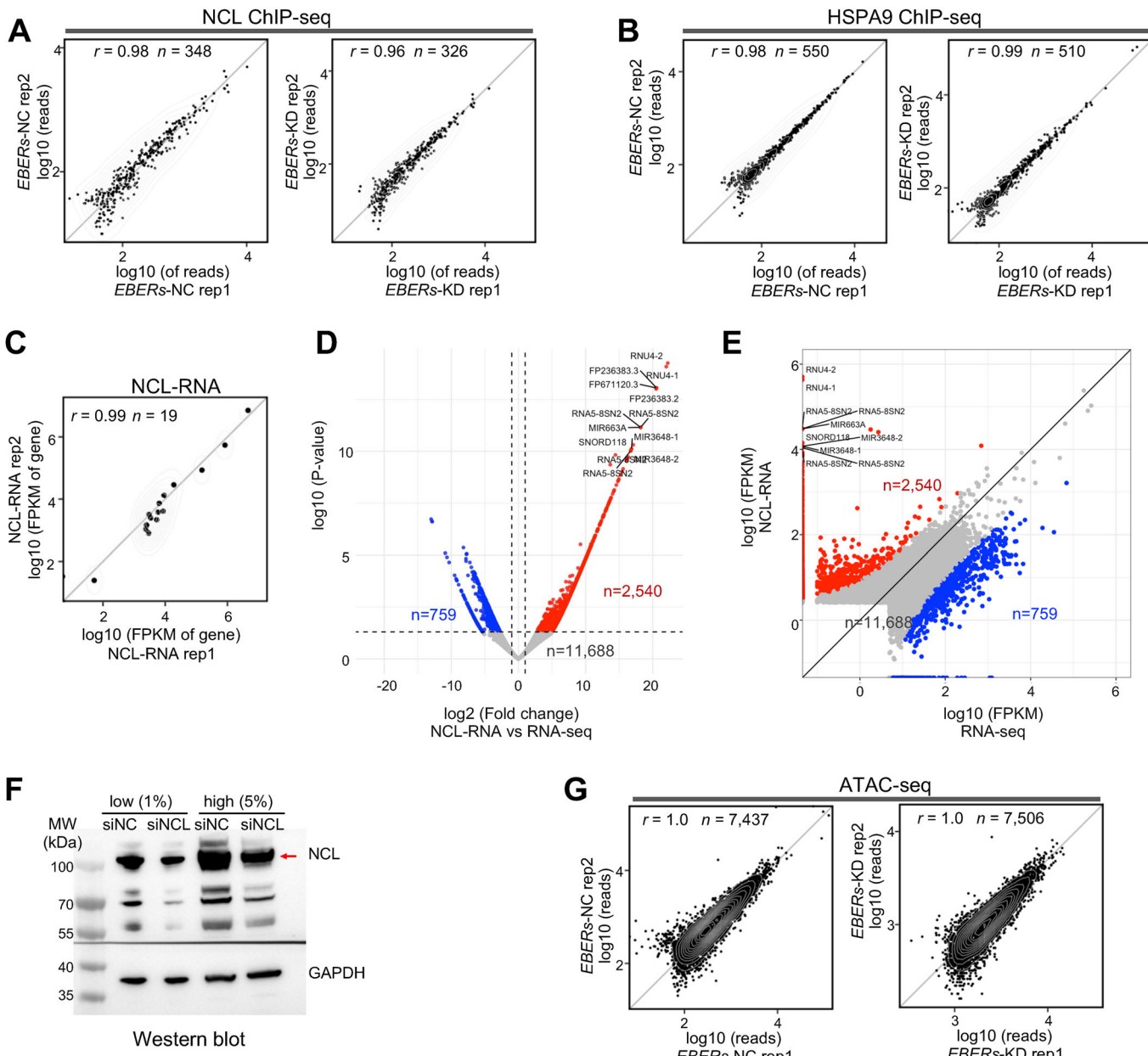

**Figure EV10.  Reproducibility analyses.**

(A–C) Scatter plots demonstrating the reproducibility of NCL ChIP-seq (A), HSPA9 ChIP-seq (B), and NCL-interacted chromatin RNA assay captured EBV gene expression associated with NCL protein (C) in *EBERs*-NC and *EBERs*-KD. (D) Volcano plot illustrating the enrichment of NCL-bound RNAs identified by the NCL-RNA assay compared to RNA-seq. The x-axis represents the log2 fold change (NCL-RNA vs. RNA-seq), while the y axis represents the -log10 P-value from the likelihood ratio test. RNAs enriched in the NCL-RNA data are shown on the right side of the plot, with significantly enriched RNAs highlighted as red points above the horizontal dashed line (P < 0.05). The bracket marks the region containing NCL-enriched RNAs. (E) Scatter plot showing the gene expression levels (FPKM) of NCL-RNA and RNA-seq, corresponding to (D). The scatter plot supplements the volcano plot by providing the original expression levels, which are not visible in the volcano plot. *n* represents the number of genes. (F) Western blot showing the knockdown efficiency of NCL using siRNA, with GAPDH as the control. (G) Scatter plots illustrating the reproducibility of ATAC-seq in *EBERs*-NC and *EBERs*-KD. "Rep" denotes replicate.

