## [Peer Review File · The EMBO Journal]

Landscape of the Epstein-Barr virus-host chromatin interactome and gene regulation

Simon Zhongyuan Tian, Yang Yang, Duo Ning, Ting Yu, Tong Gao, Yuqing Deng, Ke Fang, Yewen Xu, Kai Jing, Guangyu Huang, Gengzhan Chen, Pengfei Yin, Yiming Li, Fuxing Zeng, Ruilin Tian and Meizhen Zheng

Corresponding authors: Meizhen Zheng (zhengmz@sustech.edu.cn) , Yiming Li (Liym2019@sustech.edu.cn), Simon Zhongyuan Tian (tianzy3@sustech.edu.cn), Ruilin Tian (tianrl@sustech.edu.cn), Fuxing Zeng (zengfx@sustech.edu.cn)

Review Timeline:

Submission Date:	29th Jan 25
Editorial Decision:	2nd Apr 25
Revision Received:	6th Apr 25
Editorial Decision:	2nd May 25
Revision Received:	5th May 25
Accepted:	7th May 25

Editor: Ieva Gailite

Transaction Report:

(Note: Please note that the manuscript was previously reviewed at another journal and the reports were taken into account in the decision making process at The EMBO Journal. With the exception of the correction of typographical or spelling errors that could be a source of ambiguity, letters and reports are not edited. Depending on transfer agreements, referee reports obtained elsewhere may or may not be included in this compilation. Referee reports are anonymous unless the Referee chooses to sign their reports.)

We appreciate all the reviewers' acknowledgment of the significant improvements in the revised manuscript. We have carefully addressed their remaining concerns, and the detailed responses are provided below (in blue text). The corresponding revisions in the manuscript are highlighted in **red**.

Reviewer #1 (Remarks to the Author):

The authors have performed new experiments and addressed most of my previous concerns. However, there are several remaining concerns, especially for the NCL CLIP-seq data to characterize its interaction with EBERs RNA.

Thank you for your thorough review and for acknowledging our efforts to address your previous concerns. We deeply value your constructive feedback, which has been instrumental in improving our manuscript. Here, we would like to specifically address the remaining issues you highlighted, particularly regarding the NCL CLIP-seq data and its interaction with EBERs RNA.

Major points:

1. It is unexpected to see the conclusions that “Experiments with individual RRM domains (RRM1, RRM2, RRM3, RRM4) showed no binding to EBERs ncRNA. However, binding was observed with the combined RRM2-4 domains.” Is this unique to EBERs RNA, or the same to NCL’s other target RNAs? What is the unique feature of NCL compared with other RNA-binding proteins containing multiple RRMs?

We greatly appreciate your thoughtful comments and questions regarding our findings on NCL binding to EBERs RNA. Your insights have allowed us to further elucidate and refine our conclusions. We address your specific questions as follows:

1) Is this binding behavior unique to EBERs RNA, or is it similar for NCL's other target RNAs?

Our results are consistent with findings from studies analyzing NCL’s interaction with other RNAs, such as telomeric repeat-containing RNA (TERRA). For example, in a recent study (Biochemistry 62, 1249–1261 (2023)) (Figure RR1), the authors demonstrated that individual RRM domains of NCL (RRM1, RRM2, RRM3, RRM4) showed no detectable binding to *TERRA* RNA. However, strong binding was observed when the combined RRM2-4 domains were tested. This aligns with our findings that individual RRMs of NCL do not bind to EBERs RNA, while the combined RRM2-4 domains are required for effective binding.

These results suggest that the cooperative action of multiple RRMs is essential for NCL’s high-affinity binding to its RNA targets, including both TERRA and EBERs RNA. This cooperative mechanism may represent a unique feature of NCL compared to other RNA-binding proteins

containing multiple RRM, enabling NCL to recognize and bind complex RNA structures with high specificity and affinity.

In summary, the findings from the *TERRA* RNA study support our conclusions regarding NCL’s interaction with EBERs RNA, highlighting the critical role of the combined RRM domains in RNA binding and suggesting a shared mechanism of RNA recognition by NCL across different RNA targets.

Figure 1
Biochemistry. 2023 Apr 4;62(7):1249-1261.

Figure 2
Biochemistry. 2023 Apr 4;62(7):1249-1261.

Figure 3
Biochemistry. 2023 Apr 4;62(7):1249-1261.

EMSA: NCL+ Telomeric Repeat-containing RNA (TERRA)

Figure RR1 | NCL binds to RNA (*TERRA*) through its RNA recognition motifs (RRMs).
(a) Schematic representation of NCL domains, including four RRM (RRM1–4) and the GAR domain.
(b) EMSA showing binding of full-length NCL to RNA (*TERRA*).
(c) EMSA demonstrating binding of NCL fragments containing RRM2–4 to *TERRA*.

(d) EMSA analysis of individual RRM domains (RRM1, RRM2, RRM3, and RRM4) showing their binding affinities to TERRA.

The figure numbers at the bottom of each panel correspond to their placement in the original article. Source: *Biochemistry* 62, 1249–1261 (2023).

2) What is the unique feature of NCL compared with other RNA-binding proteins containing multiple RRM?

The following table summarizes certain features of RRM-containing proteins.

Feature	NCL (Nucleolin)	Other RNA-Binding Proteins
Number of RRMs	4 RRMs	Varies (e.g., ELAVL1 has 3, PTBP1 has 4)
Domain Composition	Central 4 RRMs, N-terminal acidic region, C-terminal GAR domain	Typically contain multiple RRMs but may lack additional domains
Cooperative Binding	Requires multiple RRMs for effective binding to RNA	Many can bind RNA with individual RRM, but multiple RRMs enhance affinity
Biological Functions	Involved in ribosome biogenesis, transcription regulation, and RNA processing	Functions vary widely; often specialized for specific RNA metabolism tasks
RNA Binding Specificity	Binds diverse RNAs including pre-rRNA and mRNAs	Typically have specific targets based on sequence or structure

Main references

- Khan, Y., Azam, T., Sundar, J. S., Maiti, S. & Ekka, M. K. Biophysical Characterization of Nucleolin Domains Crucial for Interaction with Telomeric and TERRA G-Quadruplexes. *Biochemistry* 62, 1249–1261 (2023).
- Corley, M., Burns, M. C. & Yeo, G. W. How RNA binding proteins interact with RNA: molecules and mechanisms. *Mol Cell* 78, 9–29 (2020).
- Doron-Mandel, E., Koppel, I., Abraham, O., Rishal, I., Smith, T. P., Buchanan, C. N., Sahoo, P. K., Kadlec, J., Oses-Prieto, J. A., Kawaguchi, R., Alber, S., Zahavi, E. E., Di Matteo, P., Di Pizio, A., Song, D., Okladnikov, N., Gordon, D., Ben-Dor, S., Haffner-Krausz, R., Coppola, G., Burlingame, A. L., Jungwirth, P., Twiss, J. L. & Fainzilber, M. The glycine arginine-rich domain of the RNA-binding protein nucleolin regulates its subcellular localization. *The EMBO Journal* 40, e107158 (2021).
- Afroz, T., Cienikova, Z., Cléry, A. & Allain, F. H. T. in *Methods in Enzymology* 558, 235–278 (Elsevier, 2015).

2. The authors declared that they performed NCL CLIP-seq. However, from the procedure in the Methods section titled “CLIP-seq”, it seems to be an incorrect CLIP-seq assay, which normally requires UV-crosslinking of the cells to fix RNA-proteins interactions before IP. How many sequencing reads are there for two replicates of CLIP-seq data? Based on Fig. 7j, it seems to be a failed CLIP-seq assay.

1) Methodology Clarification

We sincerely appreciate the opportunity to clarify our methodology and thank the reviewer for their insightful comments regarding our approach. We apologize for any confusion caused by our imprecise use of the term "CLIP-seq."

Our study investigates the interaction between EBERs and NCL, specifically within the chromatin context. Initially, we demonstrated the direct interaction between EBERs and NCL through in vitro EMSA experiments. However, we aimed to further explore this interaction in vivo, focusing on its role within chromatin-associated complexes.

To address this specific research question, we employed a dual crosslinking approach using formaldehyde and EGS, consistent with the ChIA-PET and RDD sample preparation methods used in this study. This standard method captures chromatin-associated interactions while preserving chromatin integrity and enabling the simultaneous capture of protein-DNA, protein-protein, and protein-RNA interactions (see Figure RR2a). After crosslinking, we conducted immunoprecipitation with an NCL-specific antibody to enrich for chromatin-associated complexes, followed by RNA analysis to identify associated RNAs.

While our method shares conceptual similarities with CLIP-seq—such as crosslinking, immunoprecipitation, and RNA detection (Figure RR2b)—it is specifically designed to study chromatin-associated interactions rather than direct RNA-protein binding sites (Figure RR2c). To ensure consistency with other chromatin interaction assays utilized in our study (e.g., RDD and ChIA-PET), we employed the same chromatin preparation protocol.

To avoid further confusion, we have revised the manuscript to describe this method as an "NCL-interacted chromatin RNA assay" (termed NCL-RNA), emphasizing its focus on chromatin-bound complexes. Furthermore, we recognize that results obtained using our approach may differ from those of standard CLIP-seq due to variations in crosslinking strategies and experimental design (Figure RR2d).

d Comparison of CLIP-seq and the manuscript's method

Feature	CLIP-seq	NCL-interacted chromatin RNA (Manuscript Method)
Crosslinking	UV crosslinking	Dual-crosslinking (formaldehyde and EGS), consistent with 3D genome mapping technologies (Hi-C, ChIA-PET, et al.)
Target	Direct RNA-protein interactions	RNA-protein-DNA complexes in chromatin
Specificity	High for direct RNA-protein binding	Captures broader chromatin RNA-associated interactions
Resolution	Single-nucleotide resolution	Lower resolution, broader regions
RNA isolation	RNase treatment to isolate protected fragments	RNase treatment to isolate protected chromatin-RNA
Protein enrichment	Immunoprecipitation of specific RNA-binding proteins	Immunoprecipitation of specific chromatin complexes with RNA
Binding site identification	Precise identification of direct binding sites	Focuses on protein-specific chromatin-associated RNA
Applications	Direct RNA-protein interactions, exact binding sites	Direct and indirect RNA-protein interactions
Coverage	May have lower transcriptome coverage	Potentially higher coverage of chromatin-associated RNAs

Figure RR2 | Comparison of standard CLIP-seq and the NCL-interacted chromatin RNA assay (NCL-RNA). (a) Schematic of the chromatin interaction assay used in this study. This approach employs dual crosslinking with

formaldehyde and EGS to preserve chromatin integrity and capture chromatin-associated complexes, including protein-DNA, protein-protein, and protein-RNA interactions. Immunoprecipitation with a specific antibody enriches chromatin-associated complexes, followed by sequencing to identify RNAs or proteins associated with the chromatin context.

(b) Schematic representation of the standard CLIP-seq workflow. CLIP-seq involves UV crosslinking to capture direct RNA-protein interactions, followed by immunoprecipitation of the protein of interest and RNA sequencing to identify RNA binding sites. This method is optimized for mapping direct RNA-protein interactions at high resolution.

(c) Conceptual illustration of the chromatin-associated RNA immunoprecipitation assay, highlighting its focus on chromatin-bound RNA-protein interactions.

(d) Comparison of the features of standard CLIP-seq and the NCL-interacted chromatin RNA assay. Key differences include the crosslinking strategy, target specificity, resolution, and application, with the latter designed to study chromatin-associated RNA-protein interactions in a broader chromatin context.

2) Sequencing Reads for Two Replicates of CLIP-seq Data

The total number of sequencing reads for the two replicates of the CLIP-seq (termed NCL-RNA) is 89,976,578, comprising 89,933,264 host reads (49,118,443 in replicate 1 and 40,814,821 in replicate 2) and 43,314 EBV reads (20,474 in replicate 1 and 22,840 in replicate 2). RNA-seq data was used as a control for comparison.

Sequencing read counts for CLIP-seq (NCL-RNA) and RNA-seq data across two replicates:

Library_seq_ID	NCL-RNA	NCL-RNA	NCL-RNA	NCL-RNA	RNA-seq	RNA-seq	RNA-seq	RNA-seq
Library_bio_ID	replicate 1	replicate 1	replicate 2	replicate 2	replicate 1	replicate 1	replicate 2	replicate 2
Reference genome	Host	EBV	Host	EBV	Host	EBV	Host	EBV
Total read pairs	49,184,443	49,184,443	51,118,019	51,118,019	56,934,775	56,934,775	51,478,057	51,478,057
Uniquely mapped reads	20,474,261	66,802	22,820,949	80,549	51,390,323	40,790	46,335,988	36,990
% Uniquely mapped reads	41.63%	0.14%	44.64%	0.16%	90.26%	0.07%	90.01%	0.07%
Number of splices	1,845,272	2,925	1,883,026	4,870	23,004,993	7,924	19,604,861	6,500
Number of splices:Annotated(sjdb)	1,844,130	2,915	1,881,821	4,843	23,004,336	7,921	19,604,003	6,500
Number of splices:GT/AG	1,809,323	2,819	1,837,155	4,738	22,796,092	7,792	19,426,406	6,409
Number of splices:GC/AG	22,624	42	26,937	56	159,397	2	138,097	2
Number of splices:AT/AC	1,944	-	1,543	-	22,186	-	18,014	-
Number of splices: Non-canonical	11,381	64	17,391	76	27,318	130	22,344	89
Number of reads mapped to multiple loci	16,991,492	13,460	15,322,543	19,703	4,546,821	901	4,008,954	635
% of reads mapped to multiple loci	34.55%	0.03%	29.97%	0.04%	7.99%	0.00%	7.79%	0.00%
Number of reads mapped to too many loci	543,885	122,339	486,092	150,080	42,175	4,760	26,247	5,601
% of reads mapped to too many loci	1.11%	0.25%	0.95%	0.29%	0.07%	0.01%	0.05%	0.01%

3) Concerns Regarding Fig. 7j

We appreciate the reviewer's feedback regarding Fig. 7j and would like to clarify its purpose and interpretation. The primary aim of our study is to identify RNAs bound by NCL, with a specific focus on EBV-related RNAs. Fig. 7j highlights the expression levels of EBV RNAs detected in the NCL-RNA assay. These results were further validated by comparison with RNA-seq data to confirm the enrichment and interaction of NCL with these RNAs (Figure RR3a-c).

It is important to note that Fig. 7j only presents **EBV RNAs**, while host RNAs, which were also successfully detected and analyzed, are not included in this figure and will be described later. The analysis employs strategies commonly used in chromatin RNA-protein interaction studies (Figure RR3d-e). Additionally, a volcano plot was used to visualize the differential enrichment of RNAs (Figure RR3c), demonstrating that EBERs are enriched among NCL-bound RNAs. We

hope this explanation addresses the reviewer's concerns and demonstrates the validity of the NCL-RNA assay in identifying NCL-interacting RNAs.

Nature Biotechnology volume 35, pages940–950 (2017)

Proc Natl Acad Sci U S A. 2011 Dec 20;108(51):20497-502

Figure RR3 | Chromatin-associated RNAs bound by NCL

(a) Genome browser view showing RNA-seq tracks of EBERs ncRNA and LMP-2 mRNA, highlighting their interaction with NCL-RNA. The y-axis represents normalized read density.

(b) Enrichment analysis of EBERs ncRNA, depicted as the ratio of RPKM (reads per kilobase per million mapped reads) of EBERs compared to other mRNAs. Statistical significance was determined using a two-tailed t-test ($P = 0.01$). Bars represent the mean values, and error bars indicate the standard deviation (SD). These results demonstrate that NCL-RNA is significantly enriched for EBERs compared to other RNAs.

(c) Volcano plot showing fold-change and p-values from differential gene expression analysis using edgeR or DESeq2. Each dot represents a transcript, with EBERs highlighted as significantly enriched.

(d, e) Visualization of chromatin-RNA-protein interactions using approaches commonly employed in chromatin-associated RNA studies, such as GRID-seq (d) and CHART-seq (e).

3. The CLIP-seq data are not fully analyzed. Generally, people perform peak calling to identify RBP binding sites enriching CLIP-seq signals, followed by motif analysis to characterize RBP's RNA binding specificity. In Fig. 7j, there are only 20 points, most of which seem to have no enrichment of CLIP-seq over RNA-seq. Does it mean that NCL has very few RNA targets? Does NCL bind to host RNAs? Can the NCL binding motif(s) be derived from the CLIP-seq data?

We appreciate the reviewer's insightful comments regarding our analysis of the CLIP-seq data. Below, we address the specific points raised.

1) Analysis of CLIP-seq Data

We fully acknowledge the importance of performing peak calling and motif analysis to identify RNA-binding protein (RBP) binding sites and characterize RNA-binding specificity. Our analysis demonstrates that NCL interacts with a substantial number of RNA targets, including both host and viral RNAs (Figure RR3-4).

In Fig. 7j, we intentionally focused on the enrichment of NCL binding to EBERs to highlight the interaction between NCL and EBV-related RNAs, which is the primary focus of this manuscript. Therefore, Fig. 7j should not be interpreted as evidence that NCL has very few RNA targets. Instead, it reflects the targeted scope of this figure in demonstrating the interaction between NCL and EBERs.

2) Host RNA Binding

Yes, NCL does bind to host RNAs, as evidenced by our CLIP-seq (NCL-RNA) analysis (Figure RR4). However, the primary aim of this study is to establish and characterize the binding relationship between NCL and EBV RNAs, particularly EBERs. While we recognize the importance of NCL's interactions with host RNAs, these results initially were not included in the main text to maintain the manuscript's focus. They have now been added.

Fig. 7j exclusively presents EBV RNAs, while host RNAs, which were also successfully detected and analyzed, are described in Extended Data Fig. 10 in the revised manuscript. A volcano plot (Figure RR3a) highlights the enrichment of NCL-bound RNAs, with significantly enriched RNAs shown as red points ($P < 0.05$). A scatter plot (Figure RR3b) further provides the original expression levels (FPKM) for NCL-RNA and RNA-seq, complementing the volcano plot. These NCL-bound host RNAs are primarily enriched in rRNA, snRNA, snoRNA, and mRNA, consistent with previously reported NCL functions.

We hope this addresses the reviewer's concerns and demonstrates the validity of the NCL-RNA assay in identifying both EBV and host RNAs bound to NCL.

3) Deriving NCL Binding Motifs

Nucleolin (NCL) is known to bind RNAs, particularly those containing structured regions such as stem-loop or hairpin structures. NCL exhibits a strong affinity for highly structured RNAs, including ribosomal RNAs (rRNAs), small nucleolar RNAs (snoRNAs), and telomerase RNA (TERC). However, when we attempted to perform peak calling on the NCL-RNA data using MACS2, no significant peaks were identified. This result suggests that the NCL-RNA assay may not produce the sharp, localized enrichment patterns typically required for peak calling, likely due to the broader and more diffuse nature of chromatin-associated RNA-protein interactions. While RNAfold could be used to predict secondary structures and identify potential stem-loop regions, we have not included a detailed structural analysis in this manuscript to maintain the study's scope and focus. Future studies could further explore this approach to uncover novel features or confirm known ones, providing deeper insights into NCL's RNA-binding specificity and regulatory roles.

References:

- Khan, Y., Azam, T., Sundar, J. S., Maiti, S. & Ekka, M. K. Biophysical Characterization of Nucleolin Domains Crucial for Interaction with Telomeric and TERRA G-Quadruplexes. *Biochemistry* 62, 1249–1261 (2023).
- Ginisty, H., Amalric, F. & Bouvet, P. Nucleolin functions in the first step of ribosomal RNA processing. *EMBO J* 17, 1476–1486 (1998).

Figure RR4 | NCL associated host RNA..

(a) Volcano plot illustrating the enrichment of NCL-bound RNAs identified by the NCL-RNA assay compared to RNA-seq. The x-axis represents the \log_2 fold change (NCL-RNA vs. RNA-seq), while the y-axis represents the $-\log_{10}$ P-value. RNAs enriched in the NCL-RNA data are shown on the right side of the plot, with significantly enriched RNAs highlighted as red points above the horizontal dashed line ($P < 0.05$). The bracket marks the region containing RNAs with significant NCL enrichment. Blue points on the left represent RNAs depleted in the NCL-RNA assay. The total number of genes in each category is indicated ($n = 759, 2,540,$ and $11,688$).

(b) Scatter plot showing the gene expression levels (FPKM) of NCL-RNA and RNA-seq, corresponding to (a). The x-axis represents RNA-seq expression levels, and the y-axis represents NCL-RNA expression levels. The scatter plot provides additional context by displaying the original expression levels, which are not visible in the volcano plot. Red and blue points correspond to significantly enriched and depleted RNAs, respectively, as shown in (a). The total number of genes in each category is indicated ($n = 759, 2,540,$ and $11,688$).

Minor points:

4. Fig. R3b or Fig. 7I. There are two different panels labeled with the same name, NCL-RRM3. Please double-check which one is NCL-RRM4.

Thank you for bringing this labeling issue to our attention. After carefully reviewing the data and figures, we confirm that the panel labeled as NCL-RRM3 at the bottom should actually be NCL-RRM4. We have thoroughly checked all figure labels to ensure accuracy and have corrected this error. The updated figure will be included in the revised manuscript. We sincerely appreciate your careful review and attention to detail.

5. For Fig. 6h, the authors should include p-values for the differentially expressed genes. Typically, this analysis is presented as a volcano plot with fold-change and p-values from differential gene expression tests by edgeR or DEseq2 packages.

We thank the reviewer for their suggestion to include p-values for the differentially expressed genes and to present the data as a volcano plot. As described and shown earlier in the manuscript, Fig. 6h clearly demonstrates the upregulation of genes in a straightforward and focused manner. While a volcano plot with fold-change and p-values is indeed another way to present the data, it provides a more comprehensive but also more complex visualization (Figure RR3c, Figure RR4). Each method has its own strengths, and after careful consideration, we have opted to retain the current presentation style in the manuscript to maintain clarity and focus. We hope the reviewer understands our rationale for this decision.

6. The method descriptions of the CLIP-seq procedure are not correct English sentences. Please rewrite the CLIP-seq section.

We thank the reviewer for pointing out the issues in the CLIP-seq method descriptions. We have carefully revised the section to ensure it is written in clear and correct English. Below is a partial example of the updated description included in the revised manuscript; for details, please refer to the full revised manuscript.

“NCL-interacted chromatin RNA assay (termed NCL-RNA)

Chromatin complexes preparation

To investigate NCL-interacted chromatin RNAs, we employed a dual crosslinking strategy using formaldehyde and EGS to preserve chromatin integrity and capture chromatin-associated complexes, as described in the RDD and ChIA-PET methods. Crosslinked chromatin was subjected to immunoprecipitation using NCL-specific antibodies to enrich chromatin-associated complexes. The NCL enriched complexes were then processed with TRIzol reagent to isolate RNA for downstream analysis. This method enables the identification of RNAs associated with NCL within the chromatin context.”

We have updated the methods section to reflect the use of the NCL-RNA assay, and corresponding corrections have been made throughout the manuscript. We appreciate the reviewer's valuable feedback and have taken steps to improve the clarity, accuracy, and readability of the text.

Reviewer #2 (Remarks to the Author):

The authors have addressed most of the previous concerns raised. A few remaining issues need to be addressed.

1. There are still very few references to previous published work on EBV epigenome and tethering. This is unfortunate oversight since the previous work has some different conclusions that should be discussed. The findings in the present study differ considerably from other findings, particularly for EBV-Human genome interactions. The authors should cite additional previous studies on EBV-Human interactions and how or why their conclusions differ from these previous studies.

We appreciate the reviewer's insightful comment regarding the need to reference previous studies on the EBV epigenome and tethering, as well as the importance of discussing how our findings compare to and differ from these studies. In response, we have included additional references to prior work and expanded the discussion in the revised manuscript to address these points. Furthermore, we have conducted a side-by-side comparison with several representative articles. Given the length of this comparison, we have included it as an appendix at the end of this document for the reviewer's convenience.

1) EBV epigenome

Previous studies, particularly those utilizing ChIP-seq, have provided valuable insights into the histone modifications associated with the EBV genome. For example, it has been shown that histone modifications linked to facultative heterochromatin (H3K27me3) or constitutive heterochromatin (H3K9me3) are generally low throughout the EBV genome. These findings suggest that the EBV genome is chromatinized with a density similar to that of the host genome but remains more open compared to host chromatin. These observations are consistent with our ATAC-seq results, which also indicate that the EBV genome exhibits a relatively open chromatin state. Furthermore, we conducted a direct comparison of the chromatin state of the EBV genome with that of the host genome, further confirming the higher openness of the EBV genome.

2) EBV-Human genome interactions

In this study, we employed state-of-the-art 3D genome mapping techniques, including protein-associated chromatin interaction analysis with paired-end tag sequencing (ChIA-PET), RNA-associated chromatin interaction technique (RDD), and super-resolution microscopy, to delineate the spatial architecture of EBV in human lymphoblastoid cells. We systematically analyzed EBV-to-EBV (E-E), EBV-to-Host (E-H), and Host-to-Host (H-H) interactions linked to host proteins and EBV RNAs.

For protein-mediated chromatin interactions, we focused on the 3D structure of EBV itself, particularly chromatin interactions associated with key factors such as CTCF and RNA Pol II. We resolved EBV's CCDs and RAIDs. These structures are anchored and tethered together to form a platform, which we visualized using 3D models. Additionally, we identified three interaction hotspots (HS1, HS2, HS3) and characterized their regulatory functions in E-H interactions with the host genome. These structural insights lay the foundation for subsequent studies presented later in this manuscript.

Previous studies have also explored EBV-Human genome interactions using techniques such as Hi-C, 4C-seq, or HiChIP. For example, some studies have indirectly analyzed EBV regulome by associating RNA Pol II-mediated host chromatin interaction (H-H) sites with EBV key protein ChIP-seq data. Others have used Hi-C data to investigate EBV-host interactions, revealing significant variability in interactions depending on the EBV latency stage of the cells. Additionally, studies employing H3K27ac HiChIP have analyzed host chromatin structural changes during different stages of EBV infection. Details of these studies are provided in the appendix for reference.

We would like to emphasize that while ChIA-PET and HiChIP are similar methods, the differences in cell types or tissues used in these studies may account for the discrepancies in results. Hi-C and 4C-seq, on the other hand, capture all protein-associated chromatin interactions, unlike ChIA-PET and HiChIP, which enrich for specific transcription factors. When analyzing EBV chromatin structure, some structural features may overlap between these methods, while others may differ. This highlights the importance of using complementary approaches to fully resolve the complex chromatin architecture of EBV and its interactions with the host genome.

We hope that the additional references, expanded discussion, and side-by-side comparison address the reviewer's concerns. However, we would greatly appreciate further guidance or suggestions from the reviewer on how we might improve this section or better address these important points.

Reference:

- Wang, C., Liu, X., Liang, J., Narita, Y., Ding, W., Li, D., Zhang, L., Wang, H., Leong, M. M. L., Hou, I., Gerdt, C., Jiang, C., Zhong, Q., Tang, Z., Forney, C., Kottyan, L., Weirauch, M. T., Gewurz, B. E., Zeng, M., Jiang, S., Teng, M. & Zhao, B. A DNA tumor virus globally reprograms host 3D genome architecture to achieve immortal growth. *Nat Commun* 14, 1598 (2023).
- D, M., G, N., A, K., S, P.-A., Lb, C. & I, T. The three-dimensional structure of the EBV genome plays a crucial role in regulating viral gene expression in EBVaGC. *Nucleic acids research* 51, (2023).
- Arvey, A., Tempera, I., Tsai, K., Chen, H.-S., Tikhmyanova, N., Klichinsky, M., Leslie, C. & Lieberman, P. M. An Atlas of the Epstein-Barr Virus Transcriptome and Epigenome Reveals Host-Virus Regulatory Interactions. *Cell Host & Microbe* 12, 233–245 (2012).
- Morgan, S. M., Tanizawa, H., Caruso, L. B., Hulse, M., Kossenkov, A., Madzo, J., Keith, K., Tan, Y., Boyle, S., Lieberman, P. M. & Tempera, I. The three-dimensional structure of Epstein-Barr virus genome varies by latency type and is regulated by PARP1 enzymatic activity. *Nat Commun* 13, 187 (2022).
- Jiang, S., Zhou, H., Liang, J., Gerdt, C., Wang, C., Ke, L., Schmidt, S. C. S., Narita, Y., Ma, Y., Wang, S., Colson, T., Gewurz, B., Li, G., Kieff, E. & Zhao, B. The Epstein-Barr Virus Regulome in Lymphoblastoid Cells. *Cell Host & Microbe* 22, 561-573.e4 (2017).

- Kim, K.-D., Tanizawa, H., De Leo, A., Vladimirova, O., Kossenkov, A., Lu, F., Showe, L. C., Noma, K. & Lieberman, P. M. Epigenetic specifications of host chromosome docking sites for latent Epstein-Barr virus. *Nat Commun* 11, 877 (2020).
- Kim, K.-D. & Lieberman, P. M. Viral remodeling of the 4D nucleome. *Exp Mol Med* 56, 799–808 (2024).
- Wang, L., Laing, J., Yan, B., Zhou, H., Ke, L., Wang, C., Narita, Y., Zhang, Z., Olson, M. R., Afzali, B., Zhao, B. & Kazemian, M. Epstein-Barr Virus Episome Physically Interacts with Active Regions of the Host Genome in Lymphoblastoid Cells. *J Virol* 94, e01390-20 (2020).
- Ding, W., Wang, C., Narita, Y., Wang, H., Leong, M. M. L., Huang, A., Liao, Y., Liu, X., Okuno, Y., Kimura, H., Gewurz, B., Teng, M., Jin, S., Sato, Y. & Zhao, B. The Epstein-Barr Virus Enhancer Interaction Landscapes in Virus-Associated Cancer Cell Lines. *J Virol* 96, e0073922 (2022).
- Arvey, A., Tempera, I. & Lieberman, P. M. Interpreting the Epstein-Barr Virus (EBV) epigenome using high-throughput data. *Viruses* 5, 1042–1054 (2013).

2. Provide the source of antibodies (catalogue numbers) for the ChIP and ChIA-PET experiments needs to be provided in Methods.

We appreciate the reviewer's comment. While the detailed information regarding the antibodies (including catalogue numbers) was already provided in the reporting summary, we have now included the specific details in the Methods section for clarity. Thank you for bringing this to our attention.

3. Fig 1E. EBV DNA “secreted” from B95.8 cells should be in virions and lack histones. What is the meaning of this “secreted” EBV. I think this image needs better explanation of preparation and validation, or should be removed from the manuscript.

We thank the reviewer for the insightful comment. To clarify, after isolating virions from B95.8 cells, we performed washing, crosslinking, filtration, and concentration steps. The sample was then divided into two parts: one was retained untreated, while the other underwent protease digestion to remove proteins, followed by purification to obtain pure DNA. Both samples were subsequently analyzed using the BALM assay, as described in the manuscript.

In response to the reviewer's concern, we have updated the figure and revised the labeling in the manuscript for clarity. Specifically, the untreated sample is now labeled as "EBV genome," while the protease-treated sample is labeled as "EBV DNA." The results clearly demonstrate that the EBV genome exhibits a circular structure, while the purified DNA appears linear, indicating that the EBV genome contains associated proteins. Therefore, we have retained the figure with updated labeling and have clarified this process in the manuscript.

4. Also, it is not stated whether the authors are investigating EBER1 or EBER2. These are not duplicated genes, and may have very different functions and interactomes. This information may be embedded in the Supplementary data but should be explicitly stated in the Main Text, Figure, and Figure Legends.

We sincerely thank the reviewer for the insightful feedback. We fully understand the importance of specifying whether we are investigating EBER1 or EBER2. We have addressed this concern as follows:

1) Specification of EBERs

In our study, we investigated both EBER1 and EBER2. Due to their close genomic proximity, we employed a CRISPR interference (CRISPRi) approach that simultaneously targets both ncRNAs. This strategy allowed us to assess their combined roles in chromatin interactions and gene regulation. While previous studies often refer to these ncRNAs collectively as "EBERs," we now explicitly clarify our focus on both EBER1 and EBER2 in the manuscript.

2) Methodological Considerations

In our chromatin interaction studies, we used probes targeting both EBER1 and EBER2 to enrich chromatin DNA interactions and construct sequencing libraries. This approach reflects our intent to explore their combined influence on chromatin dynamics.

3) Functional Similarities and Differences

While EBER1 and EBER2 share overlapping functions, such as immune evasion and modulation of host gene expression, they also exhibit distinct characteristics. For example, EBER1 has been shown to play a more prominent role in certain cellular processes compared to EBER2. However, our study focuses on their collective impact rather than isolating their individual effects.

4) Explicit Mention in the Manuscript

In the EMSA experiments, we used EBER1 as an RNA representative, while other experiments included both EBER1 and EBER2. These details have been clearly indicated in the revised manuscript.

References:

- Lee, N. The many ways Epstein-Barr virus takes advantage of the RNA tool kit. *RNA Biology* 18, 759–766 (2021).
- Notarte, K. I., Senanayake, S., Macaranas, I., Albano, P. M., Mundo, L., Fennell, E., Leoncini, L. & Murray, P. MicroRNA and Other Non-Coding RNAs in Epstein–Barr Virus-Associated Cancers. *Cancers* 13, 3909 (2021).

5. Fig. 7. The authors conclude there are multiple “conformations” of Nucleolin binding to EBERs. But it is not clear if there are more than one EBER RNA in the EMSA which could also account for the two different EMSA mobility patterns that are interpreted as “conformations”.

We thank the reviewer for the insightful comment regarding the interpretation of the EMSA results. To clarify, the RNA used in our experiments was synthesized specifically as the EBER1 sequence, and no other RNA species were present in the reaction. The observation of multiple mobility patterns in the EMSA is due to the use of native PAGE, which preserves RNA secondary structures. Consequently, EBER1 RNA can adopt different structural states, such as stem-loop and linear conformations, resulting in variations in mobility.

We hope this clarification adequately addresses the reviewer's concern.

Reviewer #3 (Remarks to the Author):

The current version looks good to me.

We sincerely thank Reviewer #3 for their positive feedback and for taking the time to review our manuscript. We are pleased to hear that the current version meets your expectations.

Attachment (for Reviewer #2): *next page*

Comparison between this study and the previous study on the EBV epigenome and EBV-Human genome interactions.

We compare the previous research on the EBV epigenome and tethering, with findings from this study highlighted in yellow and findings from previous studies highlighted in blue.

Note: For better understanding and annotation, newly added content in the manuscript is marked in red.

Comparison of the EBV Epigenome and Tethering in **This study** and **Previous Studies**

--1.2

Previous study

Decitabine disrupts EBV genomic epiallele DNA methylation patterns around CTCF binding sites to increase chromatin accessibility and lytic transcription in gastric cancer

Smith-Peterson AL¹, Liu B², Garcia-Cano J¹, Chen H³, Kelly K⁴, Samanthi S⁵, Sollen D⁶, Maestri J⁷, Josef W⁸, Andrew K⁹, Georgia N¹⁰, Benjamin G¹¹, Paul M¹², Lisa T¹³

Note:
CTCF binding peak consistency in this study and previous studies

Cell Host & Microbe
Resource

An Atlas of the Epstein-Barr Virus Transcriptome and Epigenome Reveals Host-Virus Regulatory Interactions

Aaron Arvey,¹ Babo Tempura,² Kevin Tsai,² Hong-Shen Chen,² Nadezhda Tikhonova,² Michael Kikvidze,² Christine Leslie,² and Paul M. Lieberman^{1*}

¹Computational Biology Program, Memorial Sloan-Kettering Cancer Center, New York, NY 10026, USA
²The Walter and Teresa Steyer Institute, Philadelphia, PA 19104, USA

*Correspondence: paul.lieberman@mskcc.org (P.M.L.)
http://dx.doi.org/10.1016/j.chom.2012.06.008

Note:
The inconsistency in CTCF binding peaks between this study and previous studies may be attributed to the comprehensive integration of large-scale database data in this study, which could result in variations in analysis outcomes.

Note:
CTCF binding peak consistency in this study and previous studies

Comparison of the EBV-Human interactions in This study and Previous Studies ---2.1

183 The platform of EBV 3D structure frequently contacts with the host chromatin
 184 Our research further uncovers that RNAPII and CTCF extend beyond their roles in the EBV 3D
 185 structure; they are instrumental in mediating interactions between EBV and host cells (E-H, Fig.
 186 2a). The capture of 194 E-H contacts, concentrated at three critical hotspots (HS1, HS2, and HS3;
 187 Fig. 2b) at the EBV 3D platform structure, is characterized by open chromatin configurations and
 188 a high density of histones and proteins, highlighting their regulatory potential.

Note:
 This study identifies direct EBV-host interactions captured by ChIA-PET, specifically those involved in the formation of chromatin loops or clusters, totaling 466 interactions. Among these, 194 are enriched in the HS1-3 region. In contrast, previous research focused on an association analysis, linking LCL RNAPII ChIA-PET enhancers occupied by any of the four EBNA5 or five NF-kB subunits.

Comparison of the EBV-Human interactions in This study and Previous Studies

---2.2

This study

Previous study

AMERICAN SOCIETY FOR MICROBIOLOGY Journal of Virology VIRUS-CELL INTERACTIONS

Epstein-Barr Virus Episome Physically Interacts with Active Regions of the Host Genome in Lymphoblastoid Cells

Luopin Wang,^a Jun Liang,^b Binyu Yan,^c Hufeng Zhou,^b Liangru Ke,^b Chong Wang,^b Yohei Narita,^b Zonghao Zhang,^d Matthew R. Olson,^e Behdad Afzali,^f Bo Zhao,^g Majid Kazemian^h

resolution and reproducible EBV and human host contact map, we focused on contacts at 10-kb resolution that are shared among two or more biological replicates. We found 15,000 such contacts (Fig. 1C and Table S1), of which nearly half were located in intronic regions, showing a 1.2-fold enrichment compared to all intronic regions genome-wide (Fig. 1D). Contacts also significantly overlapped promoter and exonic regions, but failed to overlap intergenic loci compared to what would be expected given the fractions of all promoters, exons, or intergenic loci in the entire genome (Fig. 1D).

To characterize epigenetic markers at the contact loci, we first sourced 22 publicly available genome-wide histone modification data in GM12878 from the encyclopedia of DNA elements (ENCODE) (31). We used an unbiased approach examining which, if any, histone modifications can distinguish EBV contacts from control regions (i.e., size-matched noncontact loci). This was done by comparing the mean signal intensity of the histone modification in EBV contacts versus control regions using receiver operating characteristic (ROC) and the area under the ROC curves (AUC) (Table S2). Histone markers, including H3K27ac and H3K4me1, that are typically associated with active chromatin regions such as enhancers and promoters had the highest predictive power (AUC ~0.7; Fig. 1E, Table S2) at separating EBV contacts from noncontacts. This was also evident as the mean binding intensity of H3K27ac in EBV contact regions was higher than in control loci (Fig. 1F). Conversely, histone modifications associated with polycomb repression or heterochromatin, including H3K27me3, did not show strong predictive power (AUC ~0.5; Fig. 1G). In addition, genomic loci with active histone markers were ~1.5- to 2.1-fold more enriched in EBV contacts compared to control regions (Fig. 1G). Importantly, for all LCL H3K27ac peaks, ~12% of them had contacts with EBV episomes (Fig. 1G). Similarly, ~8% of all H3K4m1 peaks had contacts with EBV episomes (Fig. 1G). In contrast, significantly fewer contacts were detected between control regions and the EBV episomes. These findings suggest that EBV episomes selectively target active enhancers in LCLs.

Note:

In this study, we performed genomic region enrichment analysis on EBV-host interaction sites identified through RNAPII ChIA-PET data. We found that approximately 50% of these sites are located in intronic regions, and their distribution is consistent with patterns previously identified using Hi-C or 4C-seq data. Additionally, chromatin state analysis revealed that, compared to host-host RNAPII-mediated interactions, EBV-host interactions show weaker enrichment in active regions such as enhancers and promoters. This aligns with the trends observed in the previous study. However, we believe it is premature to conclude that the EBV episome specifically interacts with active regions, and this interpretation is suggested in the previous study.

Comparison of the EBV-Human interactions in This study and Previous Studies

---2.3

Type II EBV latency:

identify the spatial and temporal genome architectural changes **during EBV B cell transformation.**

RNA Pol II HiChIP
Hi-C
focus on: host cell 3D genome organization

Note:

This study explores how Epstein-Barr virus (EBV) reshapes the 3D genome architecture of host B lymphocytes to drive their immortalization and transformation. Using techniques like Hi-C, HiChIP, and RNA-seq, the researchers found that **EBV infection induces significant reorganization of the host genome**, including changes in chromatin compartments, contact domains, and enhancer-promoter loops. Viral transcription factors, such as EBNA2, EBNA3A, and EBNA3C, play key roles in regulating these structural changes by interacting with host factors like CTCF and RBPJ. EBV also introduces viral super-enhancers (ESEs) that activate host oncogenes, such as MYC and CDKN2A/B, promoting cell proliferation and survival. These findings reveal how EBV manipulates host chromatin architecture to establish latency and drive oncogenesis, providing insights into the molecular mechanisms of EBV-associated cancers and potential therapeutic targets.

Dear Dr. Zheng,

Thank you for submitting your manuscript for consideration by the EMBO Journal. I sincerely apologise for the protracted assessment process due to delays in reviewer report submission and follow-up discussions within the team. I have now received comments from the initial reviewer #1, who is happy with the publication of the manuscript after appropriate toning down of the conclusions regarding the specificity of nucleolin interactions throughout of the manuscript, including the abstract. Additionally, there remain a few editorial points that need addressing before I can extend official acceptance of the manuscript:

1. Please submit up to five keywords.
2. Please submit a complete author checklist, which you can download from our author guidelines (<https://www.embopress.org/pb-assets/embo-site/EMBO%20Press%20Author%20Checklist-1642513524327.xlsx>). Please insert information in the checklist that is also reflected in the manuscript. The completed author checklist will also be part of the Review Process File.
3. Please upload the main and EV figures as individual production quality figure files in the .eps, .tif, or .jpg format (one file per figure).
4. The extended data figures should be renamed "Figure EV1- 10". Their legends should be placed after the main figure legends, under the heading "Expanded View Figure Legends".
5. Please check that the funding information is correct and identical both in the manuscript and our online system. Currently, grants 2022YFC340040, 62375116, 32250710678, CYJ20220818100416036, KQTD2020082011301202 and 2022A1515011174 are missing in our online system.
6. Please merge "Data availability" and "Code availability" sections. In the "Data availability" section, please add resolvable links to the datasets. More information about the format of this section can be found here: <https://www.embopress.org/page/journal/14602075/authorguide#dataavailability>.
7. CRedit has replaced the traditional author contributions section because it offers a systematic, machine-readable author contributions format that allows for more effective research assessment. Please remove the Authors Contributions from the manuscript and use the free text boxes beneath each contributing author's name in our online submission system to add specific details on the author's contribution. More information is available in our guide to authors.
8. Please rename "Competing interests" section into "Disclosure and competing interests statement" (further info: <https://www.embopress.org/page/journal/14602075/authorguide#conflictsofinterest>).
9. Please update references according to The EMBO Journal style - it should be alphabetically ordered; where there are more than 10 authors on a paper, the first 10 should be listed, followed by 'et al.' Please see further information here: <https://www.embopress.org/page/journal/14602075/authorguide#referencesformat>
10. Please upload supplementary videos and rename them into Movie EV1-EV3 and update the callouts accordingly. The legends/descriptions should be removed from the manuscript text file and zipped with each movie file. Further information is available here: <https://www.embopress.org/page/journal/14602075/authorguide#expandedview>
11. Please place "Methods" section after "Discussion".
12. All Materials and Methods need to be described in the main text using our 'Structured Methods' format. According to this format, the Methods section includes a Reagents and Tools Table (listing key reagents, experimental models, software and relevant equipment and including their sources and relevant identifiers) followed by a Methods section describing the methods, ideally using a step-by-step protocol format. The aim is to facilitate adoption of the methodologies across labs. Please download and fill our Reagents and Tools Table template (.docx), which you can find in our author guidelines: <https://www.embopress.org/page/journal/14602075/authorguide#structuredmethods>
When submitting your revised manuscript, please do not include the Reagents and Tools Table in the Methods section of the manuscript but upload it as a separate file choosing the file type "Reagent Table".
An example of a Method paper with Structured Methods can be found here: <https://www.embopress.org/doi/10.15252/msb.20178071>.
13. Supplementary figures and tables should be compiled in a PDF labelled "Appendix" and renamed "Appendix Figure S1" etc. and "Appendix Table S1" etc. The list of files should be removed from the manuscript text. The appendix file should have a table of contents with page numbers.
14. Please submit the completed source data checklist and the source data files as requested by our source data coordinator. Please remove the list of files from the manuscript text.
15. In our standard image integrity check, we noticed image reuse between Figures 1g and EV2d. If this was intentional, please indicate this in the figure legend.
16. Our data editors have flagged the following issues in figure legends that need correcting:
 - Please provide the exact p values in the legends of figures 5E, 6F, G; extended data figures 8H.
 - Please indicate the statistical test used for data analysis in the legends of figures 7C, extended data figure 10D.
 - Please define the box plots in terms of minima, maxima, percentile in the legends of figures 1B, 4F, G; 5E, I, O; 7G; extended data figures 1C, 2G, 8H, I.
 - Please define the box plots in terms of minima, maxima, centre, bounds of box and whiskers, and percentile in the legends of figures 6F, G; 8B.
 - Please provide information on the nature and number of replicates in the legend of figure 5O.
17. Papers published in The EMBO Journal are accompanied online by a 'Synopsis' to enhance discoverability of the

manuscript. It consists of A) a short (1-2 sentences) summary of the findings and their significance, B) 3-4 bullet points highlighting key results and C) a synopsis image that is 550x300-600 pixels large (width x height, jpeg or png format). You can either show a model or key data in the synopsis image. Please note that the image size is rather small and that text needs to be readable at the final size. Please send us this information together with the revised manuscript.

With best wishes,

leva

leva Gailite, PhD
Senior Scientific Editor
The EMBO Journal
Meyerhofstrasse 1
D-69117 Heidelberg
Tel: +4962218891309
i.gailite@embojournal.org

We realize that it is difficult to revise to a specific deadline. In the interest of protecting the conceptual advance provided by the work, we recommend a revision within 3 months (1st Jul 2025). Please discuss the revision progress ahead of this time with the editor if you require more time to complete the revisions.

Referee #1:

Most of the responses to the questions in the previous revision are reasonable and satisfiable. However, the NCL-RNA targets do not have strong evidence based on the new analyses and results. There are no significant peaks from NCL-RNA sequencing data. The authors propose NCL binds to structured RNAs, but no evidence demonstrates NCL targets have higher RNA structuredness. In Figure RR4b, most of the RNA targets (n=2540) actually have very small FPKM (<1 , $\log_{10}(\text{FPKM}) < 0$) in the RNA-seq sample, suggesting a risk of bias to low-level RNAs. The authors need to tone down the conclusions on NCL-RNA.

The authors addressed the editorial issues.

Dear Meizhen,

Thank you for submitting a reformatted version of your manuscript. I apologise for the delay in the handling of your revision due to holidays here in Germany and the resulting backlog. I have now gone through the revised version, and I am afraid that there remain a couple of editorial aspects as outlined below that still need to be implemented in the manuscript before its acceptance:

1. In our standard source data check, we have noted unexplained numerical duplications in the source data for several figures. I have attached the corresponding files with the detected duplications labelled in colour. Please take a look and correct if needed. A brief explanation would be very helpful - I appreciate that these duplications can also occur due to specific measurement or calculation methods used.
2. In the Author Checklist file, please fill in the column E in every row labelled with "yes" in column D.

Please feel free to contact me if have any questions regarding these final points. I look forward to your input on these aspects. I will meanwhile look into the final textual edits that might be needed from the editorial side.

With best wishes,

Ieva

We realize that it is difficult to revise to a specific deadline. In the interest of protecting the conceptual advance provided by the work, we recommend a revision within 3 months (31st Jul 2025). Please discuss the revision progress ahead of this time with the editor if you require more time to complete the revisions.

The authors addressed the remaining editorial issues.

Dear Meizhen,

Thank you for addressing the final editorial requests. I am now pleased to inform you that your manuscript has been accepted for publication.

Before we forward your manuscript to our publishers, I would like to propose some edits in the manuscript title, abstract and synopsis (please see below and the attached text file). I have also written a short blurb that will accompany the title of your manuscript in our online system. Please let me know if any corrections or adjustments are needed.

Title:

Landscape of the Epstein-Barr virus-host chromatin interactome and gene regulation

Blurb:

Advanced techniques of 3D genome mapping reveal the role of EBV ncRNAs in repression of host-gene expression via the inhibition of RNAPII-associated chromatin loops.

Synopsis:

Epstein-Barr virus (EBV) reshapes 3D genome architecture by hijacking host factors. This study shows that EBV targets inaccessible chromatin via its non-coding RNAs (ncRNAs) and disrupts RNAPII loops to suppress pathways linked to immune evasion and genomic instability.

- Advanced 3D genome mapping uncovers EBV's chromatin architecture and regulatory mechanisms.
- Genome-wide mapping reveals 3D interaction landscapes, including EBV-EBV, EBV-host, and host-host interactions mediated by RNA and protein factors.
- EBV ncRNA-associated chromatin shows symmetrical loop organization.
- EBV ncRNAs and cofactors target low-accessibility chromatin, disrupt RNAPII loops, and regulate genes critical for immune response and cell cycle control.

Finally, we would like to promote your manuscript among the Chinese readership. Therefore, we would like to invite you to prepare a short summary of the manuscript in Chinese (1500-2000 Chinese characters), which we will promote on the WeChat platform 'BioArt' with more than 610,000 followers.

If you are interested in this opportunity, we recommend covering the article very close to its online publication date. Thus, ideally we would very much appreciate if you could send us a draft within the next 7 working days. Please let us know whether or not you would be interested in contributing such a short summary in Chinese.

I have included below some general guidelines on how to prepare a summary and a link to recent examples for your reference. Please let me know if you have any questions about this.

If you have any questions, please do not hesitate to contact the Editorial Office. Thank you for this contribution to The EMBO Journal and congratulations on a nice study!

With best wishes,

Ieva

Ieva Gailite, PhD
Senior Scientific Editor
The EMBO Journal

Meyerhofstrasse 1
D-69117 Heidelberg
Tel: +4962218891309
i.gailite@embojournal.org

General WeChat Summary Guidelines

1. These summary articles are meant to be targeting general audience so please limit the use of specialized technical terms, acronyms and jargon.
2. A summary usually starts with brief background information of the reported work, which is followed by explaining the findings in some detail, and ends with a short review of the conclusions as well as the implications of the work and future directions for the research.
3. The summary should contain a visual abstract, which can be the one provided in the paper.
4. Please provide ONE SINGLE document containing all text and graphical materials, ideally as a Word.docx or .doc file. Please DO NOT provide the document as a .pdf file.
5. Please DO NOT publicly release the document before the paper is officially published online.

Summary Examples

EMBO Journal | 灵珠与魔丸：昆虫miRNA调控病毒感染虫媒和植物的双重作用

EMBO Mol Med | 陈良/舒红兵合作揭示毛花苷C通过STUB1-FOXP3促进抗肿瘤免疫的机制

EMBO Rep | 王一飞/郑楷/刘凯胜团队揭示HDAC6调控cGAS-STING介导的抗病毒免疫的分子机制
